# PIECEWISE LINEAR ACTIVATIONS SUBSTANTIALLY SHAPE THE LOSS SURFACES OF NEURAL NETWORKS

**Fengxiang He**[*], **Bohan Wang**[*†] **& Dacheng Tao**
UBTECH Sydney AI Centre, School of Computer Science, Faculty of Engineering
The University of Sydney
Darlington, NSW 2008, Australia
{fengxiang.he, dacheng.tao}@sydney.edu.au, bhwangfy@gmail.com

## ABSTRACT

Understanding the loss surface of a neural network is fundamentally important to the understanding of deep learning. This paper presents how piecewise linear activation functions substantially shape the loss surfaces of neural networks. We first prove that *the loss surfaces of many neural networks have infinite spurious local minima* which are defined as the local minima with higher empirical risks than the global minima. Our result demonstrates that the networks with piecewise linear activations possess substantial differences to the well-studied linear neural networks. This result holds for any neural network with arbitrary depth and arbitrary piecewise linear activation functions (excluding linear functions) under most loss functions in practice. Essentially, the underlying assumptions are consistent with most practical circumstances where the output layer is narrower than any hidden layer. In addition, the loss surface of a neural network with piecewise linear activations is partitioned into multiple smooth and multilinear cells by nondifferentiable boundaries. The constructed spurious local minima are concentrated in one cell as a valley: they are connected by a continuous path, on which empirical risk is invariant. Further for one-hidden-layer networks, we prove that all local minima in a cell constitute an equivalence class; they are concentrated in a valley; and they are all global minima in the cell.

## 1 INTRODUCTION

Neural networks have been successfully deployed in many real-world applications (LeCun et al., 2015; Witten et al., 2016; Silver et al., 2016; He et al., 2016; Litjens et al., 2017). In spite of this, the theoretical foundations of neural networks are somewhat premature. To the many deficiencies in our knowledge of deep learning theory, the investigation into the loss surfaces of neural networks is of fundamental importance. Understanding the loss surface would be helpful in several relevant research areas, such as the ability to estimate data distributions, the optimization of neural networks, and the generalization to unseen data.

This paper studies the role of the nonlinearities in activation functions in shaping the loss surfaces of neural networks. Our results demonstrate that the impact of nonlinearities is profound.

First, we prove that the loss surfaces of nonlinear neural networks are substantially different to those of linear neural networks, in which local minima are created equal, and also, they are all global minima (Kawaguchi, 2016; Baldi & Hornik, 1989; Lu & Kawaguchi, 2017; Freeman & Bruna, 2017; Zhou & Liang, 2018; Laurent & von Brecht, 2018; Yun et al., 2018). By contrast,

*Neural networks with arbitrary depth and arbitrary piecewise linear activations (excluding linear functions) have infinitely many spurious local minima under arbitrary continuously differentiable loss functions.*

---

[*]Both authors contributed equally.

[†]Bohan Wang is also affiliated with University of Science and Technology of China. This work was completed when he was a summer intern at UBTECH Sydney AI Centre, School of Computer Science, Faculty of Engineering, the University of Sydney.

This result only relies on four mild assumptions that cover most practical circumstances: (1) the training sample set is linearly inseparable; (2) all training sample points are distinct; (3) the output layer is narrower than the other hidden layers; and (4) there exists some turning point in the piece-wise linear activations that the sum of the slops on the two sides does not equal to 0.

Our result significantly extends the existing study on the existence of spurious local minimum. For example, Zhou & Liang (2018) prove that one-hidden-layer neural networks with two nodes in the hidden layer and two-piece linear (ReLU-like) activations have spurious local minima; Swirszcz et al. (2016) prove that ReLU networks have spurious local minima under the squared loss when most of the neurons are not activated; Safran & Shamir (2018) present a computer-assisted proof that two-layer ReLU networks have spurious local minima; a recent work (Yun et al., 2019b) have proven that neural networks with two-piece linear activations have infinite spurious local minima, but the results only apply to the networks with one hidden layer and one-dimensional outputs; and a concurrent work (Goldblum et al., 2020) proves that for multi-layer perceptrons of any depth, the performance of every local minimum on the training data equals to a linear model, which is also verified by experiments.

The proposed theorem is proved in three stages: (1) we prove that neural networks with one hidden layer and two-piece linear activations have spurious local minima; (2) we extend the conditions to neural networks with arbitrary hidden layers and two-piece linear activations; and (3) we further extend the conditions to neural networks with arbitrary depth and arbitrary piecewise linear activations. Since some parameters of the constructed spurious local minima are from continuous intervals, we have obtained infinitely many spurious local minima. At each stage, the proof follows a two-step strategy that: (a) constructs an infinite series of local minima; and (b) constructs a point in the parameter space whose empirical risk is lower than the constructed local minimum in Step (a). This strategy is inspired by Yun et al. (2019b) but we have made significant and non-trivial development.

Second, we draw a "big picture" for the loss surfaces of nonlinear neural networks. Soudry & Hoffer (2018) highlight a *smooth and multilinear partition* of the loss surfaces of neural networks. The nonlinearities in the piecewise linear activations partition the loss surface of any nonlinear neural network into multiple smooth and multilinear open cells. Specifically, every nonlinear point in the activation functions creates a group of the non-differentiable boundaries between the cells, while the linear parts of activations correspond to the smooth and multilinear interiors. Based on the partition, we discover a degenerate nature of the large amounts of local minima from the following aspects:

- **Every local minimum is globally minimal within a cell.** This property demonstrates that the local geometry within every cell is similar to the global geometry of linear networks, although technically, they are substantially different. It applies to any one-hidden-layer neural network with two-piece linear activations for regression under convex loss. We rigorously prove this property in two stages: (1) we prove that within every cell, the empirical risk $\hat{\mathcal{R}}$ is convex with respect to a variable $\hat{W}$ mapped from the weights $W$ by a mapping $Q$. Therefore, the local minima with respect to the variable $\hat{W}$ are also the global minima in the cell; and then (2) we prove that the local optimality is maintained under the constructed mapping. Specifically, the local minima of the empirical risk $\hat{\mathcal{R}}$ with respect to the parameter $W$ are also the local minima with respect to the variable $\hat{W}$. We thereby prove this property by combining the convexity and the correspondence of the minima. This proof is technically novel and non-trivial, though the intuitions are natural.

- **Equivalence classes and quotient space of local minimum valleys.** All local minima in a cell are concentrated as a *local minimum valley*: on a local minimum valley, all local minima are connected with each other by a continuous path, on which the empirical risk is invariant. Further, all these local minima constitute an equivalence class. This local minima valley may have several parallel valleys that are in the same equivalence class but do not appear because of the restraints from cell boundaries. If such constraints are ignored, all the equivalence classes constitute a quotient space. The constructed mapping $Q$ is exactly the quotient map. This result coincides with the property of *mode connectivity* that the minima found by gradient-based methods are connected by a path in the parameter space with almost invariant empirical risk (Garipov et al., 2018; Draxler et al., 2018; Kuditipudi et al., 2019). Additionally, this property suggests that we would need to study every local minimum valley as a whole.

- **Linear collapse.** Linear neural networks are covered by our theories as a simplified case. When all activations are linear, the partitioned loss surface collapses to one single cell, in which all local minima are globally optimal, as suggested by the existing works on linear networks (Kawaguchi, 2016; Baldi & Hornik, 1989; Lu & Kawaguchi, 2017; Freeman & Bruna, 2017; Zhou & Liang, 2018; Laurent & von Brecht, 2018; Yun et al., 2018).

**Notations.** If $M$ is a matrix, $M_{i,j}$ denotes the $(i, j)$-th component of $M$. If $M$ is a vector, $M_i$ denotes the $i$-th component of $M$. Define $E_{ij}$ as a matrix in which the $(i, j)$-th component is $1$ while all other components are $0$. Also, denote $e_i$ as a vector such that the $i$-th component is $1$ while all others are $0$. Additionally, we define $1_k \in \mathbb{R}^{k \times 1}$ is a vector whose components are all $1$, while those of $0_{n \times m} \in \mathbb{R}^{n \times m}$ (or briefly, $0$) are all $0$. For the brevity, $[i : j]$ denotes $\{i, \cdots, j\}$.

## 2 RELATED WORK

Some works suggest that linear neural networks have no spurious local minima. Kawaguchi (2016) proves that linear neural networks with squared loss do not have any spurious local minimum under three assumptions about the data matrix $X$ and the label matrix $Y$: (1) both matrices $XX^T$ and $XY^T$ have full ranks; and (2) the input layer is wider than the output layer; and (3) the eigenvalues of matrix $YX^\top \left(XX^T\right)^{-1} XY^T$ are distinct with each other. Zhou & Liang (2018) give an analytic formulation of the critical points for the loss function of deep linear networks, and thereby obtain a group of equivalence conditions for that critical point is a global minimum. Lu & Kawaguchi (2017) prove the argument under one assumption that both matrices $X$ and $Y$ have full ranks, which is even more restrictive. However, in practice, the activations of most neural networks are not linear. The nonlinearities would make the loss surface extremely non-convex and even non-smooth and therefore far different from the linear case.

The loss surfaces of over-parameterized neural networks have some special properties. Choromanska et al. (2015) empirically suggest that: (1) most local minima of over-parameterized networks are equivalent; and (2) small-size networks have spurious local minima but the probability of finding one decreases rapidly with the network size. Li et al. (2018) prove that over-parameterized fully-connected deep neural networks with continuous activation functions and convex, differentiable loss functions, have no bad strict local minimum. Nguyen et al. (2019) suggest that "sufficiently over-parameterized" neural networks have no bad local valley under the cross-entropy loss. Nguyen (2019) further suggests that the global minima of sufficiently over-parameterized neural networks are connected within a unique valley. Many other works study the convergence, generalization, and other properties of stochastic gradient descent on the loss surfaces of over-parameterized networks (Chizat & Bach; Arora et al., 2018; Brutzkus et al., 2018; Du et al., 2019; Soltanolkotabi et al., 2018; Allen-Zhu et al., 2019a;b; Oymak & Soltanolkotabi, 2019).

Many advances on the loss surfaces of neural networks are focused on other problems. Zhou & Feng (2018) and Mei et al. (2018) prove that the empirical risk surface and expected risk surface are linked. This correspondence highlights the value of investigating loss surfaces (empirical risk surfaces) to the study of generalization (the gap between empirical risks to expected risks). Hanin & Rolnick (2019) demonstrate that the input space of neural networks with piecewise linear activations are partitioned by multiple regions, while our work focuses on the partition of the loss surface. Xie et al. (2017) proves that the training error and test error are upper bounded by the magnitude of the gradient, under the assumption that the geometry discrepancy of the parameter $W$ is bounded. Sagun et al. (2016; 2018) present empirical results that the eigenvalues of the Hessian of the loss surface are two-fold: (1) a bulk centered closed to zero; and (2) outliers away from the bulk. Kawaguchi & Kaelbling (2020) prove that we can eliminate the spurious local minima by adding one unit per output unit for almost any neural network in practice. Tian (2017); Andrychowicz et al. (2016); Soltanolkotabi (2017); Zhong et al. (2017); Brutzkus & Globerson (2017); Tian (2017); Li & Yuan (2017); Zou et al. (2019); Li & Liang (2018); Du et al. (2018a; 2019); Zhang et al. (2019b); Zhou et al. (2019); Wang et al. (2019) study the optimization methods for neural networks. Other relevant works include Sagun et al. (2016; 2018); Nguyen & Hein (2018); Du et al. (2018b); Haeffele & Vidal (2017); Liang et al. (2018); Wu et al. (2018); Yun et al. (2019a); Zhang et al. (2019a); Kuditipudi et al. (2019); Garipov et al. (2018); Draxler et al. (2018); He et al. (2019); Kawaguchi & Kaelbling (2020).

## 3  NEURAL NETWORK HAS INFINITE SPURIOUS LOCAL MINIMA

This section investigates the existence of spurious local minima on the loss surfaces of neural networks. We find that almost all practical neural networks have infinitely many spurious local minima. This result stands for any neural network with arbitrary depth and arbitrary piecewise linear activations excluding linear functions under arbitrary continuously differentiable loss.

### 3.1  PRELIMINARIES

Consider a training sample set $\{(X_1, Y_1), (X_2, Y_2), \ldots, (X_n, Y_n)\}$ of size $n$. Suppose the dimensions of feature $X_i$ and label $Y_i$ are $d_X$ and $d_Y$, respectively. By aggregating the training sample set, we obtain the feature matrix $X \in \mathbb{R}^{d_X \times n}$ and label matrix $Y \in \mathbb{R}^{d_Y \times n}$.

Suppose a neural network has $L$ layers. Denote the weight matrix, bias, and activation in the $j$-th layer respectively by $W_j \in \mathbb{R}^{d_j \times d_{j-1}}$, $b_j \in \mathbb{R}^{d_j}$, and $h : \mathbb{R}^{d_j \times n} \to \mathbb{R}^{d_j \times n}$, where $d_j$ is the dimension of the output of the $j$-th layer. Also, for the input matrix $X$, the output of the $j$-th layer is denoted as the $Y^{(j)}$ and the output of the $j$-th layer before the activation is denoted as the $\tilde{Y}^{(j)}$,

$$\tilde{Y}^{(j)} = W_j Y^{(j-1)} + b_i \mathbf{1}_n^T, \tag{1}$$

$$Y^{(j)} = h\left(W_j Y^{(j-1)} + b_i \mathbf{1}_n^T\right). \tag{2}$$

The output of the network is defined as follows,

$$\hat{Y} = h_L\left(W_L h_{L-1}\left(W_{L-1} h_{L-2}\left(\ldots h_1\left(W_1 X + b_1 \mathbf{1}_n^T\right) \ldots\right) + b_{L-1} \mathbf{1}_n^T\right) + b_L \mathbf{1}_n^T\right). \tag{3}$$

Also, we define $Y^{(0)} = X$, $Y^{(L)} = \hat{Y}$, $d_0 = d_X$, and $d_L = d_Y$. In some situations, we use $\hat{Y}\left([W_i]_{i=1}^L, [b_i]_{i=1}^L\right)$ to clarify the parameters, as well as $\tilde{Y}^{(j)}$, $Y^{(j)}$, etc.

This section discusses neural networks with piecewise linear activations. A part of the proof uses two-piece linear activations $h_{s_-, s_+}$ which are defined as follows,

$$h_{s_-, s_+}(x) = \mathbf{I}_{\{x \leq 0\}} s_- x + \mathbf{I}_{\{x > 0\}} s_+ x, \tag{4}$$

where $|s_+| \neq |s_-|$ and $\mathbf{I}_{\{\cdot\}}$ is the indicator function.

**Remark.** *Piecewise linear functions are dense in the space of continuous functions. In other words, for any continuous function, we can always find a piecewise linear function to estimate it with arbitrary small distance.*

This section uses continuously differentiable loss to evaluate the performance of neural networks. Continuous differentiability is defined as follows.

**Definition 1** (Continuously differentiable). *We call a function $f : \mathbb{R}^n \to \mathbb{R}$ continuously differentiable with respect to the variable $x$ if: (1) the function $f$ is differentiable with respect to $x$; and (2) the gradient $\nabla_x f(x)$ of the function $f$ is continuous with respect to the variable $x$.*

### 3.2  MAIN RESULT

The theorem in this section relies on the following assumptions.

**Assumption 1.** *The training data cannot be fit by a linear model.*

**Assumption 2.** *All data points are distinct.*

**Assumption 3.** *All hidden layers are wider than the output layer.*

**Assumption 4.** *For the piece-wise linear activations, there exists some turning point that the sum of the slops on the two sides does not equal to $0$.*

To our best knowledge, our assumptions are the least restrictive compared with the relevant works in the literature. These assumptions are respectively justified as follows: (1) most real-world datasets are extremely complex and cannot be simply fit using linear models; (2) it is easy to guarantee that the data points are distinct by employing data cleansing methods; (3) for regression and many classification tasks, the width of output layer is limited and narrower than the hidden layers; and (4) this assumption is invalid only for activations like $f(x) = a|x|$.

Based on these four assumptions, we can prove the following theorem.

**Theorem 1.** *Neural networks with arbitrary depth and arbitrary piecewise linear activations (excluding linear functions) have infinitely many spurious local minima under arbitrary continuously differentiable loss whose derivative can equal $0$ only when the prediction and label are the same.*

In practice, most loss functions are continuously differentiable and the derivative can equal $0$ only when the prediction and label are the same, such as squared loss and cross-entropy loss (see Appendix A.1, Lemmas 2 and 3). Squared loss is a standard loss for regression and is defined as the $L_2$ norm of the difference between the ground-truth label and the prediction as follows.

$$l_2\left(Y_i, \hat{Y}_i\right) = \frac{1}{2}\left\|Y_i - \hat{Y}_i\right\|_F^2. \tag{5}$$

Meanwhile, cross-entropy loss is used as a standard loss in multiclass classification, which is defined as follows. Here, we treat the softmax function as a part of the loss function.

$$l_{ce}(Y_i, \hat{Y}_i) = -\sum_{j=1}^{d_Y} Y_{i,j} \log\left(\frac{\hat{Y}_{i,j}}{\sum_{k=1}^{d_Y} \hat{Y}_{i,k}}\right). \tag{6}$$

One can also remove Assumption 4, if Assumption 3 is replaced by the following assumption, which is mildly more restrictive (see a detailed proof in pp. 34–37).

**Assumption 5.** *The dimensions of the layers satisfy that:*

$$d_1 \geq d_Y + 2,$$
$$d_i \geq d_Y + 1, \ i = 2, \ldots, L - 1.$$

Our result demonstrates that introducing nonlinearities into activations substantially reshapes the loss surface: they bring infinitely many spurious local minima into the loss surface. This result highlights the substantial difference from linear neural networks that all local minima of linear neural networks are equally good, and therefore, they are all global minima (Kawaguchi, 2016; Baldi & Hornik, 1989; Lu & Kawaguchi, 2017; Freeman & Bruna, 2017; Zhou & Liang, 2018; Laurent & von Brecht, 2018; Yun et al., 2018).

Some works have noticed the existence of spurious local minima on the loss surfaces of nonlinear neural networks, which however has a limited applicable domain (Choromanska et al., 2015; Swirszcz et al., 2016; Safran & Shamir, 2018; Yun et al., 2019b). A notable work by Yun et al. (2019b) proves that one-hidden-layer neural networks with two-piece linear (ReLU-like) activations for one-dimensional regression have infinitely many spurious local minima under squared loss. This work first constructs a series of local minima and then prove they are spurious. This idea inspires some of this work. However, our work makes significant and non-trivial development that extends the conditions to arbitrary depth, piecewise linear activations excluding linear functions, and continuously differentiable loss.

### 3.3 PROOF SKELETON

This section presents the skeleton of the proof. Theorem 1 is proved in three stages. We first prove a simplified version of Theorem 1 and then extend the conditions in the last two stages. The proof is partially inspired by Yun et al. (2019b) but the proof in this paper has made nontrivial development and the results are significantly extended.

Yun et al. (2019b) and our paper both employ the following strategy: (a) construct a series of local minima based on a linear classifier; and (b) construct a new point with smaller empirical risk and thereby we prove that the constructed local minima are spurious. However, due to the differences in the loss function and the output dimensions, the exact constructions of local minima are substantially different.

Our extensions from Yun et al. (2019b) are three-fold: (1) From one hidden layer to arbitrary depth: To prove that networks with an arbitrary depth have infinite spurious local minima, we develop a novel strategy that employs transformation operations to force data flow through the same linear parts of the activations, in order to construct the spurious local minima; (2) From squared loss to arbitrary differentiable loss: Yun et al. (2019b) calculate the analytic formations of derivatives of

the loss to construct the local minima and then prove they are spurious. This technique cannot be transplanted to the case of arbitrary differentiable loss functions, because we cannot assume the analytic formation. To prove that the loss surface under an arbitrary differentiable loss has an infinite number of spurious local minima, we employ a new proof technique based on Taylor series and a new separation lemma; and (3) From one-dimensional output to arbitrary-dimensional output: To prove the loss surface of a neural network with an arbitrary-dimensional output has an infinite number of spurious local minima, we need to deal with the calculus of functions whose domain and codomain are a matrix space and a vector space, respectively. By contrast, when the output dimension is one, the codomain is only the space of real numbers. Therefore, the extension of the output dimension significantly mounts the difficulty of the whole proof.

**Stage (1): Neural networks with one hidden layer and two-piece linear activations.**

We first prove that nonlinear neural networks with one hidden layer and two-piece linear activation functions (ReLU-like activations) have spurious local minima. The proof in this stage further follows a two-step strategy:

(a) We first construct local minima of the empirical risk $\hat{\mathcal{R}}$ (see Appendix A.2, Lemma 4). These local minimizers are constructed based on a linear neural network which has the same network size (dimension of weight matrices) and evaluated under the same loss. The design of the hidden layer guarantees that the components of the output $\tilde{Y}^{(1)}$ in the hidden layer before the activation are all positive. The activation is thus effectively reduced to a linear function. Therefore, the local geometry around the local minima with respect to the weights $W$ is similar to those of linear neural networks. Further, the design of the output layer guarantees that its output $\hat{Y}$ is the same as the linear neural network. This construction helps to utilize the results of linear neural networks to solve the problems in nonlinear neural networks.

(b) We then prove that all the constructed local minima in Step (a) are spurious (see Appendix A.2, Theorem 4). Specifically, we assumed by Assumption 1 that the dataset cannot be fit by a linear model. Therefore, the gradient $\nabla_{\hat{Y}} \hat{\mathcal{R}}$ of the empirical risk $\hat{\mathcal{R}}$ with respect to the prediction $\hat{Y}$ is not zero. Suppose the $i$-th row of the gradient $\nabla_{\hat{Y}} \hat{\mathcal{R}}$ is not zero. Then, we use Taylor series and a preparation lemma (see Appendix A.5, Lemma 7) to construct another point in the parameter space that has smaller empirical risk. Therefore, we prove that the constructed local minima are spurious. Furthermore, the constructions involve some parameters that are randomly picked from a continuous interval. Thus, we constructed infinitely many spurious local minima.

**Stage (2) - Neural networks with arbitrary hidden layers and two-piece linear activations.**

We extend the condition in Stage (1) to any neural network with arbitrary depth and two-piece linear activations. The proof in this stage follows the same two-step strategy but has different implementations:

(a) We first construct a series of local minima of the empirical risk $\hat{\mathcal{R}}$ (see Appendix A.3, Lemma 5). The construction guarantees that every component of the output $\tilde{Y}^{(i)}$ in each layer before the activations is positive, which secure all the input examples flow through the same part of the activations. Thereby, the nonlinear activations are reduced to linear functions. Also, our construction guarantees that the output $\hat{Y}$ of the network is the same as a linear network with the same weight matrix dimensions.

(b) We then prove that the constructed local minima are spurious (see Appendix A.3, Theorem 5). The idea is to find a point in the parameter space that has the same empirical risk $\hat{\mathcal{R}}$ with the constructed point in Stage (1), Step (b).

**Stage (3) - Neural networks with arbitrary hidden layer and piecewise linear activations.**

We further extend the conditions in Stage (2) to any neural network with arbitrary depth and arbitrary piecewise linear activations. We continue to adapt the two-step strategy in this stage:

(a) We first construct a local minimizer of the empirical risk $\hat{\mathcal{R}}$ based on the results in Stages (1) and (2) (see Appendix A.4, Lemma 6). This construction is based on Stage (2), Step (a). The difference of the construction in this stage is that every linear part in activations can be a finite interval. The constructed weight matrices use several uniform scaling and translation operations to the outputs

of hidden layers in order to guarantee that all the input training sample points flow through the same linear parts of the activations. We thereby reduce the nonlinear activations to linear functions, effectively. Also, our construction guarantees that the output $\hat{Y}$ of the neural network equals to that of the corresponding linear neural network.

(b) We then prove that the constructed local minima are spurious (see Appendix A.4). We use the same strategy in Stage (2), Step (b). Some adaptations are implemented for the new conditions.

## 4 A BIG PICTURE OF THE LOSS SURFACE

This section draws a big picture for the loss surfaces of neural networks. Based on a recent result by Soudry & Hoffer (2018), we present four profound properties of the loss surface that collectively characterize how the nonlinearities in activations shape the loss surface.

### 4.1 PRELIMINARIES

The discussions in this section use the following concepts.

**Definition 2** (Open ball and open set)**.** *The open ball in $\mathcal{H}$ centered at $x \in \mathcal{H}$ and of radius $r > 0$ is defined by $B(h, r) = \{x : \|x - h\| < r\}$. A subset $A \subset \mathcal{H}$ of a space $\mathcal{H}$ is called a open set, if for every point $h \in A$, there exists a positive real $r > 0$, such that the open ball $B(h, r)$ with center $h$ and radius $r$ is in the subset $A$: $B(h, r) \subset A$.*

**Definition 3** (Interior point and interior)**.** *For a subset $A \subset \mathcal{H}$ of a space $\mathcal{H}$, a point $h \in A$ is called an interior point of $A$, if there exists a positive real $r > 0$, such that the open ball $B(h, r)$ with center $h$ and radius $r$ is in the subset $A$: $B(h, r) \subset A$. The set of all the interior points of the set $A$ is called the interior of the set $A$.*

**Definition 4** (Limit point, closure, and boundary)**.** *For a subset $A \subset \mathcal{H}$ of a space $\mathcal{H}$, a point $h \in A$ is called a limit point, if for every $r > 0$, the open ball $B(h, r)$ with center $h$ and radius $r$ contains some point of $A$: $B(h, r) \cap A \neq \emptyset$. The closure $\bar{A}$ of the set $A$ consists of the union of the set $A$ and all its limit points. The boundary $\partial A$ is defined as the set of points which are in the closure of set $A$ but not in the interior of set $A$.*

**Definition 5** (Multilinear)**.** *A function $f\colon \mathcal{X}_1 \times \mathcal{X}_2 \to \mathcal{Y}$ is called multilinear if for arbitrary $x_1^1, x_1^2 \in \mathcal{X}$, $x_2^1, x_2^2 \in \mathcal{X}_2$, and constants $\lambda_1$, $\lambda_2$, $\mu_1$, and $\mu_2$, we have*

$$f(\lambda_1 x_1^1 + \lambda_2 x_1^2, \mu_1 x_2^1 + \mu_2 x_2^2) = \lambda_1 \mu_1 f(x_1^1, x_2^1) + \lambda_1 \mu_2 f(x_1^1, x_2^2) + \lambda_2 \mu_1 f(x_1^2, x_2^1) + \lambda_2 \mu_2 f(x_1^2, x_2^2).$$

**Remark.** *The definition of "multilinear" implies that the domain of any multilinear function $f$ is a connective and convex set, such as the smooth and multilinear cells below.*

**Definition 6** (Equivalence class, and quotient space)**.** *Suppose $X$ is a linear space. $[x] = \{v \in X : v \sim x\}$ is an equivalence class, if there is an equivalent relation $\sim$ on $[x]$, such that for any $a, b, c \in [x]$, we have: (1) reflexivity: $a \sim a$; (2) symmetry: if $a \sim b$, $b \sim a$; and (3) transitivity: if $a \sim b$ and $b \sim c$, $a \sim c$. The quotient space and quotient map are defined to be $X/\sim = \{\{v \in X : v \sim x\} : x \in X\}$ and $x \to [x]$, respectively.*

### 4.2 MAIN RESULTS

In this section, the loss surface is defined under convex loss with respect to the prediction $\hat{Y}$ of the neural network. Convex loss covers many popular loss functions in practice, such as the squared loss for the regression tasks and many others based on norms. The triangle inequality of the norms secures the convexity of the corresponding loss functions. The convexity of the squared loss is checked in the appendix (see Appendix B, Lemma 8).

We now present four propositions to express the loss surfaces of nonlinear neural networks. These propositions give four major properties of the loss surface that collectively draw a big picture for the loss surface.

We first recall a lemma by Soudry & Hoffer (2018). It proves that the loss surfaces of neural networks have smooth and multilinear partitions.

**Lemma 1** (Smooth and multilinear partition; cf. Soudry & Hoffer (2018)). *The loss surfaces of neural networks of arbitrary depth with piecewise linear functions excluding linear functions are partitioned into multiple smooth and multilinear open cells, while the boundaries are nondifferentiable.*

Based on the smooth and multilinear partition, we prove four propositions as follows.

**Theorem 2** (Analogous convexity). *For one-hidden-layer neural networks with two-piece linear activation for regression under convex loss, within every cell, all local minima are equally good, and also, they are all global minima in the cell.*

**Theorem 3** (Equivalence classes of local minimum valleys). *Suppose all conditions of Theorem 2 hold. Assume the loss function is strictly convex. Then, all local minima in a cell are concentrated as a local minimum valley: they are connected with each other by a continuous path and have the same empirical risk. Additionally, all local minima in a cell constitute an equivalence class.*

**Corollary 1** (Quotient space of local minimum valleys). *Suppose all conditions of Theorem 3 hold. There might exist some "parallel" local minimum valleys in the equivalence class of a local minimum valley. They do not appear because of the constraints from the cell boundaries. If we ignore such constraints, all equivalence classes of local minima valleys constitute a quotient space.*

**Corollary 2** (Linear collapse). *The partitioned loss surface collapses to one single smooth and multilinear cell, when all activations are linear.*

### 4.3 DISCUSSIONS AND PROOF TECHNIQUES

The four propositions collectively characterize how the nonlinearities in activations shape the loss surfaces of neural networks. This section discusses the results and the structure of the proofs. A detailed proof is omitted here and given in Appendix B.

**Smooth and multilinear partition.** Intuitively, the nonlinearities in the piecewise linear activation functions partition the surface into multiple smooth and multilinear cells. Zhou & Liang (2018); Soudry & Hoffer (2018) highlight the partition of the loss surface. We restate it here to make the picture self-contained. A similar but also markedly different notions recently proposed by Hanin & Rolnick (2019) demonstrate that the input data space is partitioned into multiple linear regions, while our work focuses on the partition in the parameter space.

**Every local minimum is globally minimal within a cell.** In convex optimization, convexity guarantees that all the local minima are global minima. This theorem proves that the local minima within a cell are equally good, and also, they are all global minima in the cell. This result is not surprising provided the excellent training performance of deep learning algorithms. However, the proof is technically non-trivial.

Soudry & Hoffer (2018) proved that the local minima in a cell are the same. However, there would be some point near the boundary has a smaller empirical risk and is not locally minimal. Unfortunately, the proof by Soudry & Hoffer (2018) cannot exclude this possibility. By contrast, our proof completely solves this problem. Furthermore, our proof holds for any convex loss, including squared loss and cross-entropy loss, but Soudry & Hoffer (2018) only stands for squared loss.

It is challenging to prove, because the proof techniques for the case of linear networks cannot be transplanted here. Technically, linear networks can be expressed by the product of a sequence of weight matrices, which guarantees good geometrical properties. Specifically, the effect of every linear activation function is just equivalently multiplying a real constant to the output. However, the loss surface within a cell of a nonlinear neural network does not have this property. Below is the skeleton of our proof.

We first prove that the empirical risk $\hat{\mathcal{R}}$ is a convex function within every cell with respect to a variable $\hat{W}$ which is calculated from the weights $W$. Therefore, all local minima of the empirical risk $\hat{\mathcal{R}}$ with respect to $\hat{W}$ are also globally optimal in the cell. Every cell corresponds to a specific series of linear parts of the activations. Therefore, in any fixed cell, the activation $h_{s_-,s_+}$ can be expressed by the slopes of the corresponding linear parts as the following equations,

$$\hat{\mathcal{R}}(W_1, W_2) = \frac{1}{n}\sum_{i=1}^{n} l\left(y_i, W_2 h(W_1 x_i)\right) = \frac{1}{n}\sum_{i=1}^{n} l\left(y_i, W_2 \text{diag}\left(A_{\cdot,i}\right) W_1 x_i\right), \quad (7)$$

where $A_{\cdot,i}$ is the $i$-th column of matrix

$$A = \begin{bmatrix} h'_{s_-,s_+}((W_1)_{1,\cdot}x_1) & \cdots & h'_{s_-,s_+}((W_1)_{1,\cdot}x_n) \\ \vdots & \ddots & \vdots \\ h'_{s_-,s_+}((W_1)_{d_1,\cdot}x_1) & \cdots & h'_{s_-,s_+}((W_1)_{d_1,\cdot}x_n) \end{bmatrix}.$$

Matrix $A$ is constituted by collecting the slopes of the activation $h$ at every point $(W_1)_{i,\cdot}x_j$.

Different elements of the matrix $A$ can be multiplied either one of $\{s_-, s_+\}$. Therefore, we cannot use a single constant to express the effect of this activation, and thus, even within the cell, a nonlinear network cannot be expressed as the product of a sequence of weight matrices. This difference ensures that the proofs of deep linear neural networks cannot be transplanted here.

Then, we prove that (see p. 40)

$$W_2 \text{diag}\,(A_{\cdot,i})\,W_1 x_i = A_{\cdot,i}^T \text{diag}(W_2)W_1 x_i. \tag{8}$$

Applying eq. (8) to eq. (7), the empirical risk $\hat{\mathcal{R}}$ equals to a formulation similar to the linear neural networks,

$$\hat{\mathcal{R}} = \frac{1}{n}\sum_{i=1}^{n} l\left(y_i - A_{\cdot,i}^T \text{diag}(W_2)W_1 x_i\right). \tag{9}$$

Afterwards, define $\hat{W}_1 = \text{diag}(W_2)W_1$ and then straighten the matrix $\hat{W}_1$ to a vector $\hat{W}$,

$$\hat{W} = \left((\hat{W}_1)_{1,\cdot} \quad \cdots \quad (\hat{W}_1)_{d_1,\cdot}\right),$$

Define $Q : (W_1, W_2) \mapsto \hat{W}$, and also define,

$$\hat{X} = (A_{\cdot,1} \otimes x_1 \quad \cdots \quad A_{\cdot,n} \otimes x_n).$$

We can prove the following equations (see p. 41),

$$\left(A_{\cdot,1}^T \hat{W}_1 x_1 \quad \cdots \quad A_{\cdot,n}^T \hat{W}_1 x_n\right) = \hat{W}\hat{X}.$$

Applying eq. (9), the empirical risk is transferred to a convex function as follows,

$$\hat{\mathcal{R}} = \frac{1}{n}\sum_{i=1}^{n} l\left(y_i, (A_{\cdot,i})^T \hat{W}_1 x_i\right) = \frac{1}{n}\sum_{i=1}^{n} l(y_i, \hat{W}\hat{X}_i).$$

We then prove that the local optimality of the empirical risk $\hat{\mathcal{R}}$ is maintained when the weights $W$ are mapped to the variable $\hat{W}$. Specifically, the local minima of the empirical risk $\hat{\mathcal{R}}$ with respect to the weight $W$ are also the local minima with respect to the variable $\hat{W}$. The maintenance of optimality is not surprising but the proof is technically non-trivial (see a detailed proof in pp. 42-43).

**Equivalence classes and quotient space of local minimum valleys.** The constructed mapping $Q$ is a quotient map. Under the setting in the previous property, all local minima in a cell is an equivalence class; they are concentrated as a local minimum valley. However, there might exist some "parallel" local minimum valley in the equivalence class, which do not appear because of the constraints from the cell boundaries. Further for neural networks of arbitrary depth, we also constructed a local minimum valley (the spurious local minima constructed in Section 3). This result explains the property of *mode connectivity* that the minima found by gradient-based methods are connected by a path in the parameter space with almost constant empirical risk, which is proposed in two empirical works (Garipov et al., 2018; Draxler et al., 2018). A recent theoretical work (Kuditipudi et al., 2019) proves that dropout stability and noise stability guarantee the mode connectivity.

**Linear collapse.** Our theories also cover the case of linear neural networks. Linear neural networks do not have any nonlinearity in their activations. Correspondingly, the loss surface does not have any non-differentiable boundaries. In our theories, when there is no nonlinearity in the activations, the partitioned loss surface collapses to a single smooth, multilinear cell. All local minima wherein are equally good, and also, they are all global minima as follows. This result unites the existing results on linear neural networks (Kawaguchi, 2016; Baldi & Hornik, 1989; Lu & Kawaguchi, 2017; Freeman & Bruna, 2017; Zhou & Liang, 2018; Laurent & von Brecht, 2018; Yun et al., 2018).

## 5 CONCLUSION AND FUTURE DIRECTIONS

This paper reports that the nonlinearities in activations substantially shape the loss surfaces of neural networks. First, we prove that neural networks have infinitely many spurious local minima which are in contrast to the circumstance of linear neural networks. This result stands for any neural network with arbitrary hidden layers and arbitrary piecewise linear activations (excluding linear functions) under many popular loss functions in practice (e.g., squared loss and cross-entropy loss). This result significantly extends the conditions of the relevant results and has the least restrictive assumptions that cover most practical circumstances: (1) the training data is not linearly separable; (2) the training sample points are distinct; (3) all hidden layers are wider than the output layer; and (4) there exists some turning point in the piece-wise linear activation that the sum of the slops on the two sides does not equal to $0$. Second, based on a recent result that the loss surface has a smooth and multilinear partition, we draw a big picture of the loss surface from the following aspects: (1) local minima in any cell are equally good, and also, they are all global minima in the cell; (2) all local minima in one cell constitute an equivalence class and are concentrated as a local minimum valley; and (3) the loss surface collapses to one single cell when all activations are linear functions, which explains the results of linear neural networks. The first and second properties are rigorously proved for any one-hidden-layer nonlinear neural networks with two-piece linear (ReLU-like) activations for regression tasks under convex/strictly convex loss without any other assumption.

Theoretically understanding deep learning is of vital importance to both academia and industry. A major barrier recognized by the whole community is that deep neural networks' loss surfaces are extremely non-convex and even non-smooth. Such non-convexity and non-smoothness make the analysis of the optimization and generalization properties prohibitively difficult. A natural idea is to bypass the geometrical properties and then approach a theoretical explanation. We argue that such "intimidating" geometrical properties are exactly the major factors that shape the properties of deep neural networks, and also the key to explaining deep learning. We propose to explore the magic of deep learning from the geometrical structures of its loss surface. Future directions towards fully understanding deep learning are summarized as follows,

- **Investigate the (potential) equivalence classes and quotient space of local minimum valleys for deep neural networks.** This paper suggests a degenerate nature of the large amounts of local minima: all the local minima within one cell constitute an equivalence class. We construct a quotient map for one-hidden-layer neural networks with two-piece activations for regression. Whether deep neural networks have similar properties remains an open problem. Understanding the quotient space would be a major step of understanding the approximation, optimization, and generalization of deep learning.

- **Explore the sophisticated geometry of local minimum valleys.** The quotient space of local minima suggests a strategy that treats every local minimum valley as a whole. However, the sophisticated local geometrical properties around the local minimum valleys are still premature, such as the sharpness/flatness of the local minima, the potential categorization of the local minimum valley according to their performance, and the volumes of the local minima valleys from different categories.

- **Tackle the optimization and generalization problems of deep learning.** Empirical results have overwhelmingly suggested that deep learning has excellent optimization and generalization capabilities, which is, however, beyond the current theoretical understanding: (1) one can employ stochastic convex optimization methods (such as SGD) to minimize the extremely non-convex and non-smooth loss function in deep learning, which is expected to be NP-hard but practically solved by computationally cheap optimization methods; and (2) heavily-parametrized neural networks can generalize well in many tasks, which is beyond the expectation of most current theoretical frameworks based on hypothesis complexity and the variants. The sophisticated geometrical expression, if fortunately, we possess in the future, would be a compelling push to tackle the generalization and optimization muses of deep learning.

ACKNOWLEDGMENTS

This work was supported by Australian Research Council Project FL-170100117. The authors sincerely appreciate Micah Goldblum and the anonymous reviewers for their constructive comments.

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

# A   PROOF OF THEOREM 1

This appendix gives a detailed proof of Theorem 1 omitted from the main text. It follows the skeleton presented in Section 3.3.

## A.1   SQUARED LOSS AND CROSS-ENTROPY LOSS

We first check whether squared loss and cross-entropy loss are covered by the requirements of Theorem 1.

**Lemma 2.** *The squared loss (defined by eq. 5) is continuously differentiable with respect to the prediction of the model, whose gradient of loss equal to zero when the prediction and the label are different.*

*Proof.* Apparently, the squared loss is differentiable with respect to $\hat{Y}$. Specifically, the gradient with respect to $\hat{Y}$ is as follows,

$$\nabla_{\hat{Y}} \left\| Y - \hat{Y} \right\|^2 = 2 \left( Y - \hat{Y} \right),$$

which is continuous with respect to $\hat{Y}$.

Also, when the prediction $\hat{Y}$ does not equals to the label $Y$, we have

$$\nabla_{\hat{Y}} \left\| Y - \hat{Y} \right\|^2 \neq 0.$$

The proof is completed. $\qquad\square$

**Lemma 3.** *The cross-entropy loss eq. (6) is continuously differentiable with respect to the prediction of the model, whose gradient of loss equal to zero when the prediction and the label are different. Also, we assume that the ground-truth label is a one-hot vector.*

*Proof.* For any $i \in [1 : n]$, the cross-entropy loss is differentiable with respect to $\hat{Y}_i$. The $j$-th component of the gradient with respect to the prediction $\hat{Y}_i$ is as follows,

$$\frac{\partial \left( - \sum\limits_{k=1}^{d_Y} Y_{k,i} \log \left( \frac{e^{\hat{Y}_{k,i}}}{\sum_{k=1}^{d_Y} e^{\hat{Y}_{k,i}}} \right) \right)}{\partial \hat{Y}_{j,i}} = \left( \sum\limits_{k=1}^{d_Y} Y_{k,i} \right) \frac{e^{\hat{Y}_{j,i}}}{\sum\limits_{k=1}^{d_Y} e^{\hat{Y}_{k,i}}} - Y_{j,i}. \qquad (10)$$

which is continuous with respect to $\hat{Y}_i$. So, the cross-entropy loss is continuously differentiable with respect to $\hat{Y}_i$.

Additionally, if the gradient (eq. (10)) is zero, we have the following equations,

$$\left( \sum\limits_{k=1}^{d_Y} Y_{k,i} \right) e^{\hat{Y}_{j,i}} - Y_{j,i} \sum\limits_{k=1}^{d_Y} e^{\hat{Y}_{k,i}} = 0, \; j = 1, 2, \cdots, n.$$

Rewrite it into the matrix form, we have

$$\begin{bmatrix} \sum\limits_{k=1}^{d_Y} Y_{k,i} - Y_{1,i} & -Y_{1,i} & \cdots & -Y_{1,i} \\ -Y_{2,i} & \sum\limits_{k=1}^{d_Y} Y_{k,i} - Y_{2,i} & \cdots & -Y_{2,i} \\ \vdots & \vdots & \ddots & \vdots \\ -Y_{d_Y,i} & \cdots & -Y_{d_Y,i} & \sum\limits_{i=1}^{d_Y} Y_{k,i} - Y_{d_Y,i} \end{bmatrix} \begin{bmatrix} e^{\hat{Y}_{1,i}} \\ e^{\hat{Y}_{2,i}} \\ \vdots \\ e^{\hat{Y}_{d_Y,i}} \end{bmatrix} = 0.$$

Since $\sum_{k=1}^{d_Y} Y_{k,i} = 1$, we can easily check the rank of the left matrix is $d_Y - 1$. So the dimension of the solution space is one. Meanwhile, we have

$$
\begin{bmatrix}
\sum_{k=1}^{d_Y} Y_{k,i} - Y_{1,i} & -Y_{1,i} & \cdots & -Y_{1,i} \\
-Y_{2,i} & \sum_{k=1}^{d_Y} Y_{k,i} - Y_{2,i} & \cdots & -Y_{2,i} \\
\vdots & \vdots & \ddots & \vdots \\
-Y_{d_Y,i} & \cdots & -Y_{d_Y,i} & \sum_{i=1}^{d_Y} Y_{k,i} - Y_{d_Y,i}
\end{bmatrix}
\begin{bmatrix}
Y_{1,i} \\
Y_{2,i} \\
\vdots \\
Y_{d_Y,i}
\end{bmatrix} = 0.
$$

Therefore, $0 \neq e^{\hat{Y}_{k,i}} = \lambda Y_{k,i}$, for some $\lambda \in \mathbb{R}$, which contradicts to the assumption that some of the components of $Y$ is 0 ($Y_{i,\cdot}$ is a one-hot vector).

The proof is completed. $\qquad\square$

## A.2 STAGE (1)

In Stage (1), we prove that deep neural networks with one hidden layer, two-piece linear activation $h_{s_-,s_+}$, and multi-dimensional outputs have infinite spurious local minima.

This stage is organized as follows: (a) we construct a local minimizer by Lemma 4; and (b) we prove that the local minimizer is spurious in Theorem 4 by constructing a set of parameters with smaller empirical risk.

Without loss of generality, we assume that $s_+ \neq 0$. Otherwise, suppose that $s_+ = 0$. From the definition of ReLU-like activation (eq. (4)), we have $s_- \neq 0$. Since

$$
h_{s_-,s_+}(x) = h_{-s_+,-s_-}(-x),
$$

the output of the neural network with parameters $\left\{ [W_i]_{i=1}^L, [b_i]_{i=1}^L \right\}$ and activation $h_{s_-,s_+}$ equals to that of the neural network with parameters $\left\{ [W_i']_{i=1}^L, [b_i']_{i=1}^L \right\}$ and activation $h_{-s_+,-s_-}$ where $W_i' = -W_i, b_i' = -b_i, i = 1, 2, \cdots, L-1$ and $W_L' = W_L, b_L' = b_L$. Since $\left\{ [W_i]_{i=1}^L, [b_i]_{i=1}^L \right\} \to \left\{ [W_i']_{i=1}^L, [b_i']_{i=1}^L \right\}$ is an one-to-one map, it is equivalent to consider either the two networks, with $h_{-s_+,-s_-}(x)$ has non-zero slope when $x > 0$.

**Step (a). Construct local minima of the loss surface.**

**Lemma 4.** *Suppose that $\tilde{W}$ is a local minimizer of*

$$
f(W) \triangleq \frac{1}{n} \sum_{i=1}^n l \left( Y_i, W \begin{bmatrix} x_i \\ 1 \end{bmatrix} \right), \tag{11}
$$

*Under Assumption 3, any one-hidden-layer neural network has a local minimum at*

$$
\hat{W}_1 = \begin{bmatrix} \left[ \tilde{W} \right]_{\cdot,[1:d_X]} \\ \mathbf{0}_{(d_1-d_Y)\times d_X} \end{bmatrix}, \; \hat{b}_1 = \begin{bmatrix} \left[ \tilde{W} \right]_{\cdot,d_X+1} - \eta\mathbf{1}_{d_Y} \\ -\eta\mathbf{1}_{d_1-d_Y} \end{bmatrix}, \tag{12}
$$

*and*

$$
\hat{W}_2 = \begin{bmatrix} \frac{1}{s_+} I_{d_Y} & \mathbf{0}_{d_Y \times (d_1-d_Y)} \end{bmatrix}, \; \hat{b}_2 = \eta\mathbf{1}_{d_Y}, \tag{13}
$$

*where $\hat{W}_1$ and $\hat{b}_1$ are respectively the weight matrix and the bias of the first layer, $\hat{W}_2$ and $\hat{b}_2$ are respectively the weight matrix and the bias of the second layer, and $\eta$ is a negative constant with absolute value sufficiently large such that*

$$
\tilde{W}\tilde{X} - \eta\mathbf{1}_{d_Y}\mathbf{1}_n^T > \mathbf{0}, \tag{14}
$$

*where $>$ is element-wise.*

*Also, the loss in this lemma is continuously differentiable loss whose gradient does not equals to $0$ when the prediction is not the same as the ground-truth label.*

*Proof.* We show that the empirical risk is higher in the neiborhood of $\left\{ \left[ \hat{W}_i \right]_{i=1}^2, \left[ \hat{b}_i \right]_{i=1}^2 \right\}$, in order to prove that $\left\{ \left[ \hat{W}_i \right]_{i=1}^2, \left[ \hat{b}_i \right]_{i=1}^2 \right\}$ is a local minimizer.

The output of the first layer before the activation is

$$\tilde{Y}^{(1)} = \hat{W}_1 X + \hat{b}_1 \mathbf{1}_n^T = \begin{bmatrix} \hat{W} X - \eta \mathbf{1}_{d_Y} \mathbf{1}_n^T \\ -\eta \mathbf{1}_{d_1 - d_Y} \mathbf{1}_n^T \end{bmatrix}.$$

Because $\eta$ is a negative constant with absolute value sufficiently large such that eq. (27)) holds, the output above is positive (element-wise), the output of the neural network with parameters $\{\hat{W}_1, \hat{W}_2, \hat{b}_1, \hat{b}_2\}$ is

$$\hat{Y} = \hat{W}_2 h_{s_-, s_+} \left( \hat{W}_1 X + \hat{b}_1 \mathbf{1}_n^T \right) + \hat{b}_2 \mathbf{1}_n^T$$

$$= s_+ \hat{W}_2 \left( \hat{W}_1 X + \hat{b}_1 \mathbf{1}_n^T \right) + \hat{b}_2 \mathbf{1}_n^T$$

$$= s_+ \begin{bmatrix} \frac{1}{s_+} I_{d_Y} & \mathbf{0}_{d_Y \times (d_1 - d_Y)} \end{bmatrix} \begin{bmatrix} \hat{W} X - \eta \mathbf{1}_{d_Y} \mathbf{1}_n^T \\ -\eta \mathbf{1}_{d_1 - d_Y} \mathbf{1}_n^T \end{bmatrix} + \eta \mathbf{1}_{d_Y} \mathbf{1}_n^T$$

$$= \tilde{W} \tilde{X},$$

where $\tilde{X}$ is defined as

$$\tilde{X} = \begin{bmatrix} X \\ \mathbf{1}_n^T \end{bmatrix}. \tag{15}$$

Therefore, the empirical risk $\hat{\mathcal{R}}$ in terms of parameters $\{\hat{W}_1, \hat{W}_2, \hat{b}_1, \hat{b}_2\}$ is

$$\hat{\mathcal{R}} \left( \hat{W}_1, \hat{W}_2, \hat{b}_1, \hat{b}_2 \right) = \frac{1}{n} \sum_{i=1}^n l \left( Y_i, \left( \tilde{W} \tilde{X} \right)_{\cdot, i} \right) = \frac{1}{n} \sum_{i=1}^n l \left( Y_i, \tilde{W} \begin{bmatrix} x_i \\ 1 \end{bmatrix} \right) = f(\tilde{W}).$$

Then, we introduce a sufficiently small disturbance $\left\{ [\delta_{Wi}]_{i=1}^2, [\delta_{bi}]_{i=1}^2 \right\}$ into the parameters $\left\{ \left[ \hat{W}_i \right]_{i=1}^2, \left[ \hat{b}_i \right]_{i=1}^2 \right\}$. When the disturbance is sufficiently small, all components of the output of the first layer remain positive. Therefore, the output after the disturbance is

$$\hat{Y} \left( \left[ \hat{W}_i + \delta_{Wi} \right]_{i=1}^2, \left[ \hat{b}_i + \delta_{bi} \right]_{i=1}^2 \right)$$

$$= \left( \hat{W}_2 + \delta_{W2} \right) h_{s_-, s_+} \left( \left( \hat{W}_1 + \delta_{W1} \right) X + \left( \hat{b}_1 + \delta_{b1} \right) \mathbf{1}_n^T \right) + \left( \hat{b}_2 + \delta_{b2} \right) \mathbf{1}_n^T$$

$$\overset{(*)}{=} \left( \hat{W}_2 + \delta_{W2} \right) s_+ \left( \left( \hat{W}_1 + \delta_{W1} \right) X + \left( \hat{b}_1 + \delta_{b1} \right) \mathbf{1}_n^T \right) + \left( \hat{b}_2 + \delta_{b2} \right) \mathbf{1}_n^T$$

$$= s_+ \delta_{W2} \left( \left( \hat{W}_1 + \delta_{W1} \right) X + \left( \hat{b}_1 + \delta_{b1} \right) \mathbf{1}_n^T \right) + s_+ \hat{W}_2 \delta_{W1} X + s_+ \hat{W}_2 \delta_{b1} \mathbf{1}_n^T + \delta_{b2} \mathbf{1}_n^T$$

$$\quad + \hat{W}_2 s_+ (\hat{W}_1 X + \hat{b}_1 \mathbf{1}_n^T) + \hat{b}_2 \mathbf{1}_n^T$$

$$= \left( s_+ \delta_{W2} \left( \hat{W}_1 + \delta_{W1} \right) + s_+ \hat{W}_2 \delta_{W1} \right) X + \left( s_+ \hat{W}_2 \delta_{b1} + \delta_{b2} + s_+ \delta_{W2} \left( \hat{b}_1 + \delta_{b1} \right) \right) \mathbf{1}_n^T$$

$$\quad + \hat{W}_2 h_{s_-, s_+} (\hat{W}_1 X + \hat{b}_1 \mathbf{1}_n^T) + \hat{b}_2 \mathbf{1}_n^T$$

$$= (\tilde{W} + \delta) \begin{bmatrix} X \\ \mathbf{1}_n^T \end{bmatrix},$$

where eq. $(*)$ is because all components of $\left( \hat{W}_1 + \delta_{W1} \right) X + (b'_1 + \delta_{b1}) \mathbf{1}_n^T$ are positive, and $\delta$ is defined as the following matrix

$$\delta = \begin{bmatrix} s_+ \left( \hat{W}_2 \delta_{W1} + \delta_{W2} \hat{W}_1 + \delta_{W2} \delta_{W1} \right) & s_+ \hat{W}_2 \delta_{b1} + \delta_{b2} + s_+ \delta_{W2} \left( \hat{b}_1 + \delta_{b1} \right) \end{bmatrix}.$$

Therefore, the empirical risk $\hat{\mathcal{R}}$ with respect to $\left\{\left[\hat{W}_i + \delta_{Wi}\right]_{i=1}^2, \left[\hat{b}_i + \delta_{bi}\right]_{i=1}^2\right\}$ is

$$\hat{\mathcal{R}}\left(\left[\hat{W}_i + \delta_{Wi}\right]_{i=1}^2, \left[\hat{b}_i + \delta_{bi}\right]_{i=1}^2\right) = \frac{1}{n}\sum_{i=1}^n l\left(Y_i, \left(\left(\tilde{W} + \delta\right)\tilde{X}\right)_{\cdot,i}\right)$$

$$= \frac{1}{n}\sum_{i=1}^n l\left(Y_i, \left(\tilde{W} + \delta\right)\begin{bmatrix}x_i \\ 1\end{bmatrix}\right)$$

$$= f(\tilde{W} + \delta).$$

$\delta$ approaches zero when the disturbances $\{\delta_{W1}, \delta_{W2}, \delta_{b1}, \delta_{b2}\}$ approach zero (element-wise). Since $\hat{W}$ is the local minimizer of $f(W)$, we have

$$\hat{\mathcal{R}}\left(\left[\hat{W}_i\right]_{i=1}^2, \left[\hat{b}_i\right]_{i=1}^2\right) = f(\hat{W}) \le f(\hat{W} + \delta) = \hat{\mathcal{R}}\left(\left[\hat{W}_i + \delta_{Wi}\right]_{i=1}^2, \left[\hat{b}_i + \delta_{bi}\right]_{i=1}^2\right). \quad (16)$$

Because the disturbances $\{\delta_{W1}, \delta_{W2}, \delta_{b1}, \delta_{b2}\}$ are arbitrary, eq. (16) demonstrates that $\left\{\left[\hat{W}_i\right]_{i=1}^2, \left[\hat{b}_i\right]_{i=1}^2\right\}$ is a local minimizer.

The proof is completed. $\qquad\square$

### Step (b). Prove the constructed local minima are spurious.

**Theorem 4.** *Under the same conditions of Lemma 4 and Assumptions 1, 2, and 4, the constructed spurious local minima in Lemma 4 are spurious.*

*Proof.* The minimizer $\tilde{W}$ is the solution of the following equation

$$\nabla_W f(W) = 0.$$

Specifically, we have

$$\frac{\partial f\left(\tilde{W}\right)}{\partial W_{i,j}} = 0, \ i \in \{1, \cdots, d_Y\}, \ j \in \{1, \cdots, d_X\},$$

Applying the definition of $f(W)$ (eq. (11)),

$$\frac{\partial f\left(\tilde{W}\right)}{\partial W_{k,j}} = \sum_{i=1}^n \nabla_{\hat{Y}_i} l\left(Y_i, \tilde{W}\begin{bmatrix}x_i \\ 1\end{bmatrix}\right) E_{k,j}\begin{bmatrix}x_i \\ 1\end{bmatrix} = \sum_{i=1}^n \left(\nabla_{\hat{Y}_i} l\left(Y_i, \tilde{W}\begin{bmatrix}x_i \\ 1\end{bmatrix}\right)\right)_k \left(\begin{bmatrix}x_i \\ 1\end{bmatrix}\right)_j,$$

where $\hat{Y}_i = \tilde{W}\begin{bmatrix}x_i \\ 1\end{bmatrix}$, $\nabla_{\hat{Y}_i} l\left(Y_i, \tilde{W}\begin{bmatrix}x_i \\ 1\end{bmatrix}\right) \in \mathbb{R}^{1 \times d_Y}$. Since $k, j$ are arbitrary in $\{1, \cdots, d_Y\}$ and $\{1, \cdots, d_X\}$, respectively, we have

$$\mathbf{V}\begin{bmatrix}X^T & \mathbf{1}_n\end{bmatrix} = \mathbf{0}, \quad (17)$$

where

$$\mathbf{V} = \left[\left(\nabla_{\hat{Y}_1} l\left(Y_1, \tilde{W}\begin{bmatrix}x_1 \\ 1\end{bmatrix}\right)\right)^T \quad \cdots \quad \left(\nabla_{\hat{Y}_n} l\left(Y_n, \tilde{W}\begin{bmatrix}x_n \\ 1\end{bmatrix}\right)\right)^T\right].$$

We then define $\tilde{Y} = \tilde{W}\tilde{X}$. Applying Assumption 1, we have

$$\tilde{Y} - Y = \left(\tilde{W}\tilde{X} - Y\right) \ne \mathbf{0},$$

Thus, there exists some $k$-th row of $\tilde{Y} - Y$ that does not equal to $0$.

We can rearrange the rows of $\tilde{W}$ and $Y$ simultaneously, while $\tilde{W}$ is maintained as the local minimizer of $f(W)$ and $f(\tilde{W})$ invariant[1]. Without loss of generality, we assume $k = 1$ ($k$ is the index of the row). Set $\boldsymbol{u} = \boldsymbol{V}_{1,\cdot}$ and $v_i = \tilde{Y}_{1,i}$ in Lemma 7. There exists a non-empty separation $I = [1 : l']$ and $J = [l' + 1 : n]$ of $S = \{1, 2, \cdots, n\}$ and a vector $\beta \in \mathbb{R}^{d_x}$, such that

(1.1) for any positive constant $\alpha$ small enough, and $i \in I$, $j \in J$, $\tilde{Y}_{1,i} - \alpha\beta^T x_i < \tilde{Y}_{1,j} - \alpha\beta^T x_j$;

(1.2) $\sum_{i \in I} \boldsymbol{V}_{1,i} \neq 0$.

Define

$$\eta_1 = \tilde{Y}_{1,l'} - \alpha\beta^T x_{l'} + \frac{1}{2}\left(\min_{i \in \{l'+1,\ldots,n\}}\left(\tilde{Y}_{1,i} - \alpha\beta^T x_i\right) - \left(\tilde{Y}_{1,l'} - \alpha\beta^T x_{l'}\right)\right).$$

Applying (1.1), for any $i \in I$

$$\begin{aligned}
&\tilde{Y}_{1,i} - \alpha\beta^T x_i - \eta_1 \\
&= \left(\tilde{Y}_{1,i} - \alpha\beta^T x_i - \tilde{Y}_{1,l'} + \alpha\beta^T x_{l'}\right) - \frac{1}{2}\left(\min_{i \in \{l'+1,\ldots,n\}}\left(\tilde{Y}_{1,i} - \alpha\beta^T x_i\right) - \left(\tilde{Y}_{1,l'} - \alpha\beta^T x_{l'}\right)\right) \\
&< 0,
\end{aligned}$$

while for any $j \in J$,

$$\begin{aligned}
&\tilde{Y}_{1,j} - \alpha\beta^T x_j - \eta_1 > 0 \\
&= \left(\tilde{Y}_{1,j} - \alpha\beta^T x_i - \tilde{Y}_{1,l'} + \alpha\beta^T x_{l'}\right) - \frac{1}{2}\left(\min_{i \in \{l'+1,\ldots,n\}}\left(\tilde{Y}_{1,i} - \alpha\beta^T x_i\right) - \left(\tilde{Y}_{1,l'} - \alpha\beta^T x_{l'}\right)\right) \\
&\geq \frac{1}{2}\left(\min_{i \in \{l'+1,\ldots,n\}}\left(\tilde{Y}_{1,i} - \alpha\beta^T x_i\right) - \left(\tilde{Y}_{1,l'} - \alpha\beta^T x_{l'}\right)\right) \\
&> 0.
\end{aligned}$$

Define $\gamma \in \mathbb{R}$ which satisfies

$$|\gamma| = \begin{cases} \dfrac{1}{2}\min\limits_{i \in \{l'+1,\ldots,s_{t+1}\}} \alpha\beta^T(x_l - x_i), & l' < s_{t+1} \\ \alpha, & l' = s_{t+1} \end{cases},$$

where $s_{t+1}$ is defined in Lemma 7.

We argue that

$$\left|\frac{1}{2}\left(\min_{i \in \{l'+1,\ldots,n\}}\left(\tilde{Y}_{1,i} - \alpha\beta^T x_i\right) - \left(\tilde{Y}_{1,l'} - \alpha\beta^T x_{l'}\right)\right)\right| - |\gamma| > 0. \tag{18}$$

When $l' = s_{t+1}$, eq. (58) stands. Also,

$$\lim_{\alpha \to 0^+} \gamma = 0,$$

$$\lim_{\alpha \to 0^+}\left(\min_{i \in \{l'+1,\ldots,n\}}\left(\tilde{Y}_{1,i} - \alpha\beta^T x_i\right) - \left(\tilde{Y}_{1,l'} - \alpha\beta^T x_{l'}\right)\right) = \min_{i \in \{l'+1,\ldots,n\}}\tilde{Y}_{1,i} - \tilde{Y}_{1,l'} > 0.$$

Therefore, we get eq. (18) when $\alpha$ is small enough.

When $l' < s_{t+1}$, eq. (57) stands. Therefore,

$$|\gamma| = \frac{1}{2}\left|\frac{1}{2}\left(\min_{i \in \{l'+1,\ldots,n\}}\left(\tilde{Y}_{1,i} - \alpha\beta^T x_i\right) - \left(\tilde{Y}_{1,l'} - \alpha\beta^T x_{l'}\right)\right)\right|,$$

which apparently leads to eq. (18).

---

[1] $f$ is also the function in term of $Y$.

Therefore, for any $i \in I$, we have that

$$
\begin{aligned}
&\tilde{Y}_{1,i} - \alpha\beta^T x_i - \eta_1 + |\gamma| \\
&\leq -\frac{1}{2}\left(\min_{i\in\{l'+1,\ldots,n\}}\left(\tilde{Y}_{1,i} - \alpha\beta^T x_i\right) - \left(\tilde{Y}_{1,l'} - \alpha\beta^T x_{l'}\right)\right) + |\gamma| \\
&< 0,
\end{aligned}
$$

while for any $j \in J$,

$$
\begin{aligned}
&\tilde{Y}_{1,j} - \alpha\beta^T x_j - \eta_1 - |\gamma| \\
&\geq \frac{1}{2}\left(\min_{i\in\{l'+1,\ldots,n\}}\left(\tilde{Y}_{1,i} - \alpha\beta^T x_i\right) - \left(\tilde{Y}_{1,l'} - \alpha\beta^T x_{l'}\right)\right) - |\gamma| \\
&> 0.
\end{aligned}
$$

Furthermore, define $\eta_i$ $(2 \leq i \leq d_Y)$ as negative reals with absolute value sufficiently large, such that for any $i \in [2 : d_Y]$ and any $j \in [1 : n]$,

$$
\tilde{Y}_{i,j} - \eta_i > 0.
$$

Now we construct a point in the parameter space whose empirical risk is smaller than the proposed local minimum in Lemma 4 as follows

$$
\tilde{W}_1 = \begin{bmatrix} \tilde{W}_{1,[1:d_X]} - \alpha\beta^T \\ -\tilde{W}_{1,[1:d_X]} + \alpha\beta^T \\ W_{2,[1:d_X]} \\ \vdots \\ \tilde{W}_{d_Y,[1:d_X]} \\ 0_{(d_1-d_Y-1)\times d_X} \end{bmatrix}, \tag{19}
$$

$$
\tilde{b}_1 = \begin{bmatrix} \tilde{W}_{1,[d_X+1]} - \eta_1 + \gamma \\ -\tilde{W}_{1,[d_X+1]} + \eta_1 + \gamma \\ \tilde{W}_{2,[d_X+1]} - \eta_2 \\ \vdots \\ \tilde{W}_{d_Y,[d_X+1]} - \eta_{d_Y} \\ 0_{(d_1-d_Y-1)\times 1} \end{bmatrix}, \tag{20}
$$

$$
\tilde{W}_2 = \begin{bmatrix} \frac{1}{s_++s_-} & -\frac{1}{s_++s_-} & 0 & 0 & \cdots & 0 & 0 & \cdots & 0 \\ 0 & 0 & \frac{1}{s_+} & \cdots & 0 & \vdots & \vdots & & \vdots \\ \vdots & \vdots & \vdots & \frac{1}{s_+} & \cdots & 0 & & \ddots & \\ \vdots & \vdots & \vdots & \vdots & \ddots & \vdots & \vdots & & \vdots \\ 0 & 0 & 0 & 0 & \cdots & \frac{1}{s_+} & 0 & \cdots & 0 \end{bmatrix}, \tag{21}
$$

and

$$
\tilde{b}_2 = \begin{bmatrix} \eta_1 \\ \eta_2 \\ \vdots \\ \eta_{d_Y} \end{bmatrix}, \tag{22}
$$

where $\tilde{W}_i$ and $\tilde{b}_i$ are the weight matrix and the bias of the $i$-th layer, respectively.

After some calculations, the network output of the first layer before the activation in terms of $\left\{ \left[ \tilde{W}_i \right]_{i=1}^2, \left[ \tilde{b}_i \right]_{i=1}^2 \right\}$ is

$$\tilde{Y}^{(1)} = \tilde{W}_1 X + \tilde{b}_1 \mathbf{1}_n^T = \begin{bmatrix} \tilde{W}_{1,\cdot} \tilde{X} - \alpha \beta^T X - \eta_1 \mathbf{1}_n^T + \gamma \mathbf{1}_n^T \\ -\tilde{W}_{1,\cdot} \tilde{X} + \alpha \beta^T X + \eta_1 \mathbf{1}_n^T + \gamma \mathbf{1}_n^T \\ \tilde{W}_{2,\cdot} \tilde{X} - \eta_2 \mathbf{1}_n^T \\ \vdots \\ \tilde{W}_{d_Y,\cdot} \tilde{X} - \eta_{d_Y} \mathbf{1}_n^T \\ \mathbf{0}_{(d_1 - d_Y - 1) \times n} \end{bmatrix}.$$

Therefore, the output of the whole neural network is

$$\hat{Y} = \tilde{W}_2 h_{s_-, s_+} \left( \tilde{W}_1 X + \tilde{b}_1 \mathbf{1}_n^T \right) + \tilde{b}_2 \mathbf{1}_n^T$$

$$= \tilde{W}_2 h_{s_-, s_+} \left( \begin{bmatrix} \tilde{W}_{1,\cdot} \tilde{X} - \alpha \beta^T X - \eta_1 \mathbf{1}_n^T + \gamma \mathbf{1}_n^T \\ -\tilde{W}_{1,\cdot} \tilde{X} + \alpha \beta^T X + \eta_1 \mathbf{1}_n^T + \gamma \mathbf{1}_n^T \\ \tilde{W}_{2,\cdot} \tilde{X} - \eta_2 \mathbf{1}_n^T \\ \vdots \\ \tilde{W}_{d_Y,\cdot} \tilde{X} - \eta_{d_Y} \mathbf{1}_n^T \\ \mathbf{0}_{(d_1 - d_Y - 1) \times n} \end{bmatrix} \right) + \tilde{b}_2 \mathbf{1}_n^T.$$

Specifically, if $j \leq l'$,

$$\left( \tilde{Y}^{(1)} \left( \left[ \tilde{W}_i \right]_{i=1}^2, \left[ \tilde{b}_i \right]_{i=1}^2 \right) \right)_{1,j} = \tilde{W}_{1,\cdot} \begin{bmatrix} x_j \\ 1 \end{bmatrix} - \alpha \beta^T x_j - \eta_1 + \gamma$$

$$= \tilde{Y}_{1,j} - \alpha \beta^T x_j - \eta_1 + \gamma < 0,$$

$$\left( \tilde{Y}^{(1)} \left( \left[ \tilde{W}_i \right]_{i=1}^2, \left[ \tilde{b}_i \right]_{i=1}^2 \right) \right)_{2,j} = - \tilde{W}_{1,\cdot} \begin{bmatrix} x_j \\ 1 \end{bmatrix} + \alpha \beta^T x_j + \eta_1 + \gamma$$

$$= - \tilde{Y}_{1,j} + \alpha \beta^T x_j + \eta_1 + \gamma > 0.$$

Therefore, $(1, j)$-th component of $\hat{Y} \left( \left[ \tilde{W}_i \right]_{i=1}^2, \left[ \tilde{b}_i \right]_{i=1}^2 \right)$ is

$$\left( \hat{Y} \left( \left[ \tilde{W}_i \right]_{i=1}^2, \left[ \tilde{b}_i \right]_{i=1}^2 \right) \right)_{1,j}$$

$$= \left( \frac{1}{s_+ + s_-}, -\frac{1}{s_+ + s_-}, 0, \cdots, 0 \right) h_{s_-, s_+} \left( \begin{bmatrix} \tilde{W}_{1,\cdot} X - \alpha \beta^T X - \eta_1 \mathbf{1}_n^T + \gamma \mathbf{1}_n^T \\ -\tilde{W}_{1,\cdot} X + \alpha \beta^T X + \eta_1 \mathbf{1}_n^T + \gamma \mathbf{1}_n^T \\ \tilde{W}_{2,\cdot} X - \eta_2 \mathbf{1}_n^T \\ \vdots \\ \tilde{W}_{d_Y,\cdot} X - \eta_{d_Y} \mathbf{1}_n^T \\ \mathbf{0}_{d_1 - d_Y - 1} \mathbf{1}_n^T \end{bmatrix} \right)_{\cdot, j}$$

$$+ \eta_1$$

$$= \frac{1}{s_+ + s_-} h_{s_-, s_+} (\tilde{Y}_{1,j} - \alpha \beta^T x_j - \eta_1 + \gamma) - \frac{1}{s_+ + s_-} h_{s_-, s_+} (-\tilde{Y}_{1,j} + \alpha \beta^T x_j + \eta_1 + \gamma)$$

$$+ \eta_1$$

$$= \frac{s_-}{s_+ + s_-} (\tilde{Y}_{1,j} - \alpha \beta^T x_j - \eta_1 + \gamma) - \frac{s_+}{s_+ + s_-} (-\tilde{Y}_{1,j} + \alpha \beta^T x_j + \eta_1 + \gamma)$$

$$+ \eta_1$$

$$= \tilde{Y}_{1,j} - \alpha \beta^T x_j + \frac{s_- - s_+}{s_+ + s_-} \gamma; \tag{23}$$

Similarly, when $j > l'$, the $(1, j)$-th component is

$$
\begin{aligned}
&\left( \hat{Y} \left( \left[ \tilde{W}_i \right]_{i=1}^{2}, \left[ \tilde{b}_i \right]_{i=1}^{2} \right) \right)_{1,j} \\
&= \frac{s_+}{s_+ + s_-} (\tilde{Y}_{1,j} - \alpha \beta^T x_j - \eta_1 + \gamma) - \frac{s_-}{s_+ + s_-} (-\tilde{Y}_{1,j} + \alpha \beta^T x_j + \eta_1 + \gamma) + \eta_1 \\
&= \tilde{Y}_{1,j} - \alpha \beta^T x_j + \frac{s_+ - s_-}{s_+ + s_-} \gamma,
\end{aligned}
\tag{24}
$$

and

$$
\left( \hat{Y} \left( \left[ \tilde{W}_i \right]_{i=1}^{2}, \left[ \tilde{b}_i \right]_{i=1}^{2} \right) \right)_{i,j} = \frac{s_+}{s_+} (\tilde{Y}_{i,j} - \eta_i) + \eta_i = \tilde{Y}_{i,j}, \ i \geq 2.
\tag{25}
$$

Thus, the empirical risk of the neural network with parameters $\left\{ \left[ \tilde{W}_i \right]_{i=1}^{2}, \left[ \tilde{b}_i \right]_{i=1}^{2} \right\}$ is

$$
\begin{aligned}
&\hat{\mathcal{R}} \left( \left[ \tilde{W}_i \right]_{i=1}^{2}, \left[ \tilde{b}_i \right]_{i=1}^{2} \right) \\
&= \frac{1}{n} \sum_{i=1}^{n} l \left( Y_i, \tilde{W}_2 \left( \tilde{W}_1 x_i + \tilde{b}_1 \mathbf{1}_n^T \right) + \tilde{b}_2 \mathbf{1}_n^T \right) \\
&= \frac{1}{n} \sum_{i=1}^{n} \left( l \left( Y_i, \tilde{W} \begin{bmatrix} x_i \\ 1 \end{bmatrix} \right) + \nabla_{\hat{Y}_i} l \left( Y_i, \tilde{W} \begin{bmatrix} x_i \\ 1 \end{bmatrix} \right) \left( \tilde{W}_2 \left( \tilde{W}_1 x_i + \tilde{b}_1 \mathbf{1}_n^T \right) + \tilde{b}_2 \mathbf{1}_n^T - \tilde{W} \begin{bmatrix} x_i \\ 1 \end{bmatrix} \right) \right) \\
&\quad + \sum_{i=1}^{n} o \left( \left\| \tilde{W}_2 \left( \tilde{W}_1 x_i + \tilde{b}_1 \mathbf{1}_n^T \right) + \tilde{b}_2 \mathbf{1}_n^T - \tilde{W} \begin{bmatrix} x_i \\ 1 \end{bmatrix} \right\| \right).
\end{aligned}
\tag{26}
$$

Applying eqs. (23), (24), and (25), we have

$$
\begin{aligned}
&\sum_{i=1}^{n} \left( \tilde{W}_2 \left( \tilde{W}_1 x_i + \tilde{b}_1 \right) + \tilde{b}_2 - \tilde{W} \begin{bmatrix} x_i \\ 1 \end{bmatrix} \right)^T \nabla_{\hat{Y}_i} l \left( Y_i, \tilde{W} \begin{bmatrix} x_i \\ 1 \end{bmatrix} \right)^T \\
&\overset{(*)}{=} \sum_{i=1}^{l'} \mathbf{V}_{1,i} (-\alpha \beta^T x_i + \frac{s_+ - s_-}{s_+ + s_-} \gamma) + \sum_{i=l'+1}^{n} \mathbf{V}_{1,i} (-\alpha \beta^T x_i - \frac{s_+ - s_-}{s_+ + s_-} \gamma) \\
&= 2\gamma \sum_{i=1}^{l'} \frac{s_+ - s_-}{s_+ + s_-} \mathbf{V}_{1,i},
\end{aligned}
$$

where eq. $(*)$ is because

$$
\left( \tilde{W}_2 \left( \tilde{W}_1 x_i + \tilde{b}_1 \right) + \tilde{b}_2 - \tilde{W} \begin{bmatrix} x_i \\ 1 \end{bmatrix} \right)_j = \begin{cases} -\alpha \beta^T x_j + \frac{s_- - s_+}{s_+ + s_-} \gamma, & j = 1, i \leq l' \\ -\alpha \beta^T x_j - \frac{s_- - s_+}{s_+ + s_-} \gamma, & j = 1, i > l' \\ 0, & j \geq 2 \end{cases}.
$$

Furthermore, note that $\alpha = O(\gamma)$ (from the definition of $\gamma$). We have

$$
\begin{aligned}
&\sum_{i=1}^{n} o \left( \left\| \tilde{W}_2 \left( \tilde{W}_1 x_i + \tilde{b}_1 \right) + \tilde{b}_2 - \hat{W} \begin{bmatrix} x_i \\ 1 \end{bmatrix} \right\| \right) \\
&= \sum_{i=1}^{n} o \left( \sqrt{\sum_{j=1}^{n} \left( \tilde{W}_2 \left( \tilde{W}_1 x_i + \tilde{b}_1 \right) + \tilde{b}_2 - \tilde{W} \begin{bmatrix} x_i \\ 1 \end{bmatrix} \right)_j^2} \right) \\
&= o(\gamma).
\end{aligned}
$$

Let $\alpha$ be sufficiently small while $\mathrm{sgn}(\gamma) = -\mathrm{sgn}\left(\sum_{i=1}^{l'} \frac{s_+ - s_-}{s_+ + s_-} V_{1,i}\right)$. We have

$$\sum_{i=1}^{n} l\left(Y_i, \tilde{W}_2\left(\tilde{W}_1 x_i + \tilde{b}_1\right) + \tilde{b}_2\right) - \sum_{i=1}^{n} l\left(Y_i, \hat{W}\begin{bmatrix} x_i \\ 1 \end{bmatrix}\right)$$

$$= 2\gamma \sum_{i=1}^{l'} \frac{s_+ - s_-}{s_+ + s_-} V_{1,i} + o(\gamma)$$

$$\overset{(**)}{<} 0,$$

where inequality $(**)$ comes from (1.2) (see p. 18).

From Lemma 4, there exists a local minimizer $\left\{\left[\hat{W}_i\right]_{i=1}^{2}, \left[\hat{b}_i\right]_{i=1}^{2}\right\}$ with empirical risk that equals to $f(\tilde{W})$. Meanwhile, we just construct a point in the parameter space with empirical risk smaller than $f(\tilde{W})$.

Therefore, $\left\{\left[\hat{W}_i\right]_{i=1}^{2}, \left[\hat{b}_i\right]_{i=1}^{2}\right\}$ is a spurious local minimum.

The proof is completed. $\qquad\square$

### A.3 STAGE (2)

Stage (2) proves that neural networks with arbitrary hidden layers and two-piece linear activation $h_{s_-, s_+}$ have spurious local minima. Here, we still assume $s_+ \neq 0$. We have justified this assumption in Stage (1).

This stage is organized similarly with Stage (1): (a) Lemma 5 constructs a local minimum; and (b) Theorem 5 proves the minimum is spurious.

**Step (a). Construct local minima of the loss surface.**

**Lemma 5.** *Suppose that all the conditions of Lemma 4 hold, while the neural network has $L - 1$ hidden layers. Then, this network has a local minimum at*

$$\hat{W}_1' = \begin{bmatrix} \left[\tilde{W}\right]_{\cdot,[1:d_X]} \\ \mathbf{0}_{(d_1 - d_Y) \times d_X} \end{bmatrix}, \; \hat{b}_1' = \begin{bmatrix} \left[\tilde{W}\right]_{\cdot,d_X+1} - \eta \mathbf{1}_{d_Y} \\ -\eta \mathbf{1}_{d_1 - d_Y} \end{bmatrix},$$

$$\hat{W}_i' = \frac{1}{s_+} \sum_{j=1}^{d_Y} E_{j,j} + \frac{1}{s_+} \sum_{j=d_Y+1}^{d_i} E_{j,(d_Y+1)}, \; \hat{b}_i' = 0 \; (i = 2, 3, ..., L-1),$$

*and*

$$\hat{W}_L' = \begin{bmatrix} \frac{1}{s_+} I_{d_Y} & \mathbf{0}_{d_Y \times (d_{L-1} - d_Y)} \end{bmatrix}, \; \hat{b}_L' = \eta \mathbf{1}_{d_Y},$$

*where $\hat{W}_i'$ and $\hat{b}_i'$ are the weight matrix and the bias of the $i$-th layer, respectively, and $\eta$ is a negative constant with absolute value sufficiently large such that*

$$\tilde{W}\tilde{X} - \eta \mathbf{1}_{d_Y} \mathbf{1}_n^T > \mathbf{0}, \tag{27}$$

*where $>$ is element-wise.*

*Proof.* Recall the discussion in Lemma 4 that all components of $\hat{W}_1 X + \hat{b}_1 \mathbf{1}_n^T$ are positive. Specifically,

$$\hat{W}_1 X + \hat{b}_1 \mathbf{1}_n^T = \begin{bmatrix} \tilde{Y} - \eta \mathbf{1}_{d_Y} \mathbf{1}_n^T \\ -\eta \mathbf{1}_{d_1 - d_Y} \mathbf{1}_n^T \end{bmatrix},$$

where $\tilde{Y}$ is defined in Lemma 4.

Similar to the discussions in Lemma 4, when the parameters equal to $\left\{ \left[ \hat{W}_i' \right]_{i=1}^L, \left[ \hat{b}_i' \right]_{i=1}^L \right\}$, the output of the first layer before the activation function is

$$\tilde{Y}^{(1)} = \hat{W}_1' X + \hat{b}_1' \mathbf{1}_n^T = \begin{bmatrix} \tilde{Y} - \eta \mathbf{1}_{d_Y} \mathbf{1}_n^T \\ -\eta \mathbf{1}_{d_1 - d_Y} \mathbf{1}_n^T \end{bmatrix},$$

and

$$\tilde{Y} - \eta \mathbf{1}_{d_Y} \mathbf{1}_n^T > \mathbf{0}, \tag{28}$$

$$-\eta \mathbf{1}_{d_1 - d_Y} \mathbf{1}_n^T > \mathbf{0}. \tag{29}$$

Here $>$ is defined element-wise.

After the activation function, the output of the first layer is

$$Y^{(1)} = h_{s_-, s_+}(\hat{W}_1' X + \hat{b}_1' \mathbf{1}_n^T) = s_+(\hat{W}_1' X + \hat{b}_1' \mathbf{1}_n^T) = s_+ \begin{bmatrix} \tilde{Y} - \eta \mathbf{1}_{d_Y} \mathbf{1}_n^T \\ -\eta \mathbf{1}_{d_1 - d_Y} \mathbf{1}_n^T \end{bmatrix}.$$

We prove by induction that for all $i \in [1 : L - 1]$ that

$$\tilde{Y}^{(i)} > \mathbf{0} \text{ , element-wise}, \tag{30}$$

$$Y^{(i)} = s_+ \begin{bmatrix} \tilde{Y} - \eta \mathbf{1}_{d_Y} \mathbf{1}_n^T \\ -\eta \mathbf{1}_{d_i - d_Y} \mathbf{1}_n^T \end{bmatrix}. \tag{31}$$

Suppose that for $1 \le k \le L - 2$, $\tilde{Y}^{(k)}$ is positive (element-wise) and

$$Y^{(k)} = s_+ \begin{bmatrix} \tilde{Y} - \eta \mathbf{1}_{d_Y} \mathbf{1}_n^T \\ -\eta \mathbf{1}_{d_k - d_Y} \mathbf{1}_n^T \end{bmatrix}.$$

Then the output of the $(k + 1)$-th layer before the activation is

$$\begin{aligned} \tilde{Y}^{(k+1)} &= \hat{W}_{k+1}' Y^{(k)} + \hat{b}_{k+1}' \mathbf{1}_n^T \\ &= \frac{1}{s_+} \left( \sum_{j=1}^{d_Y} E_{j,j} + \sum_{j=d_Y+1}^{d_{k+1}} E_{j,(d_Y+1)} \right) s_+ \begin{bmatrix} \tilde{Y} - \eta \mathbf{1}_{d_Y} \mathbf{1}_n^T \\ -\eta \mathbf{1}_{d_k - d_Y} \mathbf{1}_n^T \end{bmatrix} \\ &= \begin{bmatrix} \tilde{Y} - \eta \mathbf{1}_{d_Y} \mathbf{1}_n^T \\ -\eta \mathbf{1}_{d_{k+1} - d_Y} \mathbf{1}_n^T \end{bmatrix}. \end{aligned}$$

Applying eqs. (28) and (29), we have

$$\tilde{Y}^{(k+1)} = \begin{bmatrix} \tilde{Y} - \eta \mathbf{1}_{d_Y} \mathbf{1}_n^T \\ -\eta \mathbf{1}_{d_{k+1} - d_Y} \mathbf{1}_n^T \end{bmatrix} > \mathbf{0},$$

where $>$ is defined element-wise. Therefore,

$$Y^{(k+1)} = h_{s_-, s_+} \left( \tilde{Y}^{(k+1)} \right) = s_+ \tilde{Y}^{(k+1)} = s_+ \begin{bmatrix} \tilde{Y} - \eta \mathbf{1}_{d_Y} \mathbf{1}_n^T \\ -\eta \mathbf{1}_{d_{k+1} - d_Y} \mathbf{1}_n^T \end{bmatrix}.$$

We thereby prove eqs. (30) and (31).

Therefore, $Y^{(L)}$ can be calculated as

$$\begin{aligned} \hat{Y} = Y^{(L)} &= \hat{W}_L' Y^{(L-1)} + \hat{b}_L' \mathbf{1}_n^T \\ &= \frac{1}{s_+} \begin{bmatrix} I_{d_Y} & \mathbf{0}_{d_Y \times (d_{L-1} - d_Y)} \end{bmatrix} s_+ \begin{bmatrix} \tilde{Y} - \eta \mathbf{1}_{d_Y} \mathbf{1}_n^T \\ -\eta \mathbf{1}_{d_i - d_Y} \mathbf{1}_n^T \end{bmatrix} + \eta \mathbf{1}_{d_Y} \mathbf{1}_n^T \\ &= \tilde{Y}. \end{aligned} \tag{32}$$

Then, we show the empirical risk is higher around $\left\{ \left[\hat{W}_i'\right]_{i=1}^L, \left[\hat{b}_i'\right]_{i=1}^L \right\}$ in order to prove that $\left\{ \left[\hat{W}_i'\right]_{i=1}^L, \left[\hat{b}_i'\right]_{i=1}^L \right\}$ is a local minimizer.

Let $\left\{ \left[\hat{W}_i' + \delta_{Wi}'\right]_{i=1}^L, \left[\hat{b}_i' + \delta_{bi}'\right]_{i=1}^L \right\}$ be point in the parameter space which is close enough to the point $\left\{ \left[\hat{W}_i'\right]_{i=1}^L, \left[\hat{b}_i'\right]_{i=1}^L \right\}$. Since the disturbances $\delta_{Wi}'$ and $\delta_{bi}'$ are both close to 0 (element-wise), all components of $\tilde{Y}^{(i)}\left( \left[\hat{W}_i' + \delta_{Wi}'\right]_{i=1}^L, \left[\hat{b}_i' + \delta_{bi}'\right]_{i=1}^L \right)$ remains positive. Therefore, the output of the neural network in terms of parameters $\left\{ \left[\hat{W}_i' + \delta_{Wi}'\right]_{i=1}^L, \left[\hat{b}_i' + \delta_{bi}'\right]_{i=1}^L \right\}$ is

$$
\begin{aligned}
\hat{Y}&\left( \left[\hat{W}_i' + \delta_{Wi}'\right]_{i=1}^L, \left[\hat{b}_i' + \delta_{bi}'\right]_{i=1}^L \right) \\
=&(\hat{W}_L' + \delta_{WL}')h_{s_-,s_+}\left( \ldots h_{s_-,s_+}\left( \left(\hat{W}_1' + \delta_{W1}'\right)X + \left(\hat{b}_1' + \delta_{b1}'\right)1_n^T \right) \ldots \right) \\
&+ \left(\hat{b}_L' + \delta_{bL}'\right)\mathbf{1}_n^T \\
=&(\hat{W}_L' + \delta_{WL}')s_+\left( \ldots s_+\left( \left(\hat{W}_1' + \delta_{W1}'\right)X + \left(\hat{b}_1' + \delta_{b1}'\right)1_n^T \right) \ldots \right) \\
&+ \left(\hat{b}_L' + \delta_{bL}'\right)\mathbf{1}_n^T \\
=&M_1 X + M_2 1_n^T,
\end{aligned}
$$

where $M_1$ and $M_2$ can be obtained from $\left\{ \left[\hat{W}_i'\right]_{i=1}^L, \left[\hat{b}_i'\right]_{i=1}^L \right\}$ and $\left\{ [\delta_{Wi}']_{i=1}^L, [\delta_{bi}']_{i=1}^L \right\}$ through several multiplication and summation operations[2].

Rewrite the output as

$$
M_1 X + M_2 1_n^T = \begin{bmatrix} M_1 & M_2 \end{bmatrix} \begin{bmatrix} X \\ 1_n^T \end{bmatrix}.
$$

Therefore, the empirical risk $\hat{\mathcal{R}}$ before and after the disturbance can be expressed as $f(\tilde{W})$ and $f(\begin{bmatrix} M_1 & M_2 \end{bmatrix})$, respectively.

When the disturbances $\left\{ [\delta_{Wi}']_{i=1}^L, [\delta_{bi}']_{i=1}^L \right\}$ approach 0 (element-wise), $\begin{bmatrix} M_1 & M_2 \end{bmatrix}$ approaches $\tilde{W}$. Therefore, when $\left\{ [\delta_{Wi}']_{i=1}^L, [\delta_{bi}']_{i=1}^L \right\}$ are all small enough, we have

$$
\begin{aligned}
\hat{\mathcal{R}}&\left( \left[\hat{W}_i' + \delta_{Wi}'\right]_{i=1}^L, \left[\hat{b}_i' + \delta_{bi}'\right]_{i=1}^L \right) \\
=&f(\begin{bmatrix} M_1 & M_2 \end{bmatrix}) \\
\geq&f(\tilde{W}) \\
=&\hat{\mathcal{R}}\left( \left[\hat{W}_i'\right]_{i=1}^L, \left[\hat{b}_i'\right]_{i=1}^L \right).
\end{aligned}
\tag{33}
$$

Since $\left\{ \left[\hat{W}_i' + \delta_{Wi}'\right]_{i=1}^L, \left[\hat{b}_i' + \delta_{bi}'\right]_{i=1}^L \right\}$ are arbitrary within a sufficiently small neighbour of $\left\{ \left[\hat{W}_i'\right]_{i=1}^L, \left[\hat{b}_i'\right]_{i=1}^L \right\}$, eq. (33) yields that $\left\{ \left[\hat{W}_i'\right]_{i=1}^2, \left[\hat{b}_i'\right]_{i=1}^2 \right\}$ is a local minimizer. $\qquad\square$

**Step (b). Prove the constructed local minima are spurious.**

---

[2]Since the exact form of $M_1$ and $M_2$ are not needed, we omit the exact formulations here.

**Theorem 5.** *Under the same conditions of Lemma 5 and Assumptions 1, 2, and 4, the constructed spurious local minima in Lemma 5 are spurious.*

*Proof.* We first construct the weight matrix and bias of the $i$-th layer as follows,

$$\tilde{W}_1' = \tilde{W}_1, \; \tilde{b}_1' = \tilde{b}_1,$$

$$\tilde{W}_2' = \begin{bmatrix} \tilde{W}_2 \\ \mathbf{0}_{(d_2-d_Y)\times d_1} \end{bmatrix}, \; \tilde{b}_2' = \lambda \mathbf{1}_{d_2} + \begin{bmatrix} \tilde{b}_2 \\ \mathbf{0}_{(d_2-d_Y)\times 1} \end{bmatrix},$$

$$\tilde{W}_i' = \frac{1}{s_+} \sum_{i=1}^{d_Y} E_{i,i}, \; \tilde{b}_i' = 0 \; (i = 3, 4, ..., L-1),$$

and

$$\tilde{W}_L' = \frac{1}{s_+} \sum_{i=1}^{d_Y} E_{i,i}, \; \tilde{b}_L' = -\lambda \mathbf{1}_{d_Y},$$

where $\tilde{W}_1$, $\tilde{W}_2$, $\tilde{b}_1$ and $\tilde{b}_2$ are defined by eqs. (19), (20), (21), and (22), respectively, and $\lambda$ is a sufficiently large positive real such that

$$\hat{Y}\left(\left[\tilde{W}_i\right]_{i=1}^2, \left[\tilde{b}_i\right]_{i=1}^2\right) + \lambda \mathbf{1}_{d_2} \mathbf{1}_n^T > \mathbf{0}, \tag{34}$$

where $>$ is defined element-wise.

We argue that $\left\{\left[\tilde{W}_i'\right]_{i=1}^L, \left[\tilde{b}_i'\right]_{i=1}^L\right\}$ corresponds to a smaller empirical risk than $f(\tilde{W})$ which is defined in Lemma 4.

First, Theorem 4 has proved that the point $\left\{\tilde{W}_1, \tilde{W}_2, \tilde{b}_1, \tilde{b}_2\right\}$ corresponds to a smaller empirical risk than $f(\tilde{W})$.

We prove by induction that for any $i \in \{3, 4, ..., L-1\}$,

$$\tilde{Y}^{(i)}\left(\left[\tilde{W}_i'\right]_{i=1}^L, \left[\tilde{b}_i'\right]_{i=1}^L\right) \geq \mathbf{0} \text{ , element-wise,} \tag{35}$$

$$Y^{(i)}\left(\left[\tilde{W}_i'\right]_{i=1}^L, \left[\tilde{b}_i'\right]_{i=1}^L\right) = s_+ \begin{bmatrix} \hat{Y}\left(\left[\tilde{W}_i\right]_{i=1}^2, \left[\tilde{b}_i\right]_{i=1}^2\right) + \lambda \mathbf{1}_{d_Y} \mathbf{1}_n^T \\ 0_{(d_i-d_Y)\times n} \end{bmatrix}. \tag{36}$$

Apparently the output of the first layer before the activation is

$$\tilde{Y}^{(1)}\left(\left[\tilde{W}_i'\right]_{i=1}^L, \left[\tilde{b}_i'\right]_{i=1}^L\right) = \tilde{W}_1' X + \tilde{b}_1' \mathbf{1}_n^T = \tilde{W}_1 X + \tilde{b}_1 \mathbf{1}_n^T = \tilde{Y}^{(1)}\left(\left[\tilde{W}_i\right]_{i=1}^2, \left[\tilde{b}_i\right]_{i=1}^2\right).$$

Therefore, the output of the first layer after the activation is

$$Y^{(1)}\left(\left[\tilde{W}_i'\right]_{i=1}^L, \left[\tilde{b}_i'\right]_{i=1}^L\right) = h_{s_-,s_+}\left(\tilde{Y}^{(1)}\left(\left[\tilde{W}_i'\right]_{i=1}^L, \left[\tilde{b}_i'\right]_{i=1}^L\right)\right)$$

$$= h_{s_-,s_+}\left(\tilde{Y}^{(1)}\left(\left[\tilde{W}_i\right]_{i=1}^L, \left[\tilde{b}_i\right]_{i=1}^L\right)\right)$$

$$= Y^{(1)}\left(\left[\tilde{W}_i\right]_{i=1}^2, \left[\tilde{b}_i\right]_{i=1}^2\right).$$

Thus, the output of the second layer before the activation is

$$
\begin{aligned}
\tilde{Y}^{(2)}\left(\left[\tilde{W}_i'\right]_{i=1}^{L}, \left[\tilde{b}_i'\right]_{i=1}^{L}\right) &= \tilde{W}_2' Y^{(1)}\left(\left[\tilde{W}_i'\right]_{i=1}^{L}, \left[\tilde{b}_i'\right]_{i=1}^{L}\right) + \tilde{b}_2' \mathbf{1}_n^T \\
&= \left[\begin{array}{c} \tilde{W}_2 \\ \mathbf{0}_{(d_2 - d_Y) \times d_1} \end{array}\right] Y^{(1)}\left(\left[\tilde{W}_i'\right]_{i=1}^{L}, \left[\tilde{b}_i'\right]_{i=1}^{L}\right) + \left[\begin{array}{c} \tilde{b}_2 \\ \mathbf{0}_{(d_2 - d_Y) \times 1} \end{array}\right] \mathbf{1}_n^T \\
&\quad + \lambda \mathbf{1}_{d_2} \mathbf{1}_n^T \\
&= \left[\begin{array}{c} \hat{Y}\left(\left[\tilde{W}_i\right]_{i=1}^{2}, \left[\tilde{b}_i\right]_{i=1}^{2}\right) + \lambda \mathbf{1}_{d_Y} \mathbf{1}_n^T \\ \\ \lambda \mathbf{1}_{d_2 - d_Y} \mathbf{1}_n^T \end{array}\right].
\end{aligned}
$$

Applying the definition of $\lambda$ (eq. (34)),

$$
\tilde{Y}^{(2)}\left(\left[\tilde{W}_i'\right]_{i=1}^{L}, \left[\tilde{b}_i'\right]_{i=1}^{L}\right) > \mathbf{0} \text{ , element-wise.} \tag{37}
$$

Therefore, the output of the second layer after the activation is

$$
\begin{aligned}
Y^{(2)}\left(\left[\tilde{W}_i'\right]_{i=1}^{L}, \left[\tilde{b}_i'\right]_{i=1}^{L}\right) &= h_{s_-, s_+}\left(\tilde{Y}^{(2)}\left(\left[\tilde{W}_i'\right]_{i=1}^{L}, \left[\tilde{b}_i'\right]_{i=1}^{L}\right)\right) \\
&= s_+ \left[\begin{array}{c} \hat{Y}\left(\left[\tilde{W}_i\right]_{i=1}^{2}, \left[\tilde{b}_i\right]_{i=1}^{2}\right) + \lambda \mathbf{1}_{d_Y} \mathbf{1}_n^T \\ \\ \lambda \mathbf{1}_{d_2 - d_Y} \mathbf{1}_n^T \end{array}\right].
\end{aligned}
$$

Meanwhile, the output of the third layer before the activation is $\tilde{Y}^{(3)}\left(\left[\tilde{W}_i'\right]_{i=1}^{L}, \left[\tilde{b}_i'\right]_{i=1}^{L}\right)$ can be calculated based on $Y^{(2)}\left(\left[\tilde{W}_i'\right]_{i=1}^{L}, \left[\tilde{b}_i'\right]_{i=1}^{L}\right)$:

$$
\begin{aligned}
\tilde{Y}^{(3)}\left(\left[\tilde{W}_i'\right]_{i=1}^{L}, \left[\tilde{b}_i'\right]_{i=1}^{L}\right) &= \tilde{W}_3' Y^{(2)}\left(\left[\tilde{W}_i'\right]_{i=1}^{L}, \left[\tilde{b}_i'\right]_{i=1}^{L}\right) + \tilde{b}_3' \mathbf{1}_n^T \\
&= \frac{1}{s_+}\left(\sum_{i=1}^{d_Y} E_{i,i}\right) s_+ \left[\begin{array}{c} \hat{Y}\left(\left[\tilde{W}_i\right]_{i=1}^{2}, \left[\tilde{b}_i\right]_{i=1}^{2}\right) + \lambda \mathbf{1}_{d_Y} \mathbf{1}_n^T \\ \\ \lambda \mathbf{1}_{d_2 - d_Y} \mathbf{1}_n^T \end{array}\right] \\
&= \left[\begin{array}{c} \hat{Y}\left(\left[\tilde{W}_i\right]_{i=1}^{2}, \left[\tilde{b}_i\right]_{i=1}^{2}\right) + \lambda \mathbf{1}_{d_Y} \mathbf{1}_n^T \\ \\ \mathbf{0}_{(d_3 - d_Y) \times n} \end{array}\right].
\end{aligned}
$$

Applying eq. (37),

$$
\tilde{Y}^{(3)}\left(\left[\tilde{W}_i'\right]_{i=1}^{L}, \left[\tilde{b}_i'\right]_{i=1}^{L}\right) \geq \mathbf{0} \text{ , element-wise.} \tag{38}
$$

Therefore, the output of the third layer after the activation is

$$
\begin{aligned}
Y^{(3)}\left(\left[\tilde{W}_i'\right]_{i=1}^{L}, \left[\tilde{b}_i'\right]_{i=1}^{L}\right) &= h_{s_-, s_+}\left(\tilde{Y}^{(3)}\left(\left[\tilde{W}_i'\right]_{i=1}^{L}, \left[\tilde{b}_i'\right]_{i=1}^{L}\right)\right) \\
&= s_+ \left(\tilde{Y}^{(3)}\left(\left[\tilde{W}_i'\right]_{i=1}^{L}, \left[\tilde{b}_i'\right]_{i=1}^{L}\right)\right) \\
&= s_+ \left[\begin{array}{c} \hat{Y}\left(\left[\tilde{W}_i\right]_{i=1}^{2}, \left[\tilde{b}_i\right]_{i=1}^{2}\right) + \lambda \mathbf{1}_{d_Y} \mathbf{1}_n^T \\ \\ \mathbf{0}_{(d_3 - d_Y) \times n} \end{array}\right].
\end{aligned}
$$

Suppose eqs. (35) and (36) hold for $k$ ($3 \le k \le L - 2$), when $k + 1$,

$$
\begin{aligned}
\tilde{Y}^{(k+1)} \left( \left[ \tilde{W}'_i \right]_{i=1}^{L}, \left[ \tilde{b}'_i \right]_{i=1}^{L} \right) &= \tilde{W}'_{k+1} Y^{(k)} \left( \left[ \tilde{W}'_i \right]_{i=1}^{2}, \left[ \tilde{b}'_i \right]_{i=1}^{2} \right) + \tilde{b}'_{k+1} \mathbf{1}_n^T \\
&= \frac{1}{s_+} \left( \sum_{i=1}^{d_Y} E_{i,i} \right) s_+ \left[ \begin{array}{c} \hat{Y} \left( \left[ \tilde{W}_i \right]_{i=1}^{2}, \left[ \tilde{b}_i \right]_{i=1}^{2} \right) + \lambda \mathbf{1}_{d_Y} \mathbf{1}_n^T \\ \mathbf{0}_{(d_k - d_Y) \times n} \end{array} \right] \\
&= \left[ \begin{array}{c} \hat{Y} \left( \left[ \tilde{W}_i \right]_{i=1}^{2}, \left[ \tilde{b}_i \right]_{i=1}^{2} \right) + \lambda \mathbf{1}_{d_Y} \mathbf{1}_n^T \\ \mathbf{0}_{(d_{k+1} - d_Y) \times n} \end{array} \right].
\end{aligned}
$$

Applying eq. (38),

$$
\tilde{Y}^{(k+1)} \left( \left[ \tilde{W}'_i \right]_{i=1}^{L}, \left[ \tilde{b}'_i \right]_{i=1}^{L} \right) \ge \mathbf{0} \text{ , element-wise.} \tag{39}
$$

Therefore, the output of the $(k + 1)$-th layer after the activation is

$$
\begin{aligned}
Y^{(k+1)} \left( \left[ \tilde{W}'_i \right]_{i=1}^{L}, \left[ \tilde{b}'_i \right]_{i=1}^{L} \right) &= h_{s_-, s_+} \left( \tilde{Y}^{(k+1)} \left( \left[ \tilde{W}'_i \right]_{i=1}^{L}, \left[ \tilde{b}'_i \right]_{i=1}^{L} \right) \right) \\
&= s_+ \tilde{Y}^{(k+1)} \left( \left[ \tilde{W}'_i \right]_{i=1}^{L}, \left[ \tilde{b}'_i \right]_{i=1}^{L} \right) \\
&= s_+ \left[ \begin{array}{c} \hat{Y} \left( \left[ \tilde{W}_i \right]_{i=1}^{2}, \left[ \tilde{b}_i \right]_{i=1}^{2} \right) + \lambda \mathbf{1}_{d_Y} \mathbf{1}_n^T \\ \mathbf{0}_{(d_{k+1} - d_Y) \times n} \end{array} \right].
\end{aligned}
$$

Therefore, eqs. (35) and (36) hold for any $i \in \{3, 4, ..., L - 1\}$.

Finally, the output of the network is

$$
\begin{aligned}
\hat{Y} \left( \left[ \tilde{W}'_i \right]_{i=1}^{L}, \left[ \tilde{b}'_i \right]_{i=1}^{L} \right) &= Y^{(L)} \left( \left[ \tilde{W}'_i \right]_{i=1}^{L}, \left[ \tilde{b}'_i \right]_{i=1}^{L} \right) \\
&= \tilde{W}'_L Y^{(L-1)} \left( \left[ \tilde{W}'_i \right]_{i=1}^{L}, \left[ \tilde{b}'_i \right]_{i=1}^{L} \right) + \tilde{b}'_L \mathbf{1}_n^T \\
&= \left( \frac{1}{s_+} \sum_{i=1}^{d_Y} E_{i,i} \right) s_+ \left[ \begin{array}{c} \hat{Y} \left( \left[ \tilde{W}_i \right]_{i=1}^{2}, \left[ \tilde{b}_i \right]_{i=1}^{2} \right) + \lambda \mathbf{1}_{d_Y} \mathbf{1}_n^T \\ 0_{(d_{L-1} - d_Y) \times n} \end{array} \right] \\
&\quad - \lambda \mathbf{1}_{d_Y} \mathbf{1}_n^T \\
&= \hat{Y} \left( \left[ \tilde{W}_i \right]_{i=1}^{2}, \left[ \tilde{b}_i \right]_{i=1}^{2} \right).
\end{aligned}
$$

Applying Theorem 4, we have

$$
\hat{\mathcal{R}} \left( \left[ \tilde{W}'_i \right]_{i=1}^{L}, \left[ \tilde{W}'_i \right]_{i=1}^{L} \right) = \hat{\mathcal{R}} \left( \left[ \tilde{W}_i \right]_{i=1}^{2}, \left[ \tilde{b}_i \right]_{i=1}^{2} \right) < f(\tilde{W}).
$$

The proof is completed. $\qquad\square$

### A.4 STAGE (3)

Finally, we prove Theorem 1.

This stage also follows the two-step strategy.

**Step (a). Construct local minima of the loss surface.**

**Lemma 6.** *Suppose $t$ is a non-differentiable point for the piece-wise linear activation function $h$ and $\sigma$ is a constant such that the activation $h$ is differentiable in the intervals $(t - \sigma, t)$ and $(t, t + \sigma)$. Assume that $M$ is a sufficiently large positive real such that*

$$\frac{1}{M} \left\| \hat{W}_1' X + \hat{b}_1' \mathbf{1}_n^T \right\|_F < \sigma. \tag{40}$$

*Let $\alpha_i$ be any positive real such that*

$$\alpha_1 = 1$$
$$0 < \alpha_i < 1, \ i = 2, \cdots, L - 1. \tag{41}$$

*Then, under Assumption 3, any neural network with piecewise linear activations and $L - 1$ hidden layers has local minima at*

$$\hat{W}_1'' = \frac{1}{M} \hat{W}_1', \ \hat{b}_1'' = \frac{1}{M} \hat{b}_1' + t \mathbf{1}_{d_1},$$

$$\hat{W}_i'' = \alpha_i \hat{W}_i', \ \hat{b}_i'' = -\alpha_i \hat{W}_i' h(t) \mathbf{1}_{d_{i-1}} + t \mathbf{1}_{d_i} + \frac{\Pi_{j=2}^i \alpha_j}{M} \hat{b}_i', \ (i = 2, 3, ..., L - 1),$$

*and*

$$\hat{W}_L'' = \frac{1}{\Pi_{j=2}^L \alpha_j} M \hat{W}_L', \ \hat{b}_L'' = -\frac{M}{\Pi_{j=2}^{L-1} \alpha_j} \hat{W}_L' h(t) \mathbf{1}_{d_{L-1}} + \hat{b}_L'$$

*where $\left\{ \left[ \hat{W}_i' \right]_{i=1}^L, \left[ \hat{b}_i' \right]_{i=1}^L \right\}$ is the local minimizer constructed in Lemma 5. Also, the loss is continuously differentiable, whose derivative with respect to the prediction $\hat{Y}_i$ may equal to $0$ only when the prediction $\hat{Y}_i$ and label $Y_i$ are the same.*

*Proof.* Define $s_- = \lim_{\theta \to 0^-} h'(\theta)$ and $s_+ = \lim_{\theta \to 0^+} h'(\theta)$.

We then prove by induction that for all $i \in [1 : L - 1]$, all components of the $i$-th layer output before the activation $\tilde{Y}^{(i)} \left( \left[ \hat{W}_i'' \right]_{i=1}^L, \left[ \hat{b}_i'' \right]_{i=1}^L \right)$ are in interval $(t, t + \sigma)$, and

$$Y^{(i)} \left( \left[ \hat{W}_i'' \right]_{i=1}^L, \left[ \hat{b}_i'' \right]_{i=1}^L \right) = h(t) \mathbf{1}_{d_i} \mathbf{1}_n^T + \frac{\Pi_{j=1}^i \alpha_j}{M} Y^{(i)} \left( \left[ \hat{W}_i' \right]_{i=1}^L, \left[ \hat{b}_i' \right]_{i=1}^L \right).$$

The first layer output before the activation is,

$$\tilde{Y}^{(1)} \left( \left[ \hat{W}_i'' \right]_{i=1}^L, \left[ \hat{b}_i'' \right]_{i=1}^L \right) = \hat{W}_1'' X + \hat{b}_1'' \mathbf{1}_n^T = \frac{1}{M} \hat{W}_1' X + \frac{1}{M} \hat{b}_1' \mathbf{1}_n^T + t \mathbf{1}_{d_1} \mathbf{1}_n^T. \tag{42}$$

We proved in Lemma 5 that $\hat{W}_1' X + \hat{b}_1' \mathbf{1}_n^T$ is positive (element-wise). Since the Frobenius norm of a matrix is no smaller than any component's absolute value, applying eq. (40), we have that for all $i \in [1, d_1]$ and $j \in [1 : n]$,

$$0 < \frac{1}{M} \left( \hat{W}_1' X + \hat{b}_1' \mathbf{1}_n^T \right)_{ij} < \sigma. \tag{43}$$

Therefore, $\left( \frac{1}{M} \left( \hat{W}_1' X + \hat{b}_1' \mathbf{1}_n^T \right)_{ij} + t \right) \in (t, t + \sigma)$. So,

$$\begin{aligned}
Y^{(1)} \left( \left[ \hat{W}_i'' \right]_{i=1}^L, \left[ \hat{b}_i'' \right]_{i=1}^L \right) &= h \left( \tilde{Y}^{(1)} \left( \left[ \hat{W}_i'' \right]_{i=1}^L, \left[ \hat{b}_i'' \right]_{i=1}^L \right) \right) \\
&\stackrel{(*)}{=} h_{s_-, s_+} \left( \frac{1}{M} \hat{W}_1' X + \frac{1}{M} \hat{b}_1' \mathbf{1}_n^T \right) + h(t) \mathbf{1}_{d_1} \mathbf{1}_n^T \\
&= \frac{1}{M} Y^{(1)} \left( \left[ \hat{W}_i' \right]_{i=1}^L, \left[ \hat{b}_i' \right]_{i=1}^L \right) + h(t) \mathbf{1}_{d_1} \mathbf{1}_n^T,
\end{aligned}$$

where eq.$(*)$ is because for any $x \in (t - \sigma, t + \sigma)$,

$$h(x) = h(t) + h_{s_-, s_+}(x - t). \tag{44}$$

Suppose the above argument holds for $k$ $(1 \leq k \leq L - 2)$. Then

$$
\tilde{Y}^{(k+1)} \left( \left[ \hat{W}_i'' \right]_{i=1}^L, \left[ \hat{b}_i'' \right]_{i=1}^L \right)
$$

$$
= \hat{W}_{k+1}'' Y^{(k)} \left( \left[ \hat{W}_i'' \right]_{i=1}^L, \left[ \hat{b}_i'' \right]_{i=1}^L \right) + \hat{b}_{k+1}'' \mathbf{1}_n^T
$$

$$
= (-\alpha_{k+1} \hat{W}_{k+1}' h(t) \mathbf{1}_{d_{k+1}}
$$

$$
+ t \mathbf{1}_{d_{k+1}} + \frac{\Pi_{i=1}^{k+1} \alpha_i}{M} \hat{b}_{k+1}') \mathbf{1}_n^T + \alpha_{k+1} \hat{W}_{k+1}' Y^{(k)} \left( \left[ \hat{W}_i'' \right]_{i=1}^L, \left[ \hat{b}_i'' \right]_{i=1}^L \right)
$$

$$
= \alpha_{k+1} \hat{W}_{k+1}' \left( h(t) \mathbf{1}_{d_k} \mathbf{1}_n^T + \frac{\Pi_{i=1}^k \alpha_i}{M} Y^{(k)} \left( \left[ \hat{W}_i' \right]_{i=1}^L, \left[ \hat{b}_i' \right]_{i=1}^L \right) \right)
$$

$$
+ \left( -\alpha_{k+1} \hat{W}_{k+1}' h(t) \mathbf{1}_{d_k} + t \mathbf{1}_{d_{k+1}} + \frac{\Pi_{i=1}^{k+1} \alpha_i}{M} \hat{b}_{k+1}' \right) \mathbf{1}_n^T
$$

$$
= \frac{\Pi_{i=1}^{k+1} \alpha_i}{M} \hat{W}_{k+1}' Y^{(k)} \left( \left[ \hat{W}_i' \right]_{i=1}^L, \left[ \hat{b}_i' \right]_{i=1}^L \right) + \frac{\Pi_{i=1}^{k+1} \alpha_i}{M} \hat{b}_{k+1}' \mathbf{1}_n^T + t \mathbf{1}_{d_{k+1}} \mathbf{1}_n^T
$$

$$
= t \mathbf{1}_{d_{k+1}} \mathbf{1}_n^T + \frac{\Pi_{i=1}^{k+1} \alpha_i}{M} \tilde{Y}^{(k+1)} \left( \left[ \hat{W}_i' \right]_{i=1}^L, \left[ \hat{b}_i' \right]_{i=1}^L \right) .
$$

Lemma 5 has proved that all components of $\tilde{Y}^{(k+1)} \left( \left[ \hat{W}_i' \right]_{i=1}^L, \left[ \hat{b}_i' \right]_{i=1}^L \right)$ are contained in $\tilde{Y}^{(1)} \left( \left[ \hat{W}_i' \right]_{i=1}^L, \left[ \hat{b}_i' \right]_{i=1}^L \right)$. Combining

$$
t \mathbf{1}_{d_1} \mathbf{1}_n^T < t \mathbf{1}_{d_1} \mathbf{1}_n^T + \frac{1}{M} \tilde{Y}^{(1)} \left( \left[ \hat{W}_i' \right]_{i=1}^L, \left[ \hat{b}_i' \right]_{i=1}^L \right) < (t + \sigma) \mathbf{1}_{d_1} \mathbf{1}_n^T,
$$

we have

$$
t \mathbf{1}_{d_{k+1}} \mathbf{1}_n^T < t \mathbf{1}_{d_{k+1}} \mathbf{1}_n^T + \frac{\Pi_{i=1}^{k+1} \alpha_i}{M} \tilde{Y}^{(k+1)} \left( \left[ \hat{W}_i' \right]_{i=1}^L, \left[ \hat{b}_i' \right]_{i=1}^L \right) \overset{(*)}{<} (t + \sigma) \mathbf{1}_{d_{k+1}} \mathbf{1}_n^T.
$$

Here $<$ are all element-wise, and inequality $(*)$ comes from the property of $\alpha_i$ (eq. (41)).
Furthermore, the $(k + 1)$-th layer output after the activation is

$$
Y^{(k+1)} \left( \left[ \hat{W}_i'' \right]_{i=1}^L, \left[ \hat{b}_i'' \right]_{i=1}^L \right) = h \left( \tilde{Y}^{(k+1)} \left( \left[ \hat{W}_i'' \right]_{i=1}^L, \left[ \hat{b}_i'' \right]_{i=1}^L \right) \right)
$$

$$
= h \left( t \mathbf{1}_{d_{k+1}} \mathbf{1}_n^T + \frac{1}{M} \tilde{Y}^{(k+1)} \left( \left[ \hat{W}_i' \right]_{i=1}^L, \left[ \hat{b}_i' \right]_{i=1}^L \right) \right)
$$

$$
\overset{(*)}{=} h(t) \mathbf{1}_{d_{k+1}} \mathbf{1}_n^T + h_{s_-, s_+} \left( \frac{\Pi_{i=1}^{k+1} \alpha_i}{M} \tilde{Y}^{(k+1)} \left( \left[ \hat{W}_i' \right]_{i=1}^L, \left[ \hat{b}_i' \right]_{i=1}^L \right) \right)
$$

$$
= h(t) \mathbf{1}_{d_{k+1}} \mathbf{1}_n^T + \frac{\Pi_{i=1}^{k+1} \alpha_i}{M} h_{s_-, s_+} \left( \tilde{Y}^{(k+1)} \left( \left[ \hat{W}_i' \right]_{i=1}^L, \left[ \hat{b}_i' \right]_{i=1}^L \right) \right)
$$

$$
= h(t) \mathbf{1}_{d_{k+1}} \mathbf{1}_n^T + \frac{\Pi_{i=1}^{k+1} \alpha_i}{M} Y^{(k+1)} \left( \left[ \hat{W}_i' \right]_{i=1}^L, \left[ \hat{b}_i' \right]_{i=1}^L \right),
$$

where eq. $(*)$ is because of eq. (44). The above argument is proved for any index $k \in \{1, \ldots, L-1\}$.

Therefore, the output of the network is

$$Y^{(L)}\left(\left[\hat{W}_i''\right]_{i=1}^L, \left[\hat{b}_i''\right]_{i=1}^L\right)$$

$$=\hat{W}_L''Y^{(L-1)}\left(\left[\hat{W}_i''\right]_{i=1}^L, \left[\hat{b}_i''\right]_{i=1}^L\right) + \hat{b}_L''\mathbf{1}_n^T$$

$$=\frac{M}{\Pi_{i=1}^{L-1}\alpha_i}\hat{W}_L'\left(h(t)\mathbf{1}_{d_{L-1}}\mathbf{1}_n^T + \frac{\Pi_{i=1}^{L-1}\alpha_i}{M}Y^{(L-1)}\left(\left[\hat{W}_i'\right]_{i=1}^L, \left[\hat{b}_i'\right]_{i=1}^L\right)\right)$$

$$+\left(-\frac{M}{\Pi_{i=1}^{L-1}\alpha_i}\hat{W}_L'h(t)\mathbf{1}_{d_{L-1}} + \hat{b}_L'\right)\mathbf{1}_n^T$$

$$=\hat{W}_L'Y^{(L-1)}\left(\left[\hat{W}_i'\right]_{i=1}^L, \left[\hat{b}_i'\right]_{i=1}^L\right) + \hat{b}_L'\mathbf{1}_n^T$$

$$=Y^{(L)}\left(\left[\hat{W}_i'\right]_{i=1}^L, \left[\hat{b}_i'\right]_{i=1}^L\right).$$

Therefore,

$$\hat{\mathcal{R}}\left(\left[\hat{W}_i''\right]_{i=1}^L, \left[\hat{b}_i''\right]_{i=1}^L\right) = \hat{\mathcal{R}}\left(\left[\hat{W}_i'\right]_{i=1}^L, \left[\hat{b}_i'\right]_{i=1}^L\right) = f(\tilde{W}).$$

We then introduce some small disturbances $\left\{[\delta_{Wi}'']_{i=1}^L, [\delta_{bi}'']_{i=1}^L\right\}$ into $\left\{\left[\hat{W}_i''\right]_{i=1}^L, \left[\hat{b}_i''\right]_{i=1}^L\right\}$ in order to check the local optimality.

Since all comonents of $Y^{(i)}$ are in interval $(t, t+\sigma)$, the activations in every hidden layers is realized at linear parts. Therefore, the output of network is

$$\hat{Y}\left(\left[\hat{W}_i'' + \delta_{Wi}''\right]_{i=1}^L, \left[\hat{b}_i'' + \delta_{bi}''\right]_{i=1}^L\right)$$

$$=\left(\hat{W}_L'' + \delta_{WL}''\right)h\left(\cdots h\left(\left(\hat{W}_1'' + \delta_{W1}''\right)X + \left(\hat{b}_1'' + \delta_{b1}''\right)\mathbf{1}_n^T\right)\cdots\right) + \left(\hat{b}_L'' + \delta_{bL}''\right)\mathbf{1}_n^T$$

$$=\left(\hat{W}_L'' + \delta_{WL}''\right)s_+\left(\cdots s_+\left(\left(\hat{W}_1'' + \delta_{W1}''\right)X + \left(\hat{b}_1'' + \delta_{b1}''\right)\mathbf{1}_n^T\right) + f(t)\mathbf{1}_{d_1}\mathbf{1}_n^T\cdots\right)$$

$$+ f(t)\mathbf{1}_{d_L}\mathbf{1}_n^T + \left(\hat{b}_L'' + \delta_{bL}''\right)\mathbf{1}_n^T$$

$$=M_1X + M_2\mathbf{1}_n^T$$

$$=[M_1 \quad M_2]\begin{bmatrix} X \\ \mathbf{1}_n^T \end{bmatrix}.$$

Similar to Lemma 5, $[M_1 \quad M_2]$ approaches $\tilde{W}$ as disturbances $\left\{[\delta_{Wi}]_{i=1}^L, [\delta_{bi}]_{i=1}^L\right\}$ approach $\mathbf{0}$ (element-wise). Combining that $\tilde{W}$ is a local minimizer of $f(W)$, we have

$$\hat{\mathcal{R}}\left(\left[\hat{W}_i'' + \delta_{Wi}''\right]_{i=1}^L, \left[\hat{b}_i'' + \delta_{bi}''\right]_{i=1}^L\right) = f\left([M_1 \quad M_2]\right) \geq f(\tilde{W}) = \hat{\mathcal{R}}\left(\left[\hat{W}_i''\right]_{i=1}^L, \left[\hat{b}_i''\right]_{i=1}^L\right).$$

The proof is completed.

$\square$

## Step (b). Prove the constructed local minima are spurious.

*Proof of Theorem 1.* Without loss of generality, we assume that all activations are the same.

Let $t$ be a non-differentiable point of the piece-wise linear activation function $h$ with

$$s_- = \lim_{\theta \to 0^-} h'(\theta),$$

$$s_+ = \lim_{\theta \to 0^+} h'(\theta).$$

Let $\sigma$ be a constant such that $h$ is linear in interval $(t - \sigma, t)$ and interval $(t, t + \sigma)$.

Then construct that

$$\tilde{W}_1'' = \frac{1}{M}\tilde{W}_1', \; \tilde{b}_1'' = \frac{1}{M}\tilde{b}_1' + t\mathbf{1}_{d_1},$$

$$\tilde{W}_2'' = \frac{1}{\tilde{M}}\tilde{W}_2', \; \tilde{b}_2'' = t\mathbf{1}_{d_2} - \frac{1}{\tilde{M}}h(t)\tilde{W}_2'\mathbf{1}_{d_2} + \frac{1}{M\tilde{M}}\tilde{b}_2',$$

$$\tilde{W}_i'' = \tilde{W}_i', \; \tilde{b}_i'' = -\tilde{W}_i'h(t)\mathbf{1}_{d_{i-1}} + t\mathbf{1}_{d_i} + \frac{1}{M\tilde{M}}\tilde{b}_i', \; (i = 3, 4, ..., L-1)$$

and

$$\tilde{W}_L'' = M\tilde{M}\tilde{W}_L', \; \tilde{b}_L'' = \tilde{b}_L' - M\tilde{M}\tilde{W}_L'h(t)\mathbf{1}_{L-1},$$

where $\left\{ \left[ \tilde{W}_i' \right]_{i=1}^L, \left[ \tilde{b}_i' \right]_{i=1}^L \right\}$ are constructed in Theorem 5, $M$ is a large enough positive real such that

$$\frac{1}{M} \left\| \tilde{W}_1'X + \tilde{b}_1'\mathbf{1}_n^T \right\|_F < \sigma, \tag{45}$$

and $\tilde{M}$ a large enough positive real such that

$$\frac{1}{\tilde{M}} \left\| \frac{1}{M}\tilde{Y}^{(2)} \left( \left[ \tilde{W}_i' \right]_{i=1}^L, \left[ \tilde{b}_i' \right]_{i=1}^L \right) \right\|_F < \sigma. \tag{46}$$

Then, we prove by induction that for any $i \in [2 : L-1]$, all components of $\tilde{Y}^{(i)} \left( \left[ \tilde{W}_i'' \right]_{i=1}^L, \left[ \tilde{b}_i'' \right]_{i=1}^L \right)$ are in interval $(t, t+\delta)$, and

$$Y^{(i)} \left( \left[ \tilde{W}_i'' \right]_{i=1}^L, \left[ \tilde{b}_i'' \right]_{i=1}^L \right) = h(t)\mathbf{1}_{d_i}\mathbf{1}_n^T + \frac{1}{\tilde{M}M}Y^{(i)} \left( \left[ \tilde{W}_i' \right]_{i=1}^L, \left[ \tilde{b}_i' \right]_{i=1}^L \right).$$

First,

$$\tilde{Y}^{(1)} \left( \left[ \tilde{W}_i'' \right]_{i=1}^L, \left[ \tilde{b}_i'' \right]_{i=1}^L \right) = \tilde{W}_1''X + \tilde{b}_1''\mathbf{1}_n^T = \frac{1}{M}(\tilde{W}_1'X + \tilde{b}_1'\mathbf{1}_n^T) + t\mathbf{1}_{d_1}^T\mathbf{1}_n^T. \tag{47}$$

For any $i \in [1 : d_1]$ and $j \in [1 : n]$, eq. (45) implies

$$\left| \left( \frac{1}{M}(\tilde{W}_1'X + \tilde{b}_1'\mathbf{1}_n^T) \right)_{ij} \right| \leq \frac{1}{M} \left\| \tilde{W}_1'X + \tilde{b}_1'\mathbf{1}_n^T \right\|_F < \sigma.$$

Thus,

$$\left( \frac{1}{M}(\tilde{W}_1'X + \tilde{b}_1'\mathbf{1}_n^T) + t\mathbf{1}_{d_1}^T\mathbf{1}_n^T \right)_{ij} \in (t - \sigma, t + \sigma). \tag{48}$$

Therefore, the output of the first layer after the activation is

$$Y^{(1)} \left( \left[ \tilde{W}_i'' \right]_{i=1}^L, \left[ \tilde{b}_i'' \right]_{i=1}^L \right) = h \left( \tilde{Y}^{(1)} \left( \left[ \tilde{W}_i'' \right]_{i=1}^L, \left[ \tilde{b}_i'' \right]_{i=1}^L \right) \right)$$

$$= h \left( \frac{1}{M}(\tilde{W}_1'X + \tilde{b}_1'\mathbf{1}_n^T) + t\mathbf{1}_{d_1}\mathbf{1}_n^T \right)$$

$$\overset{(*)}{=} h(t)\mathbf{1}_{d_1}\mathbf{1}_n^T + h_{s_-,s_+} \left( \frac{1}{M} \left( \tilde{W}_1'X + \tilde{b}_1' \right) \right)$$

$$= h(t)\mathbf{1}_{d_1}\mathbf{1}_n^T + \frac{1}{M}h_{s_-,s_+} \left( \left( \tilde{W}_1'X + \tilde{b}_1' \right) \right)$$

$$= h(t)\mathbf{1}_{d_1}\mathbf{1}_n^T + \frac{1}{M}Y^{(1)} \left( \left[ \tilde{W}_i' \right]_{i=1}^L, \left[ \tilde{b}_i' \right]_{i=1}^L \right),$$

where eq. $(*)$ is from eq. (44) for any $x \in (t - \delta, t + \delta)$.

Also,

$$
\begin{aligned}
\tilde{Y}^{(2)} \left( \left[ \tilde{W}_i'' \right]_{i=1}^{L}, \left[ \tilde{b}_i'' \right]_{i=1}^{L} \right) &= \tilde{W}_2'' Y^{(1)} \left( \left[ \tilde{W}_i'' \right]_{i=1}^{L}, \left[ \tilde{b}_i'' \right]_{i=1}^{L} \right) + \tilde{b}_2'' \mathbf{1}_n^T \\
&= \frac{1}{\tilde{M}} \left( \tilde{W}_2' \right) \left( h(t) \mathbf{1}_{d_1} \mathbf{1}_n^T + \frac{1}{M} Y^{(1)} \left( \left[ \tilde{W}_i' \right]_{i=1}^{L}, \left[ \tilde{b}_i' \right]_{i=1}^{L} \right) \right) \\
&\quad + t \mathbf{1}_{d_2} \mathbf{1}_n^T - \frac{1}{\tilde{M}} h(t) \tilde{W}_2' \mathbf{1}_{d_1} \mathbf{1}_n^T + \frac{1}{M\tilde{M}} \tilde{b}_2' \mathbf{1}_n^T \\
&= \frac{1}{\tilde{M} M} \tilde{W}_2' Y^{(1)} \left( \left[ \tilde{W}_i' \right]_{i=1}^{L}, \left[ \tilde{b}_i' \right]_{i=1}^{L} \right) + \frac{1}{M\tilde{M}} \tilde{b}_2' \mathbf{1}_n^T + t \mathbf{1}_{d_2} \mathbf{1}_n^T \\
&= \frac{1}{M\tilde{M}} \tilde{Y}^{(2)} \left( \left[ \tilde{W}_i' \right]_{i=1}^{L}, \left[ \tilde{b}_i' \right]_{i=1}^{L} \right) + t \mathbf{1}_{d_2} \mathbf{1}_n^T .
\end{aligned}
$$

Recall in Theorem 5 we prove all components of $\tilde{Y}^{(2)} \left( \left[ \tilde{W}_i' \right]_{i=1}^{L}, \left[ \tilde{b}_i' \right]_{i=1}^{L} \right)$ are positive. Combining the definition of $\tilde{M}$ (eq. (46)), we have

$$
\begin{aligned}
t \mathbf{1}_{d_2} \mathbf{1}_n^T &< \tilde{Y}^{(2)} \left( \left[ \tilde{W}_i'' \right]_{i=1}^{L}, \left[ \tilde{b}_i'' \right]_{i=1}^{L} \right) \\
&= \frac{1}{\tilde{M} M} \tilde{Y}^{(2)} \left( \left[ \tilde{W}_i' \right]_{i=1}^{L}, \left[ \tilde{b}_i' \right]_{i=1}^{L} \right) + t \mathbf{1}_{d_2} \mathbf{1}_n^T \\
&< (t + \sigma) \mathbf{1}_{d_2} \mathbf{1}_n^T .
\end{aligned}
$$

Therefore,

$$
\begin{aligned}
Y^{(2)} \left( \left[ \tilde{W}_i'' \right]_{i=1}^{L}, \left[ \tilde{b}_i'' \right]_{i=1}^{L} \right) &= h \left( \tilde{Y}^{(2)} \left( \left[ \tilde{W}_i'' \right]_{i=1}^{L}, \left[ \tilde{b}_i'' \right]_{i=1}^{L} \right) \right) \\
&= h \left( \frac{1}{\tilde{M} M} Y^{(2)} \left( \left[ \tilde{W}_i' \right]_{i=1}^{L}, \left[ \tilde{b}_i' \right]_{i=1}^{L} \right) + t \mathbf{1}_{d_2} \mathbf{1}_n^T \right) \\
&= h(t) \mathbf{1}_{d_2} \mathbf{1}_n^T + h_{s_-, s_+} \left( \frac{1}{\tilde{M} M} \tilde{Y}^{(2)} \left( \left[ \tilde{W}_i' \right]_{i=1}^{L}, \left[ \tilde{b}_i' \right]_{i=1}^{L} \right) \right) \\
&= h(t) \mathbf{1}_{d_2} \mathbf{1}_n^T + \frac{1}{\tilde{M} M} h_{s_-, s_+} \left( \tilde{Y}^{(2)} \left( \left[ \tilde{W}_i' \right]_{i=1}^{L}, \left[ \tilde{b}_i' \right]_{i=1}^{L} \right) \right) \\
&= h(t) \mathbf{1}_{d_2} \mathbf{1}_n^T + \frac{1}{\tilde{M} M} Y^{(2)} \left( \left[ \tilde{W}_i' \right]_{i=1}^{L}, \left[ \tilde{b}_i' \right]_{i=1}^{L} \right) .
\end{aligned}
$$

Suppose the above argument holds for $k$-th layer.

The output of $(k + 1)$-th layer before the activation is

$$
\begin{aligned}
&\tilde{Y}^{(k+1)} \left( \left[ \tilde{W}_i'' \right]_{i=1}^{L}, \left[ \tilde{b}_i'' \right]_{i=1}^{L} \right) \\
&= \tilde{W}_{k+1}'' Y^{(k)} \left( \left[ \tilde{W}_i'' \right]_{i=1}^{L}, \left[ \tilde{b}_i'' \right]_{i=1}^{L} \right) + \tilde{b}_{k+1}'' \mathbf{1}_n^T \\
&= \tilde{W}_{k+1}' \left( h(t) \mathbf{1}_{d_k} \mathbf{1}_n^T + \frac{1}{\tilde{M} M} Y^{(k)} \left( \left[ \tilde{W}_i' \right]_{i=1}^{L}, \left[ \tilde{b}_i' \right]_{i=1}^{L} \right) \right) \\
&\quad + \left( -\tilde{W}_{k+1}' h(t) \mathbf{1}_{d_k} + t \mathbf{1}_{d_{k+1}} + \frac{1}{M\tilde{M}} \tilde{b}_{k+1}' \right) \mathbf{1}_n^T \\
&= \frac{1}{M\tilde{M}} \left( \tilde{W}_{k+1}' Y^{(k)} \left( \left[ \tilde{W}_i' \right]_{i=1}^{L}, \left[ \tilde{b}_i' \right]_{i=1}^{L} \right) + \tilde{b}_{k+1}' \mathbf{1}_n^T \right) + t \mathbf{1}_{d_{k+1}} \mathbf{1}_n^T \\
&= \frac{1}{M\tilde{M}} \tilde{Y}^{(k+1)} \left( \left[ \tilde{W}_i' \right]_{i=1}^{L}, \left[ \tilde{b}_i' \right]_{i=1}^{L} \right) + t \mathbf{1}_{d_{k+1}} \mathbf{1}_n^T .
\end{aligned}
$$

Recall proved in Theorem 5 that all components of $\tilde{Y}^{(k+1)}\left(\left[\tilde{W}_i'\right]_{i=1}^L, \left[\tilde{b}_i'\right]_{i=1}^L\right)$ except those that are 0 are contained in $\tilde{Y}^{(k)}\left(\left[\tilde{W}_i'\right]_{i=1}^L, \left[\tilde{b}_i'\right]_{i=1}^L\right)$. We have

$$t\mathbf{1}_{d_{k+1}}\mathbf{1}_n^T < \frac{1}{M\tilde{M}}\tilde{Y}^{(k+1)}\left(\left[\tilde{W}_i'\right]_{i=1}^L, \left[\tilde{b}_i'\right]_{i=1}^L\right) + t\mathbf{1}_{d_{k+1}}\mathbf{1}_n^T < (t+\sigma)\mathbf{1}_{d_{k+1}}\mathbf{1}_n^T.$$

Therefore,

$$\begin{aligned}
Y^{(k+1)}\left(\left[\tilde{W}_i''\right]_{i=1}^L, \left[\tilde{b}_i''\right]_{i=1}^L\right) &= h\left(\tilde{Y}^{(k)}\left(\left[\tilde{W}_i''\right]_{i=1}^L, \left[\tilde{b}_i''\right]_{i=1}^L\right)\right) \\
&= h\left(\frac{1}{M\tilde{M}}\tilde{Y}^{(k+1)}\left(\left[\tilde{W}_i'\right]_{i=1}^L, \left[\tilde{b}_i'\right]_{i=1}^L\right) + t\mathbf{1}_{d_{k+1}}\mathbf{1}_n^T\right) \\
&= h(t)\mathbf{1}_{d_{k+1}}\mathbf{1}_n^T + \frac{1}{M\tilde{M}}Y^{(k+1)}\left(\left[\tilde{W}_i'\right]_{i=1}^L, \left[\tilde{b}_i'\right]_{i=1}^L\right).
\end{aligned}$$

Thus, the argument holds for any $k \in \{2, \ldots, L-1\}$.

So,

$$\begin{aligned}
Y^{(L)}\left(\left[\tilde{W}_i''\right]_{i=1}^L, \left[\tilde{b}_i''\right]_{i=1}^L\right) &= \tilde{W}_L'' Y^{(L-1)}\left(\left[\tilde{W}_i''\right]_{i=1}^L, \left[\tilde{b}_i''\right]_{i=1}^L\right) + \tilde{b}_L'' \\
&= M\tilde{M}\tilde{W}_L'\left(h(t)\mathbf{1}_{d_{L-1}}\mathbf{1}_n^T + \frac{1}{M\tilde{M}}Y^{(L-1)}\left(\left[\tilde{W}_i'\right]_{i=1}^L, \left[\tilde{b}_i'\right]_{i=1}^L\right)\right) \\
&\quad + \tilde{b}_L'\mathbf{1}_n^T - M\tilde{M}\tilde{W}_L'h(t)\mathbf{1}_{d_{L-1}}\mathbf{1}_n^T \\
&= Y^{(L)}\left(\left[\tilde{W}_i'\right]_{i=1}^L, \left[\tilde{b}_i'\right]_{i=1}^L\right).
\end{aligned}$$

Therefore,

$$\hat{\mathcal{R}}\left(\left[\tilde{W}_i''\right]_{i=1}^L, \left[\tilde{b}_i''\right]_{i=1}^L\right) = \hat{\mathcal{R}}\left(\left[\tilde{W}_i'\right]_{i=1}^L, \left[\tilde{b}_i'\right]_{i=1}^L\right). \tag{49}$$

From eq. (49) and Theorem 5, we have

$$\hat{\mathcal{R}}\left(\left[\tilde{W}_i''\right]_{i=1}^L, \left[\tilde{b}_i''\right]_{i=1}^L\right) < f\left(\tilde{W}\right),$$

which completes the proof of local minimizer.

Furthermore, the parameter $M$ used in Lemma 6 (not those in this proof) is arbitrary in a continuous interval (cf. eq. (40)), we have actually constructed infinite spurious local minima.

$\square$

Theorem 1 relies on Assumption 4. We can further remove it by replacing Assumption 3 by a mildly more restrictive variant Assumption 5.

**Corollary 3.** *Suppose that Assumptions 1, 2, and 5 hold. Neural networks with arbitrary depth and arbitrary piecewise linear activations (excluding linear functions) have infinitely many spurious local minima under arbitrary continuously differentiable loss whose derivative can equal 0 only when the prediction and label are the same.*

*Proof.* The proof is delivered by modifications of Theorem 4 in Stage 1 of Theorem 1's proof. We only need to prove the corollary under the assumption that $s_- + s_+ = 0$.

Let the local minimizer constructed in Lemma 4 be $\left\{\left[\hat{W}_i\right]_{i=1}^2, \left[\hat{b}_i\right]_{i=1}^2\right\}$. Then, we construct a point in the parameter space whose empirical risk is smaller as follows:

$$\tilde{W}_1 = \begin{bmatrix} \tilde{W}_{1,[1:d_X]} - \alpha\beta^T \\ \tilde{W}_{1,[1:d_X]} \\ -\tilde{W}_{1,[1:d_X]} + \alpha\beta^T \\ \tilde{W}_{2,[1:d_X]} \\ \vdots \\ \tilde{W}_{d_Y,[1:d_X]} \\ 0_{(d_1-d_Y-2)\times d_X} \end{bmatrix},$$

$$\tilde{b}_1 = \begin{bmatrix} \tilde{W}_{1,[d_X+1]} - \eta_1 + \gamma \\ \tilde{W}_{1,[d_X+1]} - \eta \\ -\tilde{W}_{1,[d_X+1]} + \eta_1 + \gamma \\ \tilde{W}_{2,[d_X+1]} - \eta_2 \\ \vdots \\ \tilde{W}_{d_Y,[d_X+1]} - \eta_{d_Y} \\ 0_{(d_1-d_Y-2)\times 1} \end{bmatrix},$$

$$\tilde{W}_2 = \begin{bmatrix} \frac{1}{2s_+} & \frac{1}{s_+} & -\frac{1}{2s_+} & 0 & 0 & \cdots & 0 & 0 & \cdots & 0 \\ 0 & 0 & 0 & \frac{1}{s_+} & 0 & \cdots & 0 & 0 & \cdots & 0 \\ 0 & 0 & 0 & 0 & \frac{1}{s_+} & \cdots & 0 & 0 & \cdots & 0 \\ \vdots & \vdots & \vdots & \vdots & \vdots & \ddots & \vdots & \vdots & \ddots & \vdots \\ 0 & 0 & 0 & 0 & 0 & \cdots & \frac{1}{s_+} & 0 & \cdots & 0 \end{bmatrix},$$

and

$$\tilde{b}_2 = \begin{bmatrix} \eta \\ \eta_2 \\ \vdots \\ \eta_{d_Y} \end{bmatrix},$$

where $\alpha$, $\beta$, and $\eta_i$ are defined the same as those in Theorem 4, and $\eta$ is defined by eq. (27). Then, the output of the first layer is

$$
\begin{aligned}
Y^{(1)}\left(\left[\tilde{W}_i\right]_{i=1}^2, \left[\tilde{b}_i\right]_{i=1}^2\right) &= h_{s_-,s_+}\left(\tilde{W}_1 X + \tilde{b}_1 \mathbf{1}_n^T\right) \\
&= h_{s_-,s_+}\left(\begin{bmatrix} \tilde{W}_{1,.} X - \alpha\beta^T X - \eta_1 \mathbf{1}_n^T + \gamma \mathbf{1}_n^T \\ \tilde{W}_{1,.} X - \eta \mathbf{1}_n^T \\ -\tilde{W}_{1,.} X + \alpha\beta^T X + \eta_1 \mathbf{1}_n^T + \gamma \mathbf{1}_n^T \\ \tilde{W}_{2,.} X - \eta_2 \mathbf{1}_n^T \\ \vdots \\ \tilde{W}_{d_Y,.} X - \eta_{d_Y} \mathbf{1}_n^T \\ 0_{d_1-d_Y-2} \mathbf{1}_n^T \end{bmatrix}\right).
\end{aligned}
$$

Further, the output of the whole network is

$$\hat{Y}\left(\left[\tilde{W}_i\right]_{i=1}^2, \left[\tilde{b}_i\right]_{i=1}^2\right)$$

$$=\tilde{W}_2 h_{s_-,s_+}\left(\begin{bmatrix} \tilde{W}_{1,.}X - \alpha\beta^T X - \eta_1 \mathbf{1}_n^T + \gamma\mathbf{1}_n^T \\ \tilde{W}_{1,.}X - \eta\mathbf{1}_n^T \\ -\tilde{W}_{1,.}X + \alpha\beta^T X + \eta_1\mathbf{1}_n^T + \gamma\mathbf{1}_n^T \\ \tilde{W}_{2,.}X - \eta_2\mathbf{1}_n^T \\ \vdots \\ \tilde{W}_{d_Y,.}X - \eta_{d_Y}\mathbf{1}_n^T \\ \mathbf{0}_{d_1-d_Y-2}\mathbf{1}_n^T \end{bmatrix}\right) + \tilde{b}_2\mathbf{1}_n^T$$

$$=\tilde{W}_2 h_{s_-,s_+}\left(\begin{bmatrix} \tilde{W}_{1,.}X - \alpha\beta^T X - \eta_1 \mathbf{1}_n^T + \gamma\mathbf{1}_n^T \\ \tilde{W}_{1,.}X - \eta\mathbf{1}_n^T \\ -\tilde{W}_{1,.}X + \alpha\beta^T X + \eta_1\mathbf{1}_n^T + \gamma\mathbf{1}_n^T \\ \tilde{W}_{2,.}X - \eta_2\mathbf{1}_n^T \\ \vdots \\ \tilde{W}_{d_Y,.}X - \eta_{d_Y}\mathbf{1}_n^T \\ \mathbf{0}_{d_1-d_Y-2}\mathbf{1}_n^T \end{bmatrix}\right) + \begin{bmatrix} \eta \\ \eta_2 \\ \vdots \\ \eta_{d_Y} \end{bmatrix}\mathbf{1}_n^T.$$

Therefore, if $j \le l'$, the $(1,j)$-th component of $\hat{Y}\left(\left[\tilde{W}_i\right]_{i=1}^2, \left[\tilde{b}_i\right]_{i=1}^2\right)$ is

$$\left(\tilde{W}_2\right)_1 \tilde{Y}^{(1)}\left(\left[\tilde{W}_i\right]_{i=1}^2, \left[\tilde{b}_i\right]_{i=1}^2\right)_j$$

$$=\frac{1}{2s_+}\left(s_-\left(\tilde{Y}_{1,j} - \alpha\beta^T x_j - \eta_1 + \gamma\right) + 2s_+\left(\tilde{Y}_{1,j} - \eta\right) - s_+\left(-\tilde{Y}_{1j} + \alpha\beta^T x_j + \eta_1 + \gamma\right)\right)$$

$$+ \eta$$

$$=\frac{1}{2s_+}\left(-s_+\left(\tilde{Y}_{1,j} - \alpha\beta^T x_j - \eta_1 + \gamma\right) + 2s_+\left(\tilde{Y}_{1,j} - \eta\right) - s_+\left(-\tilde{Y}_{1j} + \alpha\beta^T x_j + \eta_1 + \gamma\right)\right)$$

$$+ \eta$$

$$=\frac{1}{2s_+}\left(2s_+\tilde{Y}_{1,j} - 2s_+\eta - 2s_+\gamma\right) + \eta$$

$$=\tilde{Y}_{1,j} - \gamma.$$

Otherwise ($j > l'$), the $(1,j)$-th component of $\hat{Y}\left(\left[\tilde{W}_i\right]_{i=1}^2, \left[\tilde{b}_i\right]_{i=1}^2\right)$ is

$$\left(\tilde{W}_2\right)_1 \tilde{Y}^{(1)}\left(\left[\tilde{W}_i\right]_{i=1}^2, \left[\tilde{b}_i\right]_{i=1}^2\right)_j$$

$$=\frac{1}{2s_+}\left(s_+\left(\tilde{Y}_{1,j} - \alpha\beta^T x_j - \eta_1 + \gamma\right) + 2s_+\left(\tilde{Y}_{1,j} - \eta\right) - s_-\left(-\tilde{Y}_{1j} + \alpha\beta^T x_j + \eta_1 + \gamma\right)\right)$$

$$+ \eta$$

$$=\frac{1}{2s_+}\left(s_+\left(\tilde{Y}_{1,j} - \alpha\beta^T x_j - \eta_1 + \gamma\right) + 2s_+\left(\tilde{Y}_{1,j} - \eta\right) + s_+\left(-\tilde{Y}_{1j} + \alpha\beta^T x_j + \eta_1 + \gamma\right)\right) + \eta$$

$$=\frac{1}{2s_+}\left(2s_+\tilde{Y}_{1,j} - 2s_+\eta + 2s_+\gamma\right) + \eta$$

$$=\tilde{Y}_{1,j} + \gamma,$$

and the $(i,j)$-th ($i > 1$) component of $\hat{Y}\left(\left[\tilde{W}_i\right]_{i=1}^2, \left[\tilde{b}_i\right]_{i=1}^2\right)$ is $\tilde{Y}_{i,j}$.

Therefore, we have

$$\left(\tilde{W}_2\left(\tilde{W}_1 x_i + \tilde{b}_1\right) + \tilde{b}_2 - \tilde{W}\begin{bmatrix} x_i \\ 1 \end{bmatrix}\right)_j = \begin{cases} -\gamma, & j = 1, i \le l; \\ \gamma, & j = 1, i > l; \\ 0, & j \ge 2. \end{cases}$$

Then, similar to Theorem 4, we have

$$\hat{\mathcal{R}}\left(\left[\tilde{W}_i\right]_{i=1}^L, \left[\tilde{b}_i\right]_{i=1}^L\right) - \hat{\mathcal{R}}\left(\left[\hat{W}_i\right]_{i=1}^L, \left[\hat{b}_i\right]_{i=1}^L\right)$$

$$= \frac{1}{n}\sum_{i=1}^n l\left(Y_i, \tilde{W}_2\left(\tilde{W}_1 x_i + \tilde{b}_1\right) + \tilde{b}_2\right) - \frac{1}{n}\sum_{i=1}^n l\left(Y_i, \hat{W}\begin{bmatrix} x_i \\ 1 \end{bmatrix}\right)$$

$$= \frac{1}{n}\sum_{i=1}^n \nabla_{\hat{Y}_i} l\left(Y_i, \tilde{W}\begin{bmatrix} x_i \\ 1 \end{bmatrix}\right)\left(\tilde{W}_2\left(\tilde{W}_1 x_i + \tilde{b}_1 \mathbf{1}_n^T\right) + \tilde{b}_2 \mathbf{1}_n^T - \tilde{W}\begin{bmatrix} x_i \\ 1 \end{bmatrix}\right)$$

$$+ \sum_{i=1}^n o\left(\left\|\tilde{W}_2\left(\tilde{W}_1 x_i + \tilde{b}_1 \mathbf{1}_n^T\right) + \tilde{b}_2 \mathbf{1}_n^T - \tilde{W}\begin{bmatrix} x_i \\ 1 \end{bmatrix}\right\|\right)$$

$$= -\frac{2}{n}\sum_{i=1}^{l'} \boldsymbol{V}_{1,i}\gamma + o(\gamma),$$

where $\boldsymbol{V}$ and $l'$ are also defined the same as those in Theorem 4.

When $\gamma$ is sufficiently small and $\operatorname{sgn}(\gamma) = \operatorname{sgn}\left(\sum_{i=1}^{l'} \boldsymbol{V}_{1,i}\right)$, we have that

$$\hat{\mathcal{R}}\left(\left(\left[\tilde{W}_i\right]_{i=1}^2, \left[\tilde{b}_i\right]_{i=1}^2\right)\right) < f(\tilde{W}).$$

This complete the proof of Corollary 3. $\qquad\square$

## A.5  A PREPARATION LEMMA

We now prove the preparation lemma used above.

**Lemma 7.** *Suppose $\boldsymbol{u} = (u_1 \cdots u_n) \in \mathbb{R}^{1 \times n}$ which satisfies $\boldsymbol{u} \ne \boldsymbol{0}$ and*

$$\sum_{i=1}^n u_i = 0, \tag{50}$$

*while $\{x_1, ..., x_n\}$ is a set of vector $\subset \mathbb{R}^{m \times 1}$. Suppose index set $S = \{1, 2, \cdots, n\}$. Then for any series of real number $\{v_1, \cdots, v_n\}$, there exists a non-empty separation I, J of S, which satisfies $I \cup J = S$, $I \cap J = \emptyset$ and both I and J are not empty, a vector $\beta \in \mathbb{R}^{m \times 1}$, such that,*

*(1.1) for any sufficiently small positive real $\alpha$, $i \in I$, and $j \in J$, we have $v_i - \alpha\beta^T x_i < v_j - \alpha\beta^T x_j$;*

*(1.2) $\sum_{i \in I} u_i \ne 0$.*

*Proof.* If there exists a non-empty separation $I$ and $J$ of the index set $S$, such that when $\beta = 0$, (1.1) and (1.2) hold, the lemma is apparently correct.

Otherwise, suppose that there is no non-empty separation $I$ and $J$ of the index set $S$ such that (1.1) and (1.2) hold simultaneously when $\beta = 0$.

Some number $v_i$ in the sequence $(v_1, v_2, \cdots, v_n)$ are probably equal to each other. We rearrange the sequence by the increasing order as follows,

$$v_1 = v_2 = \cdots = v_{s_1} < v_{s_1+1} = \cdots = v_{s_2} < \cdots < v_{s_{k-1}+1} = \cdots = v_{s_k} = v_n, \tag{51}$$

where $s_k = n$.

Then, for any $j \in \{1, 2, \cdots, k-1\}$, we argue that

$$\sum_{i=1}^{s_j} u_i = 0.$$

Otherwise, suppose there exists a $s_j$, such that

$$\sum_{i=1}^{s_j} u_i \neq 0.$$

Let $I = \{1, 2, ..., s_j\}$ and $J = \{s_j + 1, ..., n\}$. Then, when $\beta = 0$, we have

$$v_i - \alpha \beta^T x_i = v_i < v_j = v_j - \alpha \beta^T x_j,$$

and

$$\sum_{i \in I} u_i = \sum_{i=1}^{s_j} u_i \neq 0,$$

which are exactly the arguments (1.1) and (1.2). Thereby we construct a contrary example. Therefore, for any $j \in \{1, 2, \cdots, k-1\}$, we have

$$\sum_{i=1}^{s_j} u_i = 0.$$

Since we assume that $\mathbf{u} \neq 0$, there exists an index $t \in \{1, \ldots, k-1\}$, such that there exists an index $i \in \{s_t + 1, ..., s_{t+1}\}$ that $u_i \neq 0$.

Let $l \in \{s_t + 1, ..., s_{t+1}\}$ is the index such that $x_l$ has the largest norm while $u_l \neq 0$:

$$l = \underset{j \in \{s_t+1,...,s_{t+1}\},\, u_j \neq 0}{\arg\max} \|x_j\|. \tag{52}$$

We further rearrange the sequence $(v_{s_t+1}, ..., v_{s_{t+1}})$ such that there is an index $l' \in \{s_t + 1, \ldots, s_{t+1}\}$,

$$\|x_{l'}\| = \max_{j \in \{s_t+1,...,s_{t+1}\},\, u_j \neq 0} \|x_j\|,$$

and

$$\forall i \in \{s_t + 1, ..., l'\}, \ \langle x_{l'}, x_i \rangle \geq \|x_{l'}\|^2; \tag{53}$$

$$\forall i \in \{l' + 1, \cdots, s_{t+1}\}, \ \langle x_{l'}, x_i \rangle < \|x_{l'}\|^2. \tag{54}$$

It is worth noting that it is probably $l' = s_{t+1}$, but it is a trivial case that would not influence the result of this lemma.

Let $I = \{1, ..., l'\}$, $J = \{l' + 1, ..., n\}$, and $\beta = x_{l'}$. We prove (1.1) and (1.2) as follows.

**Proof of argument (1.1).**

We argue that for any $i \in I$, $v_i - \alpha \beta^T x_i \leq v_{l'} - \alpha \beta^T x_{l'}$ and for any $j \in J$, $v_j - \alpha \beta^T x_j > v_{l'} - \alpha \beta^T x_{l'}$.

There are three situations:

(A) $i \in \{1, \ldots, s_t\}$ and $j \in \{s_{t+1} + 1, \cdots, n\}$. Applying eq. (51), for any $i \in \{1, \ldots, s_t\}$ and $j \in \{s_{t+1} + 1, \cdots, n\}$, we have that $v_i < v_{l'}$ and $v_j > v_{l'}$. Therefore, when $\alpha$ is sufficiently small, we have the following inequalities,

$$v_i - \alpha \beta^T x_i < v_{l'} - \alpha \beta^T x_{l'},$$
$$v_j - \alpha \beta^T x_j > v_{l'} - \alpha \beta^T x_{l'}.$$

(B) $i \in \{s_t + 1, \cdots, l'\}$. Applying eq. (53) and because of $\alpha > 0$, we have

$$-\alpha \langle \beta, x_i \rangle \leq -\alpha \|\beta\|^2 = -\alpha \langle \beta, x_{l'} \rangle.$$

Since $v_i = v_{l'}$, it further leads to

$$v_i - \alpha\beta^T x_i \leq v_{l'} - \alpha\beta^T x_{l'}.$$

(C) $j \in \{l'+1, \cdots, s_{t+1}\}$. Similarly, applying eq. (54) and because of $\alpha > 0$, we have

$$-\alpha\langle\beta, x_j\rangle > -\alpha\|\beta\|^2 = -\alpha\langle\beta, x_{l'}\rangle.$$

Since $v_j = v_{l'}$, it further leads to

$$v_j - \alpha\beta^T x_j > v_{l'} - \alpha\beta^T x_{l'},$$

which is exactly the argument (1.1).

**Proof of argument (1.2).**

We argue that for any $i \in \{s_t + 1, \cdots, l'-1\}$, $u_i = 0$. Otherwise, suppose there exists an $i \in \{s_t + 1, \cdots, l'-1\}$ such that $u_i \neq 0$. From eq. (52), we have $\|x_i\| \leq \|x_{l'}\|$. Therefore,

$$\langle x_{l'}, x_i\rangle \leq \|x_{l'}\|\|x_i\| \leq \|x_{l'}\|^2,$$

where the first inequality strictly holds if the vector $x_{l'}$ and $x_i$ have the same direction, while the second inequlity strictly holds when $x_i$ and $x_{l'}$ have the same norm. Because $x_{l'} \neq x_i$, we have the following inequality,

$$\langle x_{l'}, x_i\rangle < \|x_{l'}\|^2,$$

which contradicts to eq. (53), i.e.,

$$\langle x_{l'}, x_i\rangle \geq \|x_{l'}\|^2, \ \forall i \in \{s_t + 1, \cdots, l'\}.$$

Therefore,

$$\sum_{i \in I} u_i = \sum_{i=1}^{s_t} u_i + \sum_{i=s_t+1}^{l'-1} u_i + u_{l'} = u_{l'} \neq 0,$$

which is exactly the argument (1.2).

The proof is completed. □

**Remark.** *For any $i \in \{l'+1, ..., s_{t+1}\}$, we have*

$$\left(v_i - \alpha\beta^T x_i\right) - \left(v_{l'} - \alpha\beta^T x_{l'}\right) = \alpha\beta^T(x_{l'} - x_i), \tag{55}$$

*while for any $j \in \{s_{t+1}+1, ..., n\}$, we have*

$$\left(v_j - \alpha\beta^T x_j\right) - \left(v_{l'} - \alpha\beta^T x_{l'}\right) = v_j - v_{l'} + \alpha\beta^T(x_{l'} - x_j). \tag{56}$$

*Because $v_j > v_{l'}$, when the real number $\alpha$ is sufficiently small, we have*

$$\alpha\beta^T(x_{l'} - x_i) < v_j - v_{l'} + \alpha\beta^T(x_{l'} - x_j).$$

*Applying eqs. (55) and (56), we have*

$$\left(v_i - \alpha\beta^T x_i\right) - \left(v_{l'} - \alpha\beta^T x_{l'}\right) < \left(v_j - \alpha\beta^T x_j\right) - \left(v_{l'} - \alpha\beta^T x_{l'}\right).$$

*Therefore, if $l' < s_{t+1}$, we have*

$$\min_{i \in \{l'+1,...,n\}} \left(v_i - \alpha\beta^T x_i\right) - \left(v_{l'} - \alpha\beta^T x_{l'}\right) = \min_{i \in \{l'+1,...,s_{t+1}\}} \alpha\beta^T(x_{l'} - x_i); \tag{57}$$

*while if $l' = s_{t+1}$,*

$$\min_{i \in \{l'+1,...,n\}} \left(v_i - \alpha\beta^T x_i\right) - \left(v_{l'} - \alpha\beta^T x_{l'}\right) = \min_{i \in \{l'+1,...,n\}} v_i - v_l + \alpha\beta^T(x_{l'} - x_i). \tag{58}$$

*Eqs. (57) and (58) make senses because $l' < n$. Otherwise, from Lemma 7 we have $\sum_{i=1}^n u_i \neq 0$, which contradicts to the assumption.*

## B PROOFS OF THEOREM 2, THEOREM 3, COROLLARY 1, AND COROLLARY 2

This appendix gives the proofs of Theorem 2, Theorem 3, Corollary 1, and Corollary 2 omitted from Section 4.

### B.1 SQUARED LOSS

We first check that the squared loss is strictly convex, which is even restrictive than "convex".

**Lemma 8.** *The empirical risk $\hat{\mathcal{R}}$ under squared loss (defined by eq. (5)) is strictly convex with respect to the prediction $\hat{Y}$.*

*Proof.* The second derivative of the empirical risk $\hat{\mathcal{R}}$ under squared loss with respect to the prediction $\hat{Y}$ is

$$\frac{\partial^2 l_{ce}(Y, \hat{Y})}{\partial \hat{Y}^2} = \frac{\partial^2 (y - \hat{Y})^2}{\partial \hat{Y}^2} = 2 > 0.$$

Therefore, the empirical risk $\hat{\mathcal{R}}$ under squared loss is strictly convex with respect to prediction $\hat{Y}$. $\square$

### B.2 SMOOTH AND MULTILINEAR PARTITION.

If the activations are all linear functions, the neural networks is reduced to a multilinear model. The loss surface is apparently smooth and multilinear. The nonlinearity in the activations largely reshape the landscape of the loss surface. Specifically, if the input data flows through the linear parts of every activation functions, the output falls in a smooth and multilinear region in the loss surface. When some parameter changes by a sufficiently small swift, the data flow may not move out of the linear parts of the activations. This fact guarantees that each smooth and multilinear regions expands to an open cell. Meanwhile, every nonlinear point in the activations is non-differentiable. If the input data flows through these nonlinear points, the corresponding empirical risk is not smooth with respect to the parameters. Therefore, the nonlinear points in activations correspond to the non-differentiable boundaries between cells on the loss surface.

### B.3 EVERY LOCAL MINIMUM IS GLOBALLY MINIMAL WITHIN A CELL.

*Proof of Theorem 2.* In every cell, the input sample points flows through the same linear parts of the activations no matter what values the parameters are.

(1) We first proves that the empirical risk $\hat{\mathcal{R}}$ equals to a convex function with respect to a variable $\hat{W}$ that is calculated from the parameters $W$.

Suppose $(W_1, W_2)$ is a local minimum within a cell. We argue that

$$\sum_{i=1}^{n} l\left(y_i, W_2 \text{diag}\left(A_{\cdot,i}\right) W_1 x_i\right) = \sum_{i=1}^{n} l\left(y_i, A_{\cdot,i}^T \text{diag}(W_2) W_1 x_i\right), \tag{59}$$

where $A_{\cdot,i}$ is the $i$-th column of the following matrix

$$A = \begin{bmatrix} h'_{s_-, s_+}((W_1)_{1,\cdot} x_1) & \cdots & h'_{s_-, s_+}((W_1)_{1,\cdot} x_n) \\ \vdots & \ddots & \vdots \\ h'_{s_-, s_+}((W_1)_{d_1,\cdot} x_1) & \cdots & h'_{s_-, s_+}((W_1)_{d_1,\cdot} x_n) \end{bmatrix}. \tag{60}$$

The left-hand side (LHS) is as follows,

$$\begin{aligned} \text{LHS} &= \sum_{i=1}^{n} l\left(y_i, W_2 \text{diag}\left(A_{\cdot,i}\right) W_1 x_i\right) \\ &= \sum_{i=1}^{n} l\left(y_i, [(W_2)_{1,1} A_{1,i} \quad \cdots \quad (W_2)_{1,d_1} A_{d_1,i}] W_1 x_i\right). \end{aligned}$$

Meanwhile, the right-hand side (RHS) is as follows,

$$\text{RHS} = \sum_{i=1}^{n} l\left(y_i, A_{\cdot,i}^T \text{diag}(W_2)W_1 x_i\right),$$

$$= \sum_{i=1}^{n} l\left(y_i, [(W_2)_{1,1} A_{1,i} \quad \cdots \quad (W_2)_{1,d_1} A_{d_1,i}] W_1 x_i\right).$$

Apparently, LHS = RHS. Thereby, we proved eq. (59).

Afterwards, we define

$$\hat{W}_1 = \text{diag}(W_2)W_1, \tag{61}$$

and then straighten the matrix $\hat{W}_1$ to a vector $\hat{W}$,

$$\hat{W} = \left((\hat{W}_1)_{1,\cdot} \quad \cdots \quad (\hat{W}_1)_{d_1,\cdot}\right), \tag{62}$$

Also define

$$\hat{X} = (A_{\cdot,1} \otimes x_1 \quad \cdots \quad A_{\cdot,n} \otimes x_n). \tag{63}$$

Then, we can prove that the following equations,

$$\left(A_{\cdot,1}^T \hat{W}_1 x_1 \quad \cdots \quad A_{\cdot,n}^T \hat{W}_1 x_n\right) = \left((\hat{W}_1)_{1,\cdot} \quad \cdots \quad (\hat{W}_1)_{d_1,\cdot}\right)(A_{\cdot,1} \otimes x_1 \quad \cdots \quad A_{\cdot,n} \otimes x_n)$$

$$= \hat{W}\hat{X}. \tag{64}$$

Applying eq. (64), the empirical risk is transferred to a convex function as follows,

$$\hat{\mathcal{R}}(W_1, W_2) = \frac{1}{n} \sum_{i=1}^{n} l\left(y_i, (A_{\cdot,i})^T \text{diag}(W_2)W_1 x_i\right) = \frac{1}{n} \sum_{i=1}^{n} l\left(y_i, (A_{\cdot,i})^T \hat{W}_1 x_i\right)$$

$$= \frac{1}{n} \sum_{i=1}^{n} l\left(y_i, \hat{W}\hat{X}_i\right). \tag{65}$$

We can see that the empirical risk is rearranged as a convex function in terms of $\hat{W}$ which unite the two weight matrices $W_1$ and $W_2$ and the activation $h$ are together as $\hat{W}$.

Applying eqs. (61) and (62), we have

$$\hat{W} = [(W_2)_1 (W_1)_{1,\cdot} \quad \cdots \quad (W_2)_{d_1} (W_1)_{d_1,\cdot}].$$

(2) We then prove that the local minima (including global minima) of the empirical risk $\hat{\mathcal{R}}$ with respect to the parameter $W$ is also local minima with respect to the corresponding variable $\hat{W}$.

We first prove that for any $i \in [1 : d_1 d_2]$, we have

$$e_i \hat{X} \nabla = 0,$$

where $\nabla$ is defined as follows,

$$\nabla = \left[\nabla_{(\hat{W}\hat{X})_1} l\left(Y_1, \left(\hat{W}\hat{X}\right)_1\right) \quad \cdots \quad \nabla_{(\hat{W}\hat{X})_n} l\left(Y_n, \left(\hat{W}\hat{X}\right)_n\right)\right]^T.$$

To see this, we divide $i$ into two cases: $(W_2)_i \neq 0$ and $(W_2)_i = 0$.

**Case 1: $(\mathbf{W_2})_i \neq 0$.**

The local minimizer of the empirical risk $\hat{\mathcal{R}}$ with respect to the parameter $W$ satisfies the following equation,

$$\frac{\partial \hat{\mathcal{R}}}{\partial (W_1)_{i,j}} = 0.$$

Therefore,

$$
0 = \frac{\partial \hat{\mathcal{R}}}{\partial (W_1)_{i,j}}
$$

$$
= \frac{\partial \left( \sum\limits_{k=1}^{n} l\left( Y_{\cdot,k}, \left( \hat{W}\hat{X} \right)_{\cdot,k} \right) \right)}{\partial (W_1)_{i,j}}
$$

$$
= \sum_{k=1}^{n} [\; 0 \;\; \underbrace{\cdots \quad 0}_{d_X(i-1)+j-1} \;\; (W_2)_i \;\; \underbrace{0 \quad \cdots \quad 0}_{d_X d_1 - d_X(i-1) - j} \;]\, \hat{X}_k \nabla_{\left( \hat{W}\hat{X} \right)_{\cdot,k}} l\left( Y_{\cdot,k}, \left( \hat{W}\hat{X} \right)_{\cdot,k} \right), \quad (66)
$$

where $(W_2)$ is a vector and $(W_2)_i$ is its $i$-th component.

Then, divid the both hand sides of eq. (66) with $(W_2)_i$, we can get the following equation,

$$
(e_{d_X(i-1)+j}\hat{X})\nabla = 0.
$$

**Case 2:** $(\mathbf{W_2})_\mathbf{i} = \mathbf{0}$. Suppose $\mathbf{u}_1 \in \mathbb{R}^{d_0}$ is a unitary vector, $u_2 \in \mathbb{R}$ is a real number, and $\varepsilon$ is a small enough positive constant. Then, define a disturbance of $W_1$ and $W_2$ as follows,

$$
\Delta W_1 = [\; 0 \; \underbrace{\cdots \quad 0}_{d_X(i-1)} \;\; \varepsilon\mathbf{u}_1 \;\; \underbrace{0 \quad \cdots \quad 0}_{d_1 d_X - d_X i} \;],
$$

$$
\Delta W_2 = [\; 0 \; \underbrace{\cdots \quad 0}_{i-1} \;\; \varepsilon^2 u_2 \;\; \underbrace{0 \quad \cdots \quad 0}_{d_1 - i} \;].
$$

When $\varepsilon$ is sufficiently small, $\Delta W_1$ and $\Delta W_2$ are also sufficiently small. Since $(W_1, W_2)$ is a local minimum, we have

$$
\frac{1}{n}\sum_{k=1}^{n} l\left( Y_k, \left( \left( \hat{W} + \Delta \right)\hat{X} \right)_k \right)
$$

$$
= \hat{\mathcal{R}}(W_1 + \Delta W_1, W_2 + \Delta W_2)
$$

$$
\geq \hat{\mathcal{R}}(W_1, W_2)
$$

$$
= \frac{1}{n}\sum_{k=1}^{n} l\left( Y_k, \left( \hat{W}\hat{X} \right)_k \right), \quad (67)
$$

where $\Delta$ is defined as follows,

$$
\Delta = [(W_2 + \Delta W_2)_1(W_1 + \Delta W_1)_1 \quad \cdots \quad (W_2 + \Delta W_2)_{d_1}(W_1 + \Delta W_1)_{d_1}]
$$

$$
- [(W_2)_1(W_1)_1 \quad \cdots \quad (W_2)_{d_1}(W_1)_{d_1}]
$$

$$
\overset{(*)}{=} [\; 0 \; \underbrace{\cdots \quad 0}_{d_X(i-1)} \;\; \varepsilon^2 u_2 \left( \varepsilon\mathbf{u}_1 + (W_1)_i \right) \;\; \underbrace{0 \quad \cdots \quad 0}_{d_1 d_X - d_X i} \;]. \quad (68)
$$

Here, eq. $(*)$ comes from $(W_2)_i = 0$. Rearrange eq. (67) and apply the Taylor's Theorem, we can get that

$$
\Delta \cdot \hat{X}\nabla + \mathbf{O}\left( \|\Delta \cdot \hat{X}\|^2 \right) \geq 0.
$$

Applying eq. (68), we have

$$\begin{bmatrix} 0 & \underbrace{\cdots}_{d_X(i-1)} & 0 & \varepsilon^2 u_2 \left( \varepsilon \mathbf{u}_1 + (W_1)_i \right) & 0 & \underbrace{\cdots}_{d_X d_1 - id_X} & 0 \end{bmatrix} \hat{X} \nabla$$

$$+ \varepsilon^4 O \left( \left\| \begin{bmatrix} 0 & \underbrace{\cdots}_{d_X(i-1)} & 0 & u_2 \left( \varepsilon \mathbf{u}_1 + (W_1)_i \right) & 0 & \underbrace{\cdots}_{d_X d_1 - id_X} & 0 \end{bmatrix} \hat{X} \right\|^2 \right)$$

$$\overset{(**)}{=} \begin{bmatrix} 0 & \underbrace{\cdots}_{d_X(i-1)} & 0 & \varepsilon^3 u_2 \mathbf{u}_1 & 0 & \underbrace{\cdots}_{d_X d_1 - id_X} & 0 \end{bmatrix} \hat{X} \nabla$$

$$+ \varepsilon^4 O \left( \left\| \begin{bmatrix} 0 & \underbrace{\cdots}_{d_X(i-1)} & 0 & u_2 \left( \varepsilon \mathbf{u}_1 + (W_1)_i \right) & 0 & \underbrace{\cdots}_{d_X d_1 - id_X} & 0 \end{bmatrix} \hat{X} \right\|^2 \right)$$

$$= \varepsilon^3 \begin{bmatrix} 0 & \underbrace{\cdots}_{d_X(i-1)} & 0 & u_2 \mathbf{u}_1 & 0 & \underbrace{\cdots}_{d_X d_1 - id_X} & 0 \end{bmatrix} \hat{X} \nabla + \mathbf{o}(\varepsilon^3) \tag{69}$$

$$\geq 0 . \tag{70}$$

Here, eq. $(**)$ can be obtained from follows. Because $W_2$ is a local minimizer, for any component $(W_2)_i$ of $W_2$,

$$\frac{\partial \left( \sum_{k=1}^n l \left( Y_k, \left( \hat{W} \hat{X} \right)_k \right) \right)}{\partial (W_2)_i} = 0,$$

which leads to

$$\begin{bmatrix} 0 & \underbrace{\cdots}_{d_X(i-1)} & 0 & (W_1)_i & 0 & \underbrace{\cdots}_{d_X d_1 - id_X} & 0 \end{bmatrix} \hat{X} \nabla = 0.$$

When $\varepsilon$ approaches 0, eq. (69) leads to the following inequality,

$$\begin{bmatrix} 0 & \underbrace{\cdots}_{d_X(i-1)} & 0 & u_2 \mathbf{u}_1 & 0 & \underbrace{\cdots}_{d_X d_1 - id_X} & 0 \end{bmatrix} \hat{X} \nabla \geq 0.$$

Since $\mathbf{u}_1$ and $u_2$ are arbitrarily picked (while the norms equal 1), the inequality above further leads to

$$\begin{bmatrix} 0 & \underbrace{\cdots}_{d_X(i-1)} & 0 & e_j & 0 & \underbrace{\cdots}_{d_X d_1 - id_X} & 0 \end{bmatrix} \hat{X} \nabla = 0, \tag{71}$$

which finishes the proof of the argument.

Therefore, for any $i$ and $j$, we have proven that

$$e_{d_0(i-1)+j} \hat{X} \nabla = 0,$$

which demonstrates that

$$\hat{X} \nabla = 0,$$

which means $\hat{W}$ is also a local minimizer of the empirical risk $\hat{\mathcal{R}}$,

$$\hat{\mathcal{R}}(W) = \sum_{i=1}^n l(Y_i, W \hat{X}_i). \tag{72}$$

(3) Applying the property of convex function, $\hat{W}$ is a global minimizer of the empirical risk $\hat{\mathcal{R}}$, which leads to $(W_1, W_2)$ is a global minimum inside this cell.

The proof is completed. □

### B.4 EQUIVALENCE CLASSES OF LOCAL MINIMUM VALLEYS IN CELLS.

*Proof of Theorem 3 and Corollary 1.* In the proof of Theorem 2, we constructed a map $Q$: $(W_1, W_2) \rightarrow \hat{W}$. Further, in any fixed cell, the represented hypothesis of a neural network is uniquely determined by $\hat{W}$.

**We first prove that all local minima in a cell are concentrated as a local minimum valley.**

Since the loss function $l$ is strictly convex, the empirical risk has one unique local minimum (which is also a global minimum) with respect to $\hat{W}$ in every cell, if there exists some local minimum in the cell. Meanwhile, we have proved that all local minima with respect to $(W_1, W_2)$ are also local minima with respect to the corresponding $\hat{W}$. Therefore, all local minima with respect to $(W_1, W_2)$ correspond one unique $\hat{W}$. Within a cell, when $W_1$ expands by a positive real factor $\alpha$ to $W_1'$ and $W_2$ shrinks by the same positive real factor $\alpha$ to $W_2'$, we have $Q(W_1, W_2) = Q(W_1', W_2')$, i.e., the $\hat{W}$ remains invariant.

Further, we argue that all local minima in a cell are connected with each other by a continuous path, on which the empirical risk is invariant. For every local minima pair $(W_1, W_2)$ and $(W_1', W_2')$, we have

$$\text{diag}(W_2)W_1 = \text{diag}(W_2')W_1'. \tag{73}$$

Since $h'_{s_-, s_+}(W_1 X) = h'_{s_-, s_+}(W_1' X)$ (element-wise), for every $i \in [1, d_1]$,

$$\text{sgn}\left((W_2)_i\right) = \text{sgn}\left((W_2')_i\right).$$

Therefore, a continuous path from $(W_1, W_2)$ to $(W_1', W_2')$ can be constructed by finite moves, each of which expands a component of $W_2$ by a real constant $\alpha$ and then shrinks the corresponding line of $W_1$ by the same constant $\alpha$.

**We then prove that all local minima in a cell constitute an equivalence class.**

Define an operation $\sim_R$ as follows,

$$(W_1^1, W_2^1) \sim_R (W_1^2, W_2^2),$$

if

$$Q(W_1^1, W_2^1) = Q(W_1^2, W_2^2).$$

We then argue that $\sim_R$ is an equivalence relation. The three properties of equivalence relations are checked as follows.

**(1) Reflexivity:**

For any $(W_1, W_2)$, we have

$$Q(W_1, W_2) = Q(W_1, W_2).$$

Therefore,

$$(W_1, W_2) \sim_R (W_1, W_2).$$

**(2) Symmetry:**

For any pair $(W_1^1, W_2^1)$ and $(W_1^2, W_2^2)$, Suppose that

$$(W_1^1, W_2^1) \sim_R (W_1^2, W_2^2).$$

Thus,

$$Q(W_1^1, W_2^1) = Q(W_1^2, W_2^2).$$

Apparently,

$$Q(W_1^2, W_2^2) = Q(W_1^1, W_2^1).$$

Therefore,

$$Q(W_1^2, W_2^2) \sim_R Q(W_1^1, W_2^1).$$

**(3) Transitivity:**

For any $(W_1^1, W_2^1)$, $(W_1^2, W_2^2)$, and $(W_1^3, W_2^3)$, suppose that

$$(W_1^1, W_2^1) \sim_R (W_1^2, W_2^2),$$
$$(W_1^2, W_2^2) \sim_R (W_1^3, W_2^3).$$

Then,

$$Q(W_1^1, W_2^1) = Q(W_1^2, W_2^2),$$
$$Q(W_1^2, W_2^2) = Q(W_1^3, W_2^3).$$

Apparently,

$$Q(W_1^1, W_2^1) = Q(W_1^2, W_2^2) = Q(W_1^3, W_2^3).$$

Therefore,

$$(W_1^1, W_2^1) \sim_R (W_1^3, W_2^3).$$

**We then prove the mapping $Q$ is the quotient map.**

Define a map as follows,

$$T : (W_1, W_2) \to (diag(W_2)W_1, \mathbf{1}_{1 \times d_1}).$$

We then define an operator $\oplus$ as,

$$(W_1^1, W_2^1) \oplus (W_1^2, W_2^2) = T(W_1^1, W_2^1) + T(W_1^2, W_2^2),$$

the inverse of $(W_1, W_2)$ is defined to be $(-W_1, W_2)$ and the zero element is defined to be $(\mathbf{0}, \mathbf{1}_{1 \times d_1})$.

Obviously, the following is a linear mapping:

$$Q : ((\mathbb{R}^{d_1 \times d_X}, \mathbb{R}^{1 \times d_1}), \oplus) \to (\mathbb{R}^{1 \times d_X d_1}, +).$$

For any pair $(W_1^1, W_2^1)$ and $(W_1^2, W_2^2)$, we have

$$(W_1^1, W_2^1) \sim_R (W_1^2, W_2^2),$$

if and only if

$$(W_1^1, W_2^1) \oplus (-W_1^2, W_2^2) \in \text{Ker}(Q).$$

Therefore, the quotient space $(\mathbb{R}^{d_1 \times d_X}, \mathbb{R}^{1 \times d_1})/\text{Ker}(Q)$ is a definition of the equivalence relation $\sim_R$.

The proof is completed. $\square$

## B.5 LINEAR COLLAPSE.

When there is no nonlinearities in the activations, there is apparently no non-differentiable regions on the loss surface. In other words, the loss surface is a single smooth and multilinear cell.

