# OpenReview forum: "Piecewise linear activations substantially shape the loss surfaces of neural networks"
_ICLR.cc/2020/Conference — Accept (Poster)_

### Official Review · AnonReviewer2 · 2019-10-22
**Official Blind Review #2**

**Rating:** 6

**Review:**

This paper focus on how activation functions’ nonlinearities shape the loss surface of neural networks. The authors first show why the loss surface of every neural network has infinite spurious local minima. Secondly, the authors prove one theorem to show four properties of the loss surfaces of nonlinear neural networks.
Although this paper is generally easy to follow, and the motivation about nonlinearities and the loss surface is clear, the insight of this paper is somehow shortcoming. Though this work can prove such properties within different preconditions, whether other works’ conditions are inconvenient or not may remain further discussions. This work is established based on several preconditions, while it is hard to assert that most kinds of neural networks can satisfy them perfectly. For instance, this work mentions “Deep learning without poor local minima (NeurIPS2016)”, which requires full-rank and conditional independence of each node. It could be feasible when training a stacked network with particular limitations. This work requires all hidden layers are wider than the output layer, which may not be suitable for image segmentation, generative tasks or super-resolution, etc. Besides, it is laudable to prove fundamental rules in neural networks, while showing or inspiring researchers about how to implement or approximate such results to improve neural networks might be more helpful.
Some questions:

1. The authors assert that “the loss surface of *every* neural network has infinite spurious local minima” in the abstract, while in chapter 3 line 2, authors mention, “We find that *almost all* practical neural networks have infinitely many spurious local minima.” Which one is correct? Based on the following description in this paper, I guess this result is conditionally tenable.

2. In lemma 3, authors construct the local minima by adding very negative biases and show they are spurious. However, it is less likely to learn such negative biases in the real case. Besides, some networks require biases equal to zero to achieve some specific target. My question is: if biases are conditioned on real-world data distribution, will lemma 3 and 4 still work in this case?

3. This paper mentions “infinite” many times. Based on the reference, I believe that the “neural network” in this work refers to the “artificial neural network,” which is majorly stored within float tensors. So the number of combinations of parameters is finite. So why use “infinite” instead of “many”? Finite means I can train a small scale of networks with fewer precisions and check the global minima with a fixed dataset.

All in all, I believe this paper can be significantly improved if more details and experiments are provided.

**Experience Assessment:**

I do not know much about this area.

**Review Assessment: Checking Correctness Of Derivations And Theory:**

I assessed the sensibility of the derivations and theory.

**Review Assessment: Checking Correctness Of Experiments:**

N/A

**Review Assessment: Thoroughness In Paper Reading:**

N/A

---

> ### Author Response · Authors · 2019-11-08
> **Response to Reviewer #2**
>
> We appreciate your thorough review and constructive comments. All your concerns have been duly addressed below. We have also updated the final version accordingly. We sincerely hope that you can take into account the response we have made and reevaluate the merit of this paper.
>
>
>
> Q1: The authors assert that “the loss surface of *every* neural network has infinite spurious local minima” in the abstract, while in chapter 3 line 2, authors mention, “We find that *almost all* practical neural networks have infinitely many spurious local minima.” Which one is correct? Based on the following description in this paper, I guess this result is conditionally tenable.
>
> A1: Thanks and revised accordingly. In the final version, we have added the following description.
>
> We first prove that the loss surfaces of many neural networks have infinite spurious local minima, which are defined as the local minima with higher empirical risks than the global minima. Our result holds for any neural network with arbitrary depth and arbitrary piecewise linear activation functions (excluding linear functions) under many popular loss functions in practice with some mild assumptions.
>
>
>
> Q2: In lemma 3, authors construct the local minima by adding very negative biases and show they are spurious. However, it is less likely to learn such negative biases in the real case. Besides, some networks require biases equal to zero to achieve some specific target. My question is: if biases are conditioned on real-world data distribution, will lemma 3 and 4 still work in this case?
>
> A2: We respectfully argue that the construction of negative bias does not undermine the generality of the obtained results.
>
> Under a strong restriction that all activations are linear functions, Kawaguchi (2016), Zhou & Liang (2018), and Lu & Kawaguchi (2017) showed that all local minima are global minima, which accounts for the success of deep learning. However, it has been well observed and acknowledged that SGD can converge to points with large training errors, which are apparently not globally optimal. This phenomenon motivates us to study the existence of spurious local minima by relaxing this strong restriction.
>
> Theorem 1 of this paper (based on Lemmas 3 and 4) exactly constructs spurious local minima on the loss surface of a nonlinear neural network (with an arbitrary depth, a differentiable loss and an arbitrary-dimensional output). This counterexample proves that the existing theoretical results cannot be applied to nonlinear networks. Constructing counterexamples is a widely used approach to prove a proposition is wrong. Therefore, our construction does not undermine the generality.
>
>
>
> Q3: This paper mentions “infinite” many times. Based on the reference, I believe that the “neural network” in this work refers to the “artificial neural network,” which is majorly stored within float tensors. So the number of combinations of parameters is finite. So why use “infinite” instead of “many”? Finite means I can train a small scale of networks with fewer precisions and check the global minima with a fixed dataset.
>
> A3: We respectfully argue that it is common yet mild to treat the parameters of neural networks as continuous numbers for theoretical studies, which has been widely used in related studies. Moreover, the constructed local minima are connected with each other by a continuous path, on which every point has the same empirical risk. Therefore, it is impractical to check all the constructed local minima even when they are represented by float tensors, because the number of float tensors on a continuous path is extremely large. For example, there are $2^{52} = 4.5 \times 10^{15}$ $64$-bit float values between $1$ and $2$ when using the double precision.
>
>
>
> Reference
>
> Kenji Kawaguchi. Deep learning without poor local minima. In Advances in Neural Information Processing Systems, 2016.
>
> Haihao Lu and Kenji Kawaguchi. Depth creates no bad local minima. arXiv preprint arXiv:1702.08580, 2017.
>
> Yi Zhou and Yingbin Liang. Critical points of neural networks: Analytical forms and landscape properties. In International Conference on Learning Representations, 2018.

---

> > ### Comment · AnonReviewer2 · 2019-11-13
> > **Official Blind Review #2**
> >
> > Thank you for the response. After reading the material a second time and considering the amount of time the authors spent on developing the theory, I'm willing to upgrade the score for this paper and bring this study to more readers in the field.

---

> > > ### Author Response · Authors · 2019-11-13
> > > **Thank you!**
> > >
> > > Thank you very much for your support!

---

### Official Review · AnonReviewer3 · 2019-10-23
**Official Blind Review #3**

**Rating:** 6

**Review:**

This paper studies the theoretical property of neural network's loss surface. The main contribution is to prove that the loss surface of every neural network (with arbitrary depth) with piecewise linear activations has infinite spurious local minima. Moreover, the paper further characterizes the partition of the local minima. More precisely, the loss surface is partitioned into multiple smooth and multilinear open cells and within each cell, the local minima are equally good. This result can also explain the linear neural network case where there is only one cell, implying that all local minima are global.

On one hand, I find the paper very clear and the result very clean, which unites a lot of existing results. On the other hand, with a reasonable initialization in practice, we will not attain the local minima constructed in the paper since it requires all the activations to be positive. This limits the plausible implication from this theoretical study. Overall, I am very positive of the paper, the following are some detailed comments.

a. Please be more precise in the abstract that the activation function need to be piecewise linear.
The current sentence "the loss surface of every neural network has infinite spurious local minima" does not include this specification. Moreover, if the activation is differentiable, is the claim still hold? It seems to me from the middle of page 3 that Li et al 2018 shows a non-local minima result in this case.

b. How different is the analysis comparing to existing result?
I have only go through the skeleton of the proof and have not read into the details. It seems to me the construction of the local minima is very similar to [1], since the main idea is to consider the linear region by activating all the neurons. Could you summarize the main difficulty to extend their results to multi-layer cases? (Maybe it would be good to illustrate with a simple case like 3 layers few neurons per layer)
Moreover, when considering the local convexity, is it sufficient to say that locally in each cell it is a linear network and then the results on linear network transfers to it locally?

[1] Small nonlinearities in activation functions create bad local minima in neural networks, Yun et al, 2019

**Experience Assessment:**

I have published one or two papers in this area.

**Review Assessment: Checking Correctness Of Derivations And Theory:**

I assessed the sensibility of the derivations and theory.

**Review Assessment: Checking Correctness Of Experiments:**

N/A

**Review Assessment: Thoroughness In Paper Reading:**

I read the paper at least twice and used my best judgement in assessing the paper.

---

> ### Author Response · Authors · 2019-11-08
> **Response to Reviewer #3 (Part 1 of 2)**
>
> We appreciate your thorough review and constructive comments. Thank you very much for your kind support. All your concerns have been duly addressed below. We have also updated the final version accordingly.
>
>
>
> Q1: Please be more precise in the abstract that the activation function needs to be piecewise linear. Moreover, if the activation is differentiable, is the claim still hold?
>
> A1: Thanks and revised accordingly. We have stated in the abstract that we proved the cases of piecewise linear activation functions. In addition, the results have not been extended to differentiable activations.
>
>
>
> Q2: How different is the analysis comparing to existing result with [1]? Could you summarize the main difficulty to extend their results to multi-layer cases?
>
> A2: Thanks and revised accordingly. A detailed comparison of the analysis has been added to the final version. We summarise it as follows.
>
> We first acknowledge that [1] and our paper both employ the following strategy: (a) construct a series of local minima based on a linear classifier; and (b) construct a new point with smaller empirical risk and by this way we prove that the constructed local minima are spurious. However, due to the differences in the loss function and the output dimensions, the exact constructions of local minima are substantially different. Meanwhile, our Stages (2) and (3) employ the transformation operation to force the data flow to go through the same series of the linear parts of the activations. The operations are carefully designed and the whole construction is novel and non-trivial.
>
> Besides, we also made extensions on the loss function and the output dimension. The difficulties are justified below:
>
> 1.	From squared loss to arbitrary differentiable loss: Yun et al. (2019b) calculate the analytic formations of derivatives of the loss to construct the local minima and then prove they are spurious. This technique cannot be transplanted to the case of arbitrary differentiable loss functions, because we cannot assume the analytic formation. To prove that the loss surface under an arbitrary differentiable loss has an infinite number of spurious local minima, we employ a new proof technique based on Taylor series and a new separation lemma (see Appendix A.5, Lemma 6, p. 31) to avoid the use of the analytic formulations (see a detailed proof in Appendix A.2, Step (b), pp.14-15).
>
> 2.	From one-dimensional output to arbitrary-dimensional output: To prove the loss surface of a neural network with an arbitrary-dimensional output has an infinite number of spurious local minima, we need to deal with the calculus of functions whose domain and codomain are a matrix space and a vector space, respectively. By contrast, when the output dimension is one, the codomain is only the space of real numbers. Therefore, the extension of output dimension significantly mounts the difficulty of the whole proof.

---

> ### Author Response · Authors · 2019-11-08
> **Response to Reviewer #3 (Part 2 of 2)**
>
> Q3: Moreover, when considering the local convexity, is it sufficient to say that locally in each cell it is a linear network and then the results on linear network transfers to it locally?
>
> A3: Thanks. It is not sufficient to say that. We have added a detailed explanation to the final version. The significance of our proof is justified below.
>
> Technically, linear networks can be expressed by the product of a sequence of weight matrices, which guarantees good geometrical properties. Specifically, the effect of every linear activation function is just equivalently multiplying a real constant to the output. However, the loss surface within a cell of a nonlinear neural network does not have this property.
>
> We take a one-hidden-layer network for example. Suppose the output of the hidden layer is $h(WX))$, where $X$ is the data matrix, $h$ is the activation function, and $W$ is the weight matrix. If $h$ is a linear function $h(x) = ax$, its effect is equivalently multiplying the constant a to every element of the matrix $WX$. However, when the activation h is a two-piece linear function which has slopes $\{a_1, a_2\}$, different elements in the matrix $WX$ can be multiplied by either one from $\{a_1, a_2\}$. Therefore, we cannot use a single constant to express the effect of this activation, and thus, even within the cell, a nonlinear network cannot be expressed as the product of a sequence of weight matrices. This difference ensures that the proofs of deep linear neural networks cannot be transplanted here.
>
> To address this issue, we develop a non-trivial approach to prove that all local minima in a cell are globally optimal within the cell. Specifically, we prove this in two steps: (1) we prove that within every cell, the empirical risk $\hat{\mathcal R}$ is convex with respect to a variable $\hat W$ mapped from the weights $W$. Therefore, the local minima with respect to the variable $\hat W$ are also the global minima in the cell; and then (2) we prove that the local optimality is maintained under the constructed mapping. Specifically, the local minima of the empirical risk $\hat{\mathcal R}$ with respect to the parameter $W$ are also the local minima with respect to the corresponding variable $\hat W$.

---

> > ### Comment · AnonReviewer3 · 2019-11-08
> > **Clarification on cells**
> >
> > Thank you for the detailed clarification!
> > In my mind, a cell is defined with respect to a specific configurations on the activation function, for example all uses $a_`1$ or half $a_1$ half $a_2$, in which case it becomes multilinear. This seems contradictory to your explanation. A possible reason is that the notion of cell in the paper may be different from this naive definition, in which case, could you provide a  more precise example on what a cell should look like (in a simple case)? That would be helpful for better understanding.

---

> > > ### Author Response · Authors · 2019-11-09
> > > **Re: Clarification on cells**
> > >
> > > Thank you very much for your quick reply! We will explain it below. The final version has also been revised accordingly.
> > >
> > > Please let us define some notions first. Suppose $WX$ is a $2 \times 2$ matrix, whose elements are $c_{1, 1}$, $c_{1, 2}$, $c_{2, 1}$, and $c_{2, 2}$. Then, the effect of the activation $h$ is equivalently multiplying either $a_1$ or $a_2$ to the four elements. For the brevity, let’s express the configuration by a matrix $A$ that collects the multiplied factors. Specifically, if the effect of the activation $h$ on $c_{i, j}$ is multiplying $a_1$, the element $A_{i, j}$ is defined to be $a_1$.
> > >
> > > We agree that a cell is defined with respect to a specific configuration of the activation function. In other words, the matrix $A$ is invariant within a cell. Specifically, when the weight matrix $W$ changes but is still in the cell, all elements $c_{i, j}$ of the matrix $WX$ changes small enough such that the configuration matrix $A$ remains invariant. By contrast, if the weight matrix $W$ crosses the boundaries between two cells, the configuration matrix $A$ changes. Therefore, the interior of a cell is multilinear and smooth, but the boundaries are non-differentiable.
> > >
> > > However, it is not contradictory to our explanation. Within a fixed cell, the elements of matrix $A$ are not necessarily equal. For example, the elements $A_{1, 1}$, $A_{1, 2}$, $A_{2, 1}$, and $A_{2, 2}$ can be $a_1$, $a_1$, $a_2$, and $a_1$, respectively. Therefore, the effect of the activation $h$ is not equivalently multiplying a single constant (or a single matrix) to the matrix $WX$, which is however the foundation of the proofs for linear networks. Thus, we cannot transplant the case of linear networks here.

---

> > > > ### Comment · AnonReviewer3 · 2019-11-09
> > > > **Re: Clarification on cells**
> > > >
> > > > Thanks for the clarification. I see the difference now: the linear network has coordinate-wise activation function, but when dealing with the specific configuration, the activation is no longer coordinate-wise and a matrix factor A kicks in, which breaks down the multiplicative "chain" structure and makes the analysis non-trivial. I have quickly checked the proof of this part, it seems like the proof only considers two layer cases? Will it be similar to deal with multiple A in the deep cases?

---

> > > > > ### Author Response · Authors · 2019-11-10
> > > > > **Re: Clarification on cells**
> > > > >
> > > > > Thanks for your reply! Yes, you are right. The configuration matrix $A$ changes the game. It makes the geometry within a cell more complicated than the loss surface of linear networks.
> > > > >
> > > > > In addition, the current proof rigorously holds for two-layer cases. The two-step strategy is feasible for deep neural networks: (1) we prove that within every cell, the empirical risk $\hat R$ is convex with respect to a variable $\hat W$ mapped from the weights $W$; and (2) we further prove that the local optimality is retained under the constructed mapping. However, we need a new concrete construction of the mapping from the weight $W$ to the variable $\hat W$ for deep neural networks.

---

> > > > > > ### Comment · AnonReviewer3 · 2019-11-10
> > > > > > **Re: Clarification on cells**
> > > > > >
> > > > > > Thank you for your responses! My concerns are carefully addressed. I will keep my score and discuss with other reviewers in favor of the acceptance.

---

> > > > > > > ### Author Response · Authors · 2019-11-10
> > > > > > > **Thank you!**
> > > > > > >
> > > > > > > Thank you very much for your support!

---

### Official Review · AnonReviewer1 · 2019-10-27
**Official Blind Review #1**

**Rating:** 3

**Review:**

Summary: This paper studies the landscape of deep neural networks with piecewise-linear activation functions. The paper showed that under very mild assumptions, the loss surface admits infinite spurious local minima. Further, it is shown that the loss surface is partitioned into many multilinear cells. If the network is two-layer with two-piece linear activations, it is proved that within each cell every local minimum is global.

Pros:
  --Constructed spurious local minima for piecewise linear activations, for a broader setting than previous papers.
  --The paper is well written, with detailed explanation of proof skeleton.

Cons:
The significance of the results are not clear. Details are given below.

1.	This paper only considers piecewise linear activation, which is a special type of non-linear activation. The major examples are ReLU and leaky ReLU. However, related results for ReLU have been studied for a few previous works mentioned in the last two paragraphs of Sec. 3.2. In particular, Yun et al. 2019b already proves a similar result for 1-hidden-layer neural-net with ReLU activation. I think extending the construction to broader settings (any depth, any piecewise linear and more losses) is mathematically nice, but the motivation of this extension is somewhat unclear to me. One motivation is that this is helpful for the purpose of understanding a “big picture” of the landscape, which I will discuss next.

2.	The second major result is Theorem 2, on the “big picture” with ReLU-like activations. However, Theorem 2 is somewhat trivial to prove, and the link to Theorem 1 is rather weak.
   (a) The main message of Thm 2 is the partition of the surface into multiple pieces, and each piece has good property. This partition is somewhat straightforward, and has been studied before, in, e.g., [R1].
   For a global “big picture”, partitioning itself is not very interesting. Theorem 2 mainly describes the property of each region separately for 2-layer network, which is weaker than [R1].
   (b) Theorem 2 seems easy to prove. The 1st, 3rd and 4th property are all straightforward. The 2nd property “local analogous convexity” was given a 2-page proof in the paper. However, I don’t understand why not use the following simple argument: for each region, the network behaves like a deep linear network, thus directly applying existing result shall imply “every local minimum in the region is the global minimum of the region”, right? If not, what is the difficulty?
   (c) The 3rd property says “some local minima are concentrated as a valley in some cell”. What are the formal definitions of “concentrated” and “valley” in this sentence?
   (d) The link to Theorem 1 is weak: the link is the 3rd property of Theorem 2 that “some local minima are in a valley”. It is just about some special local minima and weakly related to the other properties on the “global view”. In addition, the fact that “some of them are in a valley” may be due to the very special construction, thus it is not surprising and does not reveal anything interesting about the “big picture”.


3.	Other issues:
a) While ReLU-type activations are popular, there are still commonly-used activation functions are not piece-wise linear, e.g., tanh, swish. It is not proper to claim that "this paper presents how nonlinearities in activations substantially shape the loss surface" and "almost every practical neural network ....". I suggest replacing "nonlinearity" with "piecewise linearity" in both the title and the abstract, and modifying the over-statements.
   b) In Property 1 of Theorem 2, “smooth and multilinear partition” might be a bit misleading. The loss surface should be fractional in general, where multilinear cells are separated by non-smooth boundaries. “Smooth partition” seems to imply that the boundaries are smooth or the partition method is smooth in some sense.
   c) The name “analogous convexity” is not appropriate. Analogous convexity is not formally defined in the paper. According to Sec. 4.3 third paragraph, “the property of analogous convexity that the local minima wherein are equally good”. It seems that “analogous convexity” is just “all local minima are good”, which is very different from convexity. It is a weaker property than quasi-convexity, star-convexity, etc, and thus it is better not to call it “analogous convexity”.
    d) Property 3 of Theorem 2 is very far from “mode connectivity”. The proof of Property 3 relies on a special construction of Theorem 1, and the latter is for two arbitrary global minima.


[R1] Soudry and Hoffer. "Exponentially vanishing sub-optimal local minima in multilayer neural networks." arXiv preprint arXiv:1702.05777 (2017).


Conclusion:  I think this paper is studying an important and interesting question, and the efforts of constructing local minima and understanding big picture are both interesting to me. However, I’m afraid the current form of the paper does not meet the standard of the conference. That being said, it would be a nice paper if the big picture can be explored deeper, and the link to the spurious local minima can be built stronger.


**Experience Assessment:**

I have published in this field for several years.

**Review Assessment: Checking Correctness Of Derivations And Theory:**

I assessed the sensibility of the derivations and theory.

**Review Assessment: Checking Correctness Of Experiments:**

N/A

**Review Assessment: Thoroughness In Paper Reading:**

I read the paper at least twice and used my best judgement in assessing the paper.

---

> ### Author Response · Authors · 2019-11-08
> **Response to Reviewer #1 (Part 1 of 3)**
>
> We appreciate your thorough review and constructive comments. All your concerns have been duly addressed below. We have also updated the final version accordingly. We sincerely hope that you can take into account the response we have made and reevaluate the merit of this paper.
>
>
> Q1: Related results have been studied in a few previous works. In particular, Yun et al. (2019b) prove a similar result for 1-hidden-layer neural-net with ReLU activation. The extension is mathematically nice, but the motivation of this extension is somewhat unclear.
>
> A1: Our paper studies the theoretical foundations of deep learning. We acknowledge the significant contributions made by Yun et al. (2019b), which are however under some restrictions, including one hidden layer, squared loss, and one-dimensional output. Thus, it is not sufficient to comprehensively build the theoretical foundation for deep learning. To date, the theoretical study of deep learning is still in its infancy. Significant efforts are really demanded. The motivation and significance of our studies are justified below:
>
> 1.	From one hidden layer to arbitrary depth: Empirical results have overwhelmingly suggested that the increase of the depth of neural networks may substantially improve the performance. Additionally, training neural networks is increasingly difficult when the networks turn deeper. Therefore, the depth would play a significant role in shaping the loss surface of a neural network. To prove that networks with an arbitrary depth have infinite spurious local minima, we develop a novel strategy that employs transformation operations to force data flow through the same linear parts of the activations, in order to construct the spurious local minima (see a summary in Section 3.3, Stages 2 and 3, pp. 5-6; and a detailed proof in Appendix A.3 and A.4, pp. 20-31).
>
> 2.	From squared loss to arbitrary differentiable loss: Many other loss functions, such as cross-entropy loss, are widely utilized in deep learning. Only considering squared loss is insufficient. Yun et al. (2019b) calculate the analytic formations of derivatives of the loss to construct the local minima and then prove they are spurious. This technique cannot be transplanted to the case of arbitrary differentiable loss functions, because we cannot assume the analytic formation. To prove that the loss surface under an arbitrary differentiable loss has an infinite number of spurious local minima, we employ a new proof technique based on Taylor series and a new separation lemma (see Appendix A.5, Lemma 6, p. 31) to avoid the use of the analytic formulations (see a detailed proof in Appendix A.2, Step (b), pp.14-15).
>
> 3.	From one-dimensional output to arbitrary-dimensional output: Most datasets have high-dimensional labels. For example, the label dimension is 10 in CIFAR-10, 100 in CIFAR-100, and 1,000 in ImageNet.  To prove the loss surface of a neural network with an arbitrary-dimensional output has an infinite number of spurious local minima, we need to deal with the calculus of functions whose domain and codomain are a matrix space and a vector space, respectively. By contrast, when the output dimension is one, the codomain is only the space of real numbers. Therefore, the extension of output dimension significantly mounts the difficulty of the whole proof.
>
>
> Q2(a): The main message of Thm 2 is the partition of the surface into multiple pieces, and each piece has good property. This partition has been studied before, in, e.g., [R1]. Theorem 2 mainly describes the property of each region separately for 2-layer network, which is weaker than [R1].
>
> A2(a): Thank you for bringing the paper [R1] to our attention. We agree that we overlooked this paper in the original submission. We have duly acknowledged this paper in the final version. We noted that Lemma 2 in [R1] is similar to the 2nd property of our Thm 2. However, our proof is completely different from that in [R1] and our result is stronger and more general. Specifically, in Property 2 of Thm 2, we proved that within every cell, all local minima are glocal minimal in the cell. However, the Lemma 2 in [R1] only proves that the local minima in a cell are the same; there would be some point near the boundary has smaller empirical risk and is not locally minimal. Unfortunately, the proof in [R1] cannot exclude this possibility. Furthermore, our proof holds for any convex loss, including squared loss and cross-entropy loss, but [R1] only stands for squared loss.

---

> ### Author Response · Authors · 2019-11-08
> **Response to Reviewer #1 (Part 2 of 3)**
>
> Q2(b): Theorem 2 seems easy to prove. The 1st, 3rd and 4th property are all straightforward. The 2nd property “local analogous convexity” was given a 2-page proof in the paper. However, I don’t understand why not use the following simple argument: for each region, the network behaves like a deep linear network, thus directly applying existing result shall imply “every local minimum in the region is the global minimum of the region”, right? If not, what is the difficulty?
>
> A2(b): We agree the 1st and 4th properties can be obtained easily. The 3rd property is derived from Theorem 1, i.e., given Theorem 1, the 3rd property can be obtained easily as well. However, it is not easy to prove Theorem 1. Moreover, it is essential to include all the four properties in Theorem 2, because they collectively draw a picture of the loss surface. Specifically, without highlighting the partition (the 1st property), we have no grounds to study the geometry within the cells (the 2nd property) and we cannot show the constructed spurious local minima are exactly in one single cell (the 3rd property). Besides, the 1st property is essential to derive the 4th property to show linear networks are included as a simplified example of nonlinear networks.
>
> Also, it is challenging to prove the 2nd property, because the proof techniques for linear networks cannot be transplanted here. Technically, linear networks can be expressed by the product of a sequence of weight matrices, which guarantees good geometrical properties. Specifically, the effect of every linear activation function is just equivalently multiplying a real constant to the output. However, the loss surface within a cell of a nonlinear neural network does not have this property.
>
> We take a one-hidden-layer network for example. Suppose the output of the hidden layer is $h(WX))$, where $X$ is the data matrix, $h$ is the activation function, and $W$ is the weight matrix. If $h$ is a linear function $h(x) = ax$ (element-wise), its effect is equivalently multiplying the constant a to every element of the matrix $WX$. However, when the activation h is a two-piece linear function which has slopes $\{a_1, a_2\}$, different elements in the matrix $WX$ can be multiplied by either one from $\{a_1, a_2\}$. Therefore, we cannot use a single constant to express the effect of this activation, and thus, even within the cell, a nonlinear network cannot be expressed as the product of a sequence of weight matrices. This difference ensures that the proofs of deep linear neural networks cannot be transplanted here.
>
> To address this issue, we develop a novel and non-trivial approach to prove that all local minima in a cell are globally optimal within the cell. Specifically, we prove this in two steps: (1) we prove that within every cell, the empirical risk $\hat{\mathcal R}$ is convex with respect to a variable $\hat W$ mapped from the weights $W$. Therefore, the local minima with respect to the variable $\hat W$ are also the global minima in the cell; and then (2) we prove that the local optimality is maintained under the constructed mapping. Specifically, the local minima of the empirical risk $\hat{\mathcal R}$ with respect to the parameter W are also the local minima with respect to the corresponding variable $\hat W$.
>
>
>
> Q2(c): The 3rd property says “some local minima are concentrated as a valley in some cell”. What are the formal definitions of “concentrated” and “valley” in this sentence?
>
> A2(c): Thanks and revised accordingly. In the final version, we have changed it to: “some local minima are connected within a cell by a continuous path, on which all points have the same empirical risk”.
>
>
>
> Q2(d): The link to Theorem 1 is weak: the link is the 3rd property of Theorem 2 that “some local minima are in a valley”. It is just about some special local minima and weakly related to the other properties on the “global view”. In addition, the fact that “some of them are in a valley” may be due to the very special construction, thus it is not surprising and does not reveal anything interesting about the “big picture”.
>
> A2(d): First, Theorem 1 and Theorem 2 collectively support our argument that the nonlinearities in the activations substantially shape the loss surface. Specifically, Theorem 1 gives a negative result that the loss surfaces of nonlinear networks are substantially different from those of linear networks; and Theorem 2 provides positive results on how the nonlinearities work and how the loss surface looks like.
>
> Second, we agree that the 3rd property of Theorem 2 is based on special constructions. Therefore, we can say that “some local minima are connected by a continuous path within a cell, on which all points have the same empirical risk”.

---

> ### Author Response · Authors · 2019-11-08
> **Response to Reviewer #1 (Part 3 of 3)**
>
> Q3(a): While ReLU-type activations are popular, there are still commonly-used activation functions are not piece-wise linear, e.g., tanh, swish. It is not proper to claim that "this paper presents how nonlinearities in activations substantially shape the loss surface" and "almost every practical neural network ....". I suggest replacing "nonlinearity" with "piecewise linearity" in both the title and the abstract, and modifying the over-statements.
>
> A3(a): Thanks and revised accordingly. We have carefully improved the final version to avoid misunderstanding. However, we respectfully argue that “nonlinearity” is more common to refer to the difference from linear functions, which has been also used by Yun et al. (2019b).
>
>
>
> Q3(b): In Property 1 of Theorem 2, “smooth and multilinear partition” might be a bit misleading. The loss surface should be fractional in general, where multilinear cells are separated by non-smooth boundaries. “Smooth partition” seems to imply that the boundaries are smooth or the partition method is smooth in some sense.
>
> A3(b): Thanks and revised accordingly. Both “smooth” and “multilinear” refer to the geometry in the cell, which is defined as an open set that does not include the boundary (see abstract, introduction, and Theorem 2). To avoid misunderstanding, we have added an explanation of the both words: “the loss surface is partitioned into multiple smooth and multilinear open cells by nondifferentiable boundaries” and formally defined “open set” in the final version: “a set A is open if it does not contains any point in its boundary” (see the definition of boundary in Section 4.1, Definition 3, p. 6).
>
>
>
> Q3(c): The name “analogous convexity” is not appropriate. Analogous convexity is not formally defined in the paper. According to Sec. 4.3 third paragraph, “the property of analogous convexity that the local minima wherein are equally good”. It seems that “analogous convexity” is just “all local minima are good”, which is very different from convexity. It is a weaker property than quasi-convexity, star-convexity, etc, and thus it is better not to call it “analogous convexity”.
>
> A3(c): Thanks and revised accordingly. We have added an explanation of this name in the final version. There are two main reasons to use the name “local analogous convexity”: (1) within every cell, the empirical risk $\hat{\mathcal R}$ is convex with respect to a variable $\hat W$ mapped from the weights $W$. Therefore, the local minima with respect to the variable $\hat W$ are also the global minima in the cell; and then (2) the local optimality is maintained under the constructed mapping. Specifically, the local minima of the empirical risk $\hat{\mathcal R}$ with respect to the parameter W are also the local minima with respect to the variable $\hat W$. We will add this discussion to the final version.
>
>
>
> Q3(d): Property 3 of Theorem 2 is very far from “mode connectivity”. The proof of Property 3 relies on a special construction of Theorem 1, and the latter is for two arbitrary global minima.
>
> A3(d): We respectfully argue that the Property 3 of Theorem 2 shed lights to the study of “mode connectivity”, although it is an initial attempt. Specifically, “mode connectivity” says that the minima found by gradient-based methods are connected, while we exactly prove that some local minima are connected.

---

> > ### Comment · AnonReviewer1 · 2019-11-13
> > **Some questions remain; "big picture" somewhat oversold**
> >
> > Thanks for the explanation. I appreciate the efforts.   (I modified a previous response)
> >
> > 1) About Theorem 2.  I agree that there is a small difference between the conventional linear network and the linear network within each cell.
> >    However, due to linearity, it seems just a simple extension. There are quite a few proofs for the linear network. The 2-step proof is used before for analyzing linear networks, e.g. 1702.08580.  As for the first step of convexity, it is still a linear transformation of W^hat, thus it is convex (this is essentially the argument from (72) to (78)).  The novelty of this proof is not clear.
> >       The claim "our result is stronger and more general than [R1]" is questionable. This result is stronger in some aspects. But [R1] provides a deeper characterization by computing a certain probability. The proof is much longer than the 2-page proof here. And it is related to hyperplane geometry, not just longer. Again, I agree they are different, but the insight conveyed by [R1] is more nontrivial.
> >     Overall, I think Theorem 2 exaggerates the contribution of the big picture. The four things are highlighted in the abstract and sounds like a major contribution. But two are trivial; "every local minimum is good" in a cell is expected to researchers in this area. The proof is kind of simple, compared to the whole paper, but occupied large space of "conceptual contribution".
> >
> > 1b) Another point of exaggeration: the abstract first says "for any neural network with arbitrary depth". And later talks about "big picture". Only at the end of abstract mentions "1-hidden layer" for the second property --which is the most non-trivial one.  I think this might be misleading; readers would think the whole paper is about deep-net, and only a "minor result" is for 1-hidden layer.
> >    I did not check the paper again, but this needs to be clarified in the paper as well.
> >
> > 2) As for extension. " training neural networks is increasingly difficult when the networks turn deeper". This sentence seems to suggest that depth will make problem more difficult, so proving positive result is more difficult. This is not an argument for generating negative result.
> >     Thanks for explaining "Forcing the flow". The idea is natural (not meaning it is not good), as for piecewise linear one needs to utilize the linear part to create bad local minima. This is why [R1] is nontrivial: since sub-optimal local minima can exist for ReLU due to piecewise linear nature, thus it tried to show such bad cases are rare.
> >      Mathematically speaking, I agree there is a difference between deep and shallow. And I do appreciate the effort to really prove it. But the explanation is not for the technical nontriviality of proving "deep" instead of shallow.
> >
> > Other discussion:
> > 1) For A3(c): "empirical risk  is convex with respect to a variable mapped from the weights". "the local optimality is maintained under the constructed mapping"
> >    This is saying "convexity is kept after mapping" means "analogous convexity". This already exists in the original linear network, and this property was mentioned in, e.g., 1702.08580.  I still think calling "analogous convexity" is somewhat misleading. Why just say "every local min is global min"?
> >
> > 2) "we respectfully argue that “nonlinearity” is more common to refer to the difference from linear functions, which has been also used by Yun et al. (2019b)."
> >     I have to say it is not common. Yun et al. (2019b) analyzed a much bigger class of neurons than this paper. This paper just analyzed piecewise linear.
> >     In a field with overstatements, I would suggest using more precise title, like "the loss surface of neural networks with piecewise linear activations".

---

> > > ### Author Response · Authors · 2019-11-14
> > > **Re: Some questions remain; "big picture" somewhat oversold (Part 1 of 3)**
> > >
> > > Thank you very much for your constructive feedback. All your concerns have been carefully addressed. We will update the final version accordingly. We hope our detailed responses can fully answer your questions and the merit of this paper can be reevaluated. Much appreciated.
> > >
> > >
> > >
> > > Q1.1: About Theorem 2. I agree that there is a small difference between the conventional linear network and the linear network within each cell. However, due to linearity, it seems just a simple extension. There are quite a few proofs for the linear network. The 2-step proof is used before for analyzing linear networks, e.g. 1702.08580. As for the first step of convexity, it is still a linear transformation of W^hat, thus it is convex (this is essentially the argument from (72) to (78)).  The novelty of this proof is not clear.
> > >
> > > A1.1: We respectfully argue that all the existing proofs for linear networks cannot be transplanted here because the geometry within each cell of a nonlinear network is much more complicated than that of the loss surface of any linear network. We provide below a simple example to explain this in detail.
> > >
> > > We consider a simple one-hidden-layer neural network, in which:
> > >
> > > (1) the output $h(W_1 X)$ of the hidden layer is a $2 \times 2$ matrix with elements defined by $h(c_{1, 1})$, $h(c_{1, 2})$, $h(c_{2, 1})$, and $h(c_{2, 2})$, and
> > >
> > > (2) the output $W_2 h(W_1 X)$ of the network is another $2 \times 2$ matrix, with elements defined by $W_2 h(c_{1, 1})$, $W_2 h(c_{1, 2})$, $W_2 h(c_{2, 1})$, and $W_2 h(c_{2, 2})$.
> > >
> > > For a linear activation function $h$ defined by the slope $a$, the output of the hidden layer $h(c_{i, j})$ is $a c_{i, j}$. This means we can use the real number $a$ to express the effect of the activation function $h$, i.e., $h(W_1 X) = a W_1 X$. Therefore, a linear network is a linear model $W’ X$ with respect to the input $X$, where $W’ = W_2 a W_1 X = a W_1 W_2 X$ is the product of a sequence of weight matrices and real numbers. This is the foundation to complete the proof of the existence of spurious local minima for linear networks.
> > >
> > > By contrast, for a piecewise linear activation function $h$ with two slopes $a_1$ and $a_2$, the output of the hidden layer $h(c_{i, j})$ is either $a_1 c_{i, j}$ or $ a_2 c_{i, j}$. This means we can use neither a single real number nor a linear operator to express the effect of the activation $h$ in most cells (in some cell, the output $h(c_{i, j})$ of the hidden layer happens to be $a_1 c_{i, j}$ or $ a_2 c_{i, j}$ for all $(i, j)$). Therefore, all the existing proof techniques for linear networks do not apply here.
> > >
> > > The novelty of this proof is two-fold: (1) in Step 1, we construct a mapping from the weight $W$ to the variable $\hat W$ (it is the foundation to discuss the convexity of empirical risk $\hat{\mathcal R}$ with respect to the variable $\hat W$); and (2) in Step 2, we prove that the local optimality is retained under the constructed mapping. Although the proof of the convexity in Step 1 is simple and straightforward (due to the linearity of the empirical risk $\hat{\mathcal R}$ with respect to $\hat W$), it is a small portion of the proof of Theorem 2. Therefore, we respectfully argue that the proof for Theorem 2 in general is novel and technically non-trivial.
> > >
> > >
> > >
> > > Q1.2:  The claim "our result is stronger and more general than [R1]" is questionable. This result is stronger in some aspects. But [R1] provides a deeper characterization by computing a certain probability. The proof is much longer than the 2-page proof here. And it is related to hyperplane geometry, not just longer. Again, I agree they are different, but the insight conveyed by [R1] is more nontrivial.
> > >
> > > A1.2: Thanks. We agree to remove the claim “our result is stronger and more general than [R1]” and will acknowledge the significant contributions presented in [R1].
> > >
> > > We appreciate [R1] that helps the community understand deep learning from the perspective of the volume of the spurious local minima, which is demanded. Specifically, [R1] proves that measured by the volume, the ratio of spurious/suboptimal local minima in one-hidden-layer networks approaches $0$ when the training sample size $N$ goes to infinity.
> > >
> > > By contrast, this paper shows that (1) neural networks of arbitrary depth have infinite spurious local minima; and (2) all local minima within a cell are globally minimal. We would love to note that [R1] and this paper support each other to be more convincing, but not undermine the merit of the other side.

---

> > > ### Author Response · Authors · 2019-11-14
> > > **Re: Some questions remain; "big picture" somewhat oversold (Part 2 of 3)**
> > >
> > > Q1.3: Overall, I think Theorem 2 exaggerates the contribution of the big picture. The four things are highlighted in the abstract and sounds like a major contribution. But two are trivial; "every local minimum is good" in a cell is expected to researchers in this area. The proof is kind of simple, compared to the whole paper, but occupied large space of "conceptual contribution".
> > >
> > > A1.3: Thanks. We have revised the paper accordingly. The updated abstract is given below.
> > >
> > > Understanding the loss surface of a neural network is fundamentally important to the understanding of deep learning. This paper presents how piecewise linear activation functions substantially shape the loss surfaces of neural networks. We first prove that the loss surfaces of many neural networks have infinite spurious local minima, which are defined as the local minima with higher empirical risks than the global minima. Our result holds for any neural network with arbitrary depth and arbitrary piecewise linear activation functions (excluding linear functions) under most loss functions in practice with some mild assumptions. This result demonstrates that the networks with piecewise linear activations possess substantial differences to the well-studied linear neural networks. Essentially, the underlying assumptions for the above result are consistent with most practical circumstances where the output layer is narrower than any hidden layer. In addition, the loss surface of a neural network with piecewise linear activations is partitioned into multiple smooth and multilinear open cells by nondifferentiable boundaries. The constructed spurious local minima are exactly connected with each other in one cell by a continuous path, on which the empirical risk is invariant. We further prove that within every cell of a one-hidden-layer network, local minima are equally good, and also, they are all global minima in the cell.
> > >
> > >
> > >
> > > Q1b): Another point of exaggeration: the abstract first says "for any neural network with arbitrary depth". And later talks about "big picture". Only at the end of abstract mentions "1-hidden layer" for the second property --which is the most non-trivial one.  I think this might be misleading; readers would think the whole paper is about deep-net, and only a "minor result" is for 1-hidden layer. I did not check the paper again, but this needs to be clarified in the paper as well.
> > >
> > > A1b): Thanks and revised accordingly. The abstract in the final version has been updated to avoid misunderstanding. We have also relocated the restriction just besides the second property (please refers to A1.3).
> > >
> > > Moreover, the restriction of “one hidden layer” for Theorem 2 was stated in the original submission. For more details, please see (1) the paragraph for “Every local minimum is globally minimal within a cell” in the introduction on p. 2, (2) Theorem 2 on p. 7, and (3) the conclusion on p. 9.
> > >
> > >
> > >
> > > Q2.1: As for extension. "training neural networks is increasingly difficult when the networks turn deeper". This sentence seems to suggest that depth will make problem more difficult, so proving positive result is more difficult. This is not an argument for generating negative result.
> > >
> > > A2.1: We respectfully argue that the negative result for one-hidden-layer neural networks does not imply that deep neural networks have spurious local minima. It is correct to use one counterexample to prove a proposition is wrong. However, such a counterexample cannot be straightforwardly applied to a different condition. Large amounts of empirical evidence supports that the loss surface of deep networks could be substantially different from the shallow counterpart. Therefore, it is neither natural nor trivial to show deep networks can inherit a specific property of shallow networks.
> > >
> > > Besides, we respectfully note that the full context here is “Empirical results have overwhelmingly suggested that the increase of the depth of neural networks may substantially improve the performance. Additionally, training neural networks is increasingly difficult when the networks turn deeper.” These two sentences collectively explain the situations of shallow and deep neural networks could be substantially different.
> > >
> > >
> > >
> > > Q2.2: Thanks for explaining "Forcing the flow". The idea is natural (not meaning it is not good), as for piecewise linear one needs to utilize the linear part to create bad local minima. This is why [R1] is nontrivial: since sub-optimal local minima can exist for ReLU due to piecewise linear nature, thus it tried to show such bad cases are rare.
> > >
> > > A2.2: We appreciate the significant contributions made by [R1] and will duly cite [R1] in the final version. However, we would love to note that [R1] and this paper support each other to be more convincing, but not undermine the merit of the other side. For details, please refer to A1.2.

---

> > > ### Author Response · Authors · 2019-11-14
> > > **Re: Some questions remain; "big picture" somewhat oversold (Part 3 of 3)**
> > >
> > > Q2.3: Mathematically speaking, I agree there is a difference between deep and shallow. And I do appreciate the effort to really prove it. But the explanation is not for the technical nontriviality of proving "deep" instead of shallow.
> > >
> > > A2.3: Thank you for the recognization of our effort. We would love to clarify that the technical nontriviality mainly comes from the difficulty of the constructions to force the flow to go through the same linear parts of the activations after the first hidden layer.
> > >
> > >
> > >
> > > Other discussion:
> > >
> > >
> > >
> > > Q3.1: For A3(c): "empirical risk  is convex with respect to a variable mapped from the weights". "the local optimality is maintained under the constructed mapping"
> > >
> > > This is saying "convexity is kept after mapping" means "analogous convexity". This already exists in the original linear network, and this property was mentioned in, e.g., 1702.08580.  I still think calling "analogous convexity" is somewhat misleading. Why just say "every local min is global min"?
> > >
> > > A3.1: Thanks and revised accordingly. We have updated our manuscript to use "every local minimum is a global minimum within a cell".
> > >
> > > Besides, the proofs for linear networks cannot be transplanted here. Instead, we develop a novel and non-trivial approach (please see A1.1 and A1.2 for more detail).
> > >
> > >
> > >
> > > Q3.2: "we respectfully argue that “nonlinearity” is more common to refer to the difference from linear functions, which has been also used by Yun et al. (2019b)."
> > >
> > > I have to say it is not common. Yun et al. (2019b) analyzed a much bigger class of neurons than this paper. This paper just analyzed piecewise linear.
> > >
> > > In a field with overstatements, I would suggest using more precise title, like "the loss surface of neural networks with piecewise linear activations".
> > >
> > > A3.2: Thanks and revised accordingly. We have updated our manuscript to use “Piecewise linear activations substantially shape the loss surface of neural network”.

---

### Public Comment · ~Micah_Goldblum1 · 2019-11-08
**An Interesting Connection**

Hi Authors,
Thank you for your interesting paper.  I noticed that your theoretical result concerning the existence of suboptimal local minima is very similar to Theorem 1 in our work [1], which also contains an empirical analysis of suboptimal local minima.  Please consider mentioning the relationship with our work in your next version.  We will cite your work in our next version.

[1] https://arxiv.org/abs/1910.00359

---

> ### Author Response · Authors · 2019-11-09
> **We will discuss the relationship and cite your paper**
>
> Hi Micah,
>
> Thank you very much for bringing your interesting paper to our attention. We acknowledge the similarities shared by our papers. The final version will duly cite your paper. However, there are also substantial differences regarding the relevant parts. We will discuss the relationship between our papers below and the final version will be revised accordingly.
>
> Theorem 1 in [1] constructs some local minima of (nonlinear) multi-layer perceptrons (MLPs) that represent the hypotheses represented by the local minima of linear models. However, [1] does not rigorously prove that these constructed local minima are spurious/suboptimal. By contrast, Theorem 1 in our paper first constructs local minima of nonlinear networks based on linear networks and then proves that they are spurious/suboptimal.
>
> Besides, we really appreciate your experimental results presented in [1]. The result clearly demonstrates that MLPs have spurious/suboptimal local minima, which empirically justifies our theoretical findings.
>
> Kind regards,
> The authors
>
> [1] Micah Goldblum, Jonas Geiping, Avi Schwarzschild, Michael Moeller, Tom Goldstein. Truth or backpropaganda? An empirical investigation of deep learning theory. arXiv preprint arXiv:1910.00359, 2019.

---

### Decision · Program_Chairs · 2019-12-19

**Decision:**

Accept (Poster)

**Comment:**

Quoting R3: "This paper studies the theoretical property of neural network's loss surface. The main contribution is to prove that the loss surface of every neural network (with arbitrary depth) with piecewise linear activations has infinite spurious local minima."

There were split reviews, with two reviewers recommending acceptance and one recommending rejection.  During a robust rebuttal and discussion phase, both R2 and R3's appreciation for the work was strengthened.  The authors also provided a robust response to R1, whose main concerns included (i) that the paper's analysis is limited to piecewise linear activation functions, (ii) technical questions about the difficulty of proving theorem 2, which appear to have been answered in the discussion, and (iii) concerns about the strength of the language employed.

On the balance, the reviewers were positively impressed with the relevance of the theoretical study and its contributions.  Genuine shortcomings and misunderstandings were systematically resolved during the rebuttal process.